# Exactly solvable conformal field theories

Sylvain Ribault

CEA Saclay, Institut de Physique Théorique

`sylvain.ribault@ipht.fr`

November 27, 2024

**Abstract**

We review 2d CFT in the bootstrap approach, and sketch the known exactly solvable CFTs with no extended chiral symmetry: Liouville theory, (generalized) minimal models, limits thereof, and loop CFTs, including the $O(n)$, Potts and $PSU(n)$ CFTs.

Exact solvability relies on local conformal symmetry, and on the existence of degenerate fields. We show how these assumptions constrain the spectrum and correlation functions. We discuss how crossing symmetry equations can be solved analytically and/or numerically, leading to analytic expressions for structure constants in terms of the double Gamma function.

In the case of loop CFTs, we sketch the corresponding statistical models, and derive the relation between statistical and CFT variables. We review the resulting combinatorial description of correlation functions, and discuss what remains to be done for solving the CFTs.

# 0 Introduction

**Exact solvability in CFT**

In 1997, di Francesco, Mathieu and Sénéchal wrote a book called Conformal field theory [1], which dealt almost exclusively with conformal field theory *in two dimensions*. Today, a book with that title would be expected to also cover CFT in $d \geq 3$. However, when it comes to CFTs that we can solve exactly, we are back to $d = 2$, because exact solvability relies on 2 ingredients:

- An infinite-dimensional algebra of conformal transformations.

- Degenerate fields: fields whose OPEs with other fields have finitely many terms, modulo conformal symmetry.

This gives us access to powerful bootstrap techniques, analytic and numerical. But even in $d = 2$, solvable CFTs are very special cases. Starting from a simple CFT and following a renormalization group flow, we can easily end up at a fixed point whose CFT is intractable.

In $d \geq 3$, the conformal algebra is finite-dimensional. Degenerate fields can however be constructed, under the name of weight-shifting operators [2]. Under an OPE with a weight-shifting operator, the conformal dimension of a field is shifted by an integer. But except in particular CFTs such as generalized free theories, the spectrum is not stable under such shifts. So weight-shifting operators cannot be consistenly added to the CFT, and do not constrain it, although they are still useful for studying conformal blocks. The main bootstrap techniques in $d \geq 3$ are numerical instead, and rely on the assumption of unitarity. This not only restricts the CFTs that can be addressed, but also leads to inequalities rather than equalities, making it harder to reach exact results.

**Solvable CFTs**

We will first deal with CFTs that are considered solved, in that we exactly know the 3-point structure constants, and can compute any correlation function to arbitrary precision:

- Liouville theory,

- (generalized) minimal models, and limits thereof.

We will also propose a unified treatment of loop CFTs, which are not solved but are believed to be solvable, because we have sufficiently many exact results. By loop CFTs we mean a large class of CFTs, with various global symmetry groups, which describe critical limits of various statistical models in various phases. The known solvable CFTs have many applications to statistical physics, in particular to percolation, to polymers, and more recently to active hydraulics [3]. Liouville theory and minimal models have well-known applications to quantum gravity in 2d. Applications of non-diagonal solvable CFTs such as loop CFTs to quantum gravity in 3d are limited by the bound on the central charge $\Re c < 13$. (See Section 2.1.3.) A CFT that would be holographically dual to quantum gravity in 3d Anti-de Sitter space would need to exist for $c \to +\infty$, while for 3d de Sitter space we would need $\Re c = 13$ [4, 5].

There are other solved or solvable CFTs that we will not mention, because they have extended chiral algebras, which imply extra local symmetries in addition to conformal symmetry. Some CFTs of that type can be solved using the methods that we will cover, for example certain Wess–Zumino–Witten models.

## Solving CFTs

When a CFT is solvable, it can typically be solved in several different ways. In this text, we will only use the conformal bootstrap approach, and omit other historically relevant techniques such as Coulomb gas integrals, and the modular bootstrap. We believe that the conformal bootstrap approach is technically simpler, more general, and more powerful. In particular, our main objects are correlation functions on the sphere, and we do not have a rigid notion of which fields belong to a CFT. For example, in the Ising model, we can introduce disorder fields or compute cluster connectivities, without having to worry about modular invariance of the torus partition function.

Solving a CFT means determining its spectrum and structure constants. We deduce spectrums from assumptions such as

- the existence of degenerate fields,

- singe-valuedness of correlation functions,

- analyticity in the central charge or in conformal dimensions, which in particular allows us to find CFTs as limits of sequences of other CFTs.

These assumptions also lead to constraints on structure constants, which may or may not be enough for determining them analytically. If they are not enough, we can determine structure constants numerically by solving crossing symmetry equations. And from sufficiently precise numerical results, we can deduce analytic formulas.

## This text's approach

We strive to give simple derivations of non-trivial results, without introducing too many auxiliary objects that are later discarded. The calculations are not all done in detail in this text, but they are doable with reasonable effort. Readers who want formal exercises are directed to earlier texts in the same spirit: the minimal review [6], and the more extended review [7]. However, these texts do not cover loop CFTs.

Technical terms that are defined in this text can be found in the index. We also use standard terms that we do not define, because adequate definitions are easily found in Wikipedia or other resources. For example, the definitions of an *irreducible representation*, an *indecomposable representation* or a *projective representation* are found in Wikipedia as the first results of searches for these terms.

## Acknowledgements

- This text is based on lecture notes for a course given at IPhT Saclay in March-April 2024. Video recordings are available on IPhT-TV. I am grateful to the course organizers Riccardo Guida, Pierfrancesco Urbani and Monica Guica.

- I would like to thank Linnea Grans-Samuelsson, Jesper Jacobsen, Santiago Migliaccio, Nikita Nemkov, Rongvoram Nivesvivat, Marco Picco, Paul Roux, Hubert Saleur, and Raoul Santachiara for collaboration on loop CFTs and related subjects.

- I am grateful to Victor Godet, Jesper Jacobsen, Paul Roux, Ingo Runkel, Hubert Saleur, Xi Yin and Bernardo Zan for helpful discussions and correspondence.

- I would like to thank Max Downing, Kay Wiese and Rongvoram Nivesvivat for comments and suggestions on the manuscript.

- This work is partly a result of the project ReNewQuantum, which received funding from the European Research Council.

# 1   Basics of 2d CFT

In this section we introduce conformal symmetry, fields, operator product expansions, and correlation functions. In particular, we emphasize the 2 ingredients of exact solvability: *local* conformal symmetry and *degenerate* fields. We begin with the Virasoro algebra, which describes local conformal transformations.

## 1.1   The Virasoro algebra and its representations

### 1.1.1   The Virasoro algebra

Let us consider the Riemann sphere $\overline{\mathbb{C}} = \mathbb{C} \cup \{\infty\}$, equipped with the metric $ds^2 = dzd\bar{z}$. By definition, a **conformal transformation** of a Riemannian manifold is a transformation that preserves angles. On any open subset of the Riemann sphere, any holomorphic map $z \to f(z)$ is conformal, because it transforms the metric into itself, up to a scalar factor:

$$dzd\bar{z} \to df d\bar{f} = |f'(z)|^2 dzd\bar{z} \ . \tag{1.1}$$

Conversely, any conformal map is holomorphic. In a quantum field theory on the Riemann sphere, states live on a constant time slice $|z| = 1$, so we consider conformal transformations of that circle, equivalently of $\mathbb{C}^* = \mathbb{C}\backslash\{0\}$. Let the **Witt algebra** be the algebra of infinitesimal conformal transformation of $\mathbb{C}^*$: this algebra is infinite-dimensional, with the basis

$$(\ell_n)_{n \in \mathbb{Z}} \quad \text{with} \quad \ell_n = -z^{n+1}\frac{\partial}{\partial z} \ , \tag{1.2}$$

and the commutation relations

$$[\ell_n, \ell_m] = (n - m)\ell_{m+n} \ . \tag{1.3}$$

The generators of the Witt algebra include the translation generator $\ell_{-1} = -\frac{\partial}{\partial z}$, and the dilation generator $\ell_0 = -z\frac{\partial}{\partial z}$. In fact, $(\ell_{-1}, \ell_0, \ell_1)$ generate the infinitesimal conformal transformations of the Riemann sphere. The corresponding Lie group is the group of **global conformal transformations** $PSL_2(\mathbb{C})$, whose elements act as

$$z \mapsto \frac{az + b}{cz + d} \quad , \quad (a, b, c, d \in \mathbb{C}, \ ad - bc \neq 0) \ . \tag{1.4}$$

Now, in a quantum theory, symmetries act projectively on states. And projective representations of an algebra are equivalent to representations of that algebra's central extension. Therefore, the algebra that describes local conformal transformations in conformal field theory is the Witt algebra's central extension, called the Virasoro algebra. The **Virasoro algebra** $\mathfrak{V}$ has generators $(L_n)_{n \in \mathbb{Z}}$ that correspond to the Witt algebra generators, plus a central generator $\mathbf{1}$. The commutation relations are

$$[\mathbf{1}, L_n] = 0 \quad , \quad \boxed{[L_n, L_m] = (n - m)L_{n+m} + \frac{c}{12}(n - 1)n(n + 1)\delta_{n+m,0}\mathbf{1}} \ . \tag{1.5}$$

The **central charge** $c \in \mathbb{C}$ is a fundamental parameter not only of the Virasoro algebra, but also of any conformal field theory built on that algebra.

### 1.1.2 Highest-weight representations

In a conformal field theory, the space of states is a representation of the Virasoro algebra, and can be decomposed into indecomposable representations. But which indecomposable representations are physically relevant? To answer this question, we will focus on the properties of the dilation generator $L_0$ of the Virasoro algebra. Conceptually, this is because $L_0$ can be interpreted as the energy operator. Technically, this is because $L_0$ controls the convergence of operator product expansions, as we will see in Section 1.2.3. More specifically, a necessary condition for convergence is that the eigenvalues of $L_0$ be bounded from below in any indecomposable representation. We will always assume that this condition holds.

The eigenvalues of $L_0$ are called **conformal dimensions**. Under the action of the Virasoro generator $L_n$, the conformal dimension decreases by $n$. For any vector $V$ in a representation of the Virasoro algebra, we indeed have

$$L_0 V = \Delta V \quad \Rightarrow \quad L_0 L_n V = L_n L_0 V + [L_0, L_n] V = (\Delta - n) L_n V \ . \tag{1.6}$$

In an indecomposable representation $\mathcal{R}$, the $L_0$-eigenstate with the lowest eigenvalue must therefore be annihilated by $L_{n>0}$. This eigenstate is a **primary state**, where we define a primary state $V_\Delta$ of conformal dimension $\Delta$ by

$$\boxed{L_0 V_\Delta = \Delta V_\Delta \quad , \quad L_{n>0} V_\Delta = 0} \ . \tag{1.7}$$

Let us introduce a basis of **creation operators**,

$$\mathcal{L} = \left\{ \prod_{i=1}^{k} L_{-n_i} \right\}_{k \in \mathbb{N}, \ 0 < n_1 \leq \cdots \leq n_k} = \left\{ 1, L_{-1}, L_{-1}^2, L_{-2}, \cdots \right\} \ . \tag{1.8}$$

Any state of the type $LV$ with $L \in \mathcal{L}$ is called a **descendant state** of the primary state $V$. Let $N = |L| = \sum_{i=1}^{k} n_i \in \mathbb{N}$ be the **level** of that state, then its conformal dimension is $\Delta + N$, and we write

$$\mathcal{L}_N = \left\{ L \in \mathcal{L} \middle| |L| = N \right\} \ . \tag{1.9}$$

By extension, a linear combination of descendant states is also called a descendant state.

A primary state $V \in \mathcal{R}$ generates a subrepresentation $\mathcal{R}_V = \mathrm{Span}(LV)_{L \in \mathcal{L}} \subset \mathcal{R}$, which is the space of its descendant states. If the states $(LV)_{L \in \mathcal{L}}$ are linearly independent, then $\mathcal{R}_V$ is called the **Verma module** $\mathcal{V}_\Delta$. If not, $\mathcal{R}_V$ is a quotient of $\mathcal{V}_\Delta$ by a subrepresentation: such quotients are called **highest-weight representations**. In such a representation, $L_0$ is diagonalizable. Let us sketch a Verma module by displaying all its basis states up to the level $N = 4$, with arrows representing the action of the Virasoro

generators $L_{-1}, L_{-2}, L_{-3}, L_{-4}$:

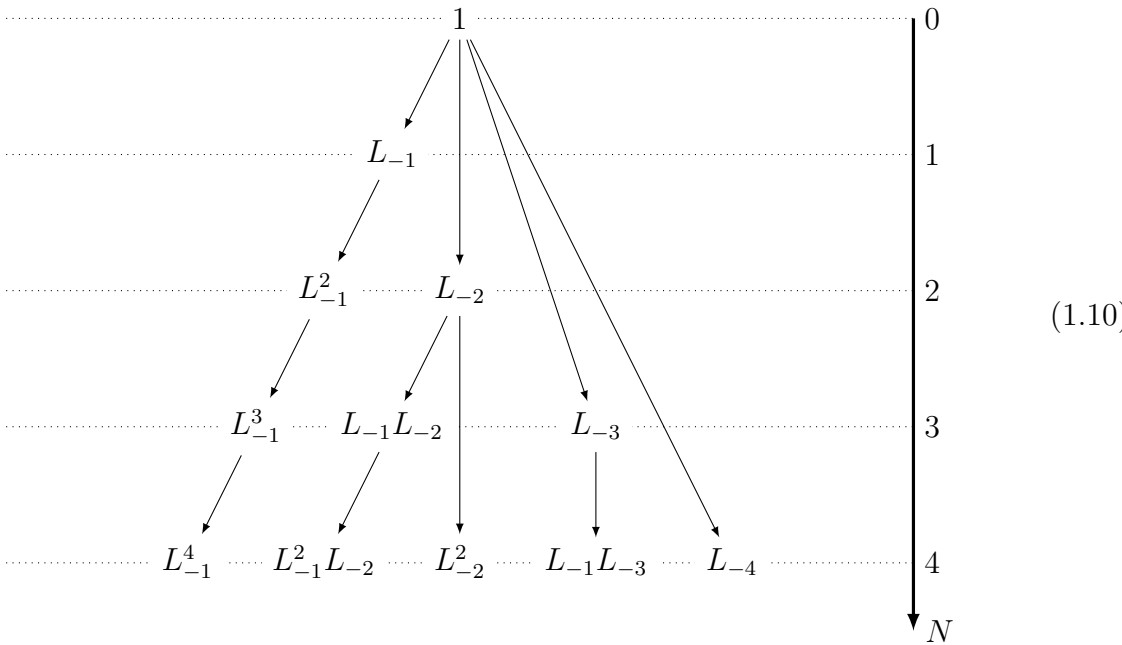

$$(1.10)$$

States such as $L_{-2}L_{-1}V = L_{-1}L_{-2}V - L_{-3}V$ are linear combination of elements of our basis, and are therefore not displayed.

### 1.1.3 Null vectors of Verma modules

In a representation of the Virasoro algebra, we define a **null vector** or singular vector as a primary state that is a descendant of another primary state. Null vectors play an important structural role: a highest-weight representation is irreducible if and only if it has no null vector.

Let us look for null vectors in the Verma module $\mathcal{V}_\Delta$. This can be done level by level, starting with level 1. That level is generated by a single state $L_{-1}V_\Delta$. To determine whether that state is primary, let us determine how it behaves under the action of $L_{n\geq 1}$:

$$L_n L_{-1} V_\Delta = [L_n, L_{-1}]V_\Delta = (n+1)L_{n-1}V_\Delta = \begin{cases} 0 & \text{if } n \geq 2 , \\ 2\Delta V_\Delta & \text{if } n = 1 . \end{cases} \quad (1.11)$$

Therefore, $\mathcal{V}_\Delta$ has a null vector at level 1 if and only if $\Delta = 0$. Next, consider the case of level 2, where we write states as

$$X = \left( a^{L_{-1}^2} L_{-1}^2 + a^{L_{-2}} L_{-2} \right) V_\Delta , \quad (1.12)$$

where $a^{L_{-1}^2}$ and $a^{L_{-2}}$ are complex coefficients. It is easy to show that $L_{n\geq 3}X = 0$, and we compute

$$L_1 X = \left( (4\Delta + 2)a^{L_{-1}^2} + 3a^{L_{-2}} \right) L_{-1}V_\Delta , \quad (1.13\text{a})$$

$$L_2 X = \left( 6\Delta a^{L_{-1}^2} + (4\Delta + \tfrac{1}{2}c)a^{L_{-2}} \right) V_\Delta . \quad (1.13\text{b})$$

Therefore, the conditions $L_1 X = L_2 X = 0$ for $X$ to be a null vector boil down to a system of 2 linear equations for the coefficients $(a^{L_{-1}^2}, a^{L_{-2}})$. The vanishing of that

system's determinant determines the conformal dimension $\Delta$ as a function of the central charge $c$,

$$\Delta = \frac{1}{16} \left( 5 - c \pm \sqrt{(c-1)(c-25)} \right) . \tag{1.14}$$

Solving CFTs will lead to fairly complicated formulas, which would become even more complicated if we tolerated a square root at this early stage. To get rid of the square root, we rewrite the central charge in terms of the parameter $\beta$ such that

$$\boxed{c = 1 - 6 \left( \beta - \beta^{-1} \right)^2} . \tag{1.15}$$

Then the conformal dimensions (1.14) become $\Delta = -\frac{1}{2} + \frac{3}{4}\beta^{\pm 2}$, and the corresponding null vectors are $X \propto \left( L_{-1}^2 - \beta^{\pm 2} L_{-2} \right) V_\Delta$. The price to pay for this simplification is that the 4 values $\beta, -\beta, \beta^{-1}, -\beta^{-1}$ all correspond to the same central charge.

More generally, there is an infinite family of conformal dimensions $(\Delta_{(r,s)})_{r,s \in \mathbb{N}^*}$ such that $\mathcal{V}_{\Delta_{(r,s)}}$ has a null vector $L_{\langle r,s \rangle} V_{\Delta_{(r,s)}}$ at level $N = rs$. Let us summarize the cases $N = 1, 2, 3, 4$ in these notations:

| $N$ | $(r,s)$ | $\Delta_{(r,s)}$ | $L_{\langle r,s \rangle}$ |
|---|---|---|---|
| 1 | $(1,1)$ | $0$ | $L_{-1}$ |
| 2 | $(2,1)$ | $-\frac{1}{2} + \frac{3}{4}\beta^2$ | $L_{-1}^2 - \beta^2 L_{-2}$ |
|   | $(1,2)$ | $-\frac{1}{2} + \frac{3}{4}\beta^{-2}$ | $L_{-1}^2 - \beta^{-2} L_{-2}$ |
| 3 | $(3,1)$ | $-1 + 2\beta^2$ | $L_{-1}^3 - 4\beta^2 L_{-1}L_{-2} + 2\beta^2(\beta^2+1)L_{-3}$ |
|   | $(1,3)$ | $-1 + 2\beta^{-2}$ | $L_{-1}^3 - 4\beta^{-2} L_{-1}L_{-2} + 2\beta^{-2}(\beta^{-2}+1)L_{-3}$ |
| 4 | $(4,1)$ | $-\frac{3}{2} + \frac{15}{4}\beta^2$ | $L_{-1}^4 - 10\beta^2 L_{-1}^2 L_{-2} + 2\beta^2 \left( 12\beta^2 + 5 \right) L_{-1}L_{-3}$ $+9\beta^4 L_{-2}^2 - 6\beta^2 \left( 6\beta^4 + 4\beta^2 + 1 \right) L_{-4}$ |
|   | $(2,2)$ | $\frac{3}{4} \left( \beta - \beta^{-1} \right)^2$ | $L_{-1}^4 - 2 \left( \beta^2 + \beta^{-2} \right) L_{-1}^2 L_{-2} + \left( \beta^2 - \beta^{-2} \right)^2 L_{-2}^2$ $+2 \left( 1 + (\beta + \beta^{-1})^2 \right) L_{-1}L_{-3} - 2 \left( \beta + \beta^{-1} \right)^2 L_{-4}$ |
|   | $(1,4)$ | $-\frac{3}{2} + \frac{15}{4}\beta^{-2}$ | $L_{-1}^4 - 10\beta^{-2} L_{-1}^2 L_{-2} + 2\beta^{-2} \left( 12\beta^{-2} + 5 \right) L_{-1}L_{-3}$ $+9\beta^{-4} L_{-2}^2 - 6\beta^{-2} \left( 6\beta^{-4} + 4\beta^{-2} + 1 \right) L_{-4}$ |

$$(1.16)$$

To write $\Delta_{(r,s)}$ for any $r, s \in \mathbb{N}^*$, we rewrite the conformal dimension in terms of the momentum $P$ defined by

$$\boxed{\Delta = \frac{c-1}{24} + P^2} . \tag{1.17}$$

We can rewrite Verma modules as $\mathcal{V}_P = \mathcal{V}_{-P}$, where the parameter $P$ is redundant due to the invariance of $\Delta$ under the **reflection** $P \to -P$. The dimensions $\Delta_{(r,s)}$ correspond to the momentums

$$\boxed{P_{(r,s)} = \frac{1}{2} \left( r\beta - s\beta^{-1} \right)} , \tag{1.18}$$

and it follows that the null vector $L_{\langle r,s \rangle} V_{\Delta_{(r,s)}}$ is a primary state of dimension

$$\boxed{\Delta_{(r,s)} + rs = \Delta_{(r,-s)}} , \tag{1.19}$$

We will provide a derivation of $P_{(r,s)}$ using operator product expansions in Section 1.2.4. On the other hand, the expressions of $L_{\langle r,s \rangle}$ for $rs \geq 3$ are complicated and not particularly useful, see [8] or [1, Appendix 8A].

For $\beta^2 \in \mathbb{C}\backslash\mathbb{Q}$, the Verma module $\mathcal{V}_\Delta$ has a null vector if and only if $\Delta = \Delta_{(r,s)}$ for some $r, s \in \mathbb{N}^*$, in which case it has only 1 null vector. For $\beta^2 \in \mathbb{Q}$ however, a Verma module can have several null vectors. If $\beta^2 = \frac{q}{p}$ with $p, q$ coprime integers, we indeed have the identity

$$\Delta_{(r+p, s+q)} = \Delta_{(r,s)} \ , \tag{1.20}$$

which together with $\Delta_{(r,s)} = \Delta_{(-r,-s)}$ implies coincidences of the type $\Delta_{(r,s)} = \Delta_{(r',s')}$ with $r', s' \in \mathbb{Z}$. Given $r_1, s_1 \in \mathbb{N}^*$, the reducible Verma module $\mathcal{V}_{\Delta_{(r_1,s_1)}}$ has a null vector $L_{\langle r_1,s_1 \rangle}V_{\Delta_{(r,s)}}$. If $pq > 0$, or if $pq < 0$ with $\left\lfloor \frac{r_1}{|p|} \right\rfloor \neq \left\lfloor \frac{s_1}{|q|} \right\rfloor$ and $\left\lceil \frac{r_1}{|p|} \right\rceil \neq \left\lceil \frac{s_1}{|q|} \right\rceil$, then the dimension $\Delta_{(r_1,-s_1)}$ (1.19) of this null vector is of the type $\Delta_{(r_2,s_2)}$ with $r_2, s_2 \in \mathbb{N}^*$, leading to another null vector $L_{\langle r_2,s_2 \rangle}L_{\langle r_1,s_1 \rangle}V_{\Delta_{(r,s)}}$. It turns out that all null vectors of Verma modules are of the type $L_{\langle r_1,s_1 \rangle}V_{\Delta_{(r,s)}}$ or $L_{\langle r_2,s_2 \rangle}L_{\langle r_1,s_1 \rangle}V_{\Delta_{(r,s)}}$. In particular, given a null vector of the type $L_{\langle r_3,s_3 \rangle}L_{\langle r_2,s_2 \rangle}L_{\langle r_1,s_1 \rangle}V_{\Delta_{(r_1,s_1)}}$,

$$\exists r_4, s_4 \in \mathbb{N}^* \ , \ L_{\langle r_3,s_3 \rangle}L_{\langle r_2,s_2 \rangle}L_{\langle r_1,s_1 \rangle}V_{\Delta_{(r_1,s_1)}} = L_{\langle r_4,s_4 \rangle}V_{\Delta_{(r_4,s_4)}} \ . \tag{1.21}$$

To see this, let $\epsilon, \eta \in \{+,-\}$ be the signs such that $P_{(-r_1,s_1)} = \epsilon P_{(r_2,s_2)}$ and $P_{(r_2,-s_2)} = \eta P_{(r_3,s_3)}$, and let $r_4 = |\epsilon r_1 - r_2 + \eta r_3|$ and $s_4 = |\epsilon s_1 + s_2 + \eta s_3|$, then $\sum_{i=1}^{3} r_i s_i = r_4 s_4$, i.e. the 2 null vectors in Eq. (1.21) have the same level.

To summarize, numbers of null vectors depend a lot on $\beta^2$. If $\beta^2 < 0$, then $\{\Delta_{(r,s)}\}_{r,s\in\mathbb{N}^*}$ is bounded from above, and Eq. (1.19) implies that a Verma module can only have finitely many null vectors. If $\beta^2 \in \mathbb{Q}_{>0}$, then by Eq. (1.20) the existence of a null vector implies the existence of infinitely many others:

| Value of $\beta^2$ | $\mathbb{C}\backslash\mathbb{Q}$ | $\mathbb{Q}_{>0}$ | $\mathbb{Q}_{<0}$ | |
|---|---|---|---|---|
| Value of $c$ | generic | $c \leq 1, c \in \mathbb{Q}$ | $c \geq 25, c \in \mathbb{Q}$ | (1.22) |
| #null vectors in $\mathcal{V}_{\Delta_{(r,s)}}$ | 1 | $\infty$ | finite | |

To illustrate the 2 cases $\beta^2 \in \mathbb{Q}_{>0}$ and $\beta^2 \in \mathbb{Q}_{<0}$, let us plot at each $(r,s) \in \{1,\ldots,10\}^2$ a dot whose size increases with $\Delta_{(r,s)}$, together with the vector $(p,q)$ of Eq. (1.20) as a red arrow:

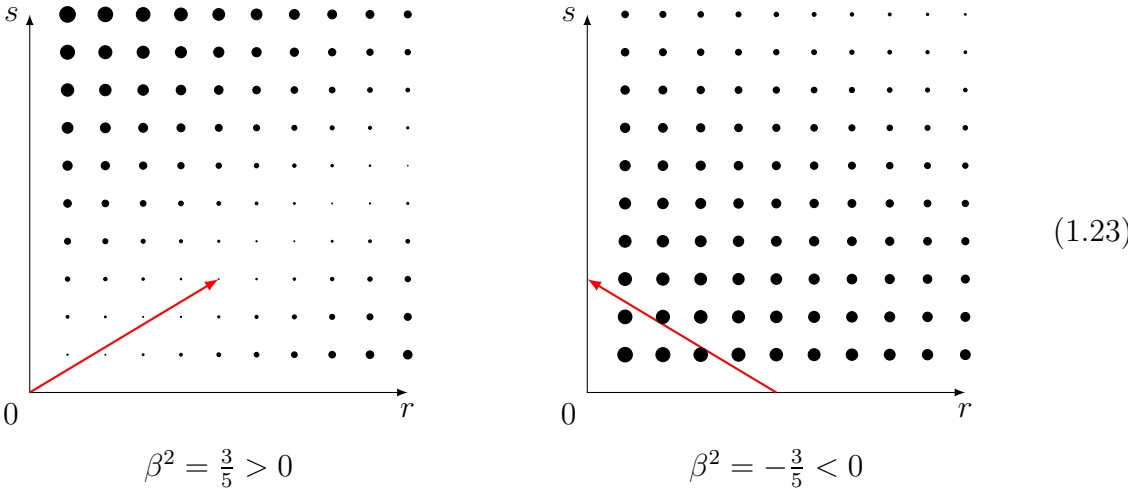

$$\beta^2 = \frac{3}{5} > 0 \qquad\qquad \beta^2 = -\frac{3}{5} < 0 \qquad\qquad (1.23)$$

### 1.1.4 Vanishing and non-vanishing null vectors

When a representation $\mathcal{R}$ has a null vector $X = LV$, this allows us to define a quotient representation $\frac{\mathcal{R}}{\mathcal{R}_X}$ by setting the null vector to zero. By an abuse of terminology, we say that $\frac{\mathcal{R}}{\mathcal{R}_X}$ has a **vanishing null vector**, which really means that its primary state $V$ obeys the **null vector equation** $LV = 0$. On the other hand, $\mathcal{R}$ itself has a non-vanishing null vector $X$.

If $\mathcal{R}$ has several null vectors, we can define various quotient representations by setting some null vectors to zero. A **degenerate representation** is a representation that has at least 1 vanishing null vector. A degenerate representation is partly degenerate if it has some non-vanishing null vectors, and **fully degenerate** if all null vectors vanish. Partly degenerate representations play an important role in CFTs with extended chiral algebras, such as conformal Toda theories [9].

For $r, s \in \mathbb{N}^*$, the level-$rs$ null vector $L_{\langle r,s \rangle} V_{\Delta_{(r,s)}} \in \mathcal{V}_{\Delta_{(r,s)}}$ is a primary state of dimension $\Delta_{(r,-s)}$, according to Eq. (1.19). Together with its descendants, this state generates a Verma module $\mathcal{V}_{\Delta_{(r,-s)}} \subset \mathcal{V}_{\Delta_{(r,s)}}$. Setting the null vector to zero, we obtain the degenerate representation

$$\mathcal{R}^d_{\langle r,s \rangle} = \frac{\mathcal{V}_{\Delta_{(r,s)}}}{\mathcal{V}_{\Delta_{(r,-s)}}} \ . \tag{1.24}$$

If $\beta^2 \in \mathbb{Q}$, the Verma module $\mathcal{V}_{\Delta_{(r,s)}}$ in general has several null vectors, and $\mathcal{R}^d_{\langle r,s \rangle}$ may be partly degenerate. Let us call $\mathcal{R}^f_{\langle r,s \rangle}$ the fully degenerate quotient of $\mathcal{V}_{\Delta_{(r,s)}}$: this quotient can generally not be obtained by setting 1 null vector to zero, but it is always enough to set 2 null vectors to zero, with all other null vectors being descendants of these two. Schematically,

$$\mathcal{R}^f_{\langle r,s \rangle} = \frac{\mathcal{V}_{\Delta_{(r,s)}}}{\mathcal{V}_{\Delta_{(r_1,-s_1)}} + \mathcal{V}_{\Delta_{(r_2,-s_2)}}} \ . \tag{1.25}$$

We define the **degenerate state** $V^d_{\langle r,s \rangle}$ and **fully degenerate state** $V^f_{\langle r,s \rangle} = V^f_{\langle r_1,s_1 \rangle} = V^f_{\langle r_2,s_2 \rangle}$ as the primary states of the representation $\mathcal{R}^d_{\langle r,s \rangle}$ and $\mathcal{R}^f_{\langle r,s \rangle}$. These states obey the null vector equation

$$L_{\langle r,s \rangle} V^d_{\langle r,s \rangle} = L_{\langle r,s \rangle} V^f_{\langle r,s \rangle} = 0 \ . \tag{1.26}$$

We use the notation $V_{\Delta_{(r,s)}}$ for a primary state that has the same dimension as $V^d_{\langle r,s \rangle}$ and $V^f_{\langle r,s \rangle}$, but does not necessarily obey the null vector equation.

A basis of the degenerate representation $\mathcal{R}^d_{\langle r,s \rangle}$ is formed by the descendant states $LV^d_{\langle r,s \rangle}$ such that the creation operator $L$ does not involve the Virasoro generator $L_{-rs}$. In the simplest example $(r, s) = (1, 1)$, let us sketch $\mathcal{R}^d_{\langle 1,1 \rangle}$ up to the level $N = 4$, so that it may be compared to a Verma module (1.10):

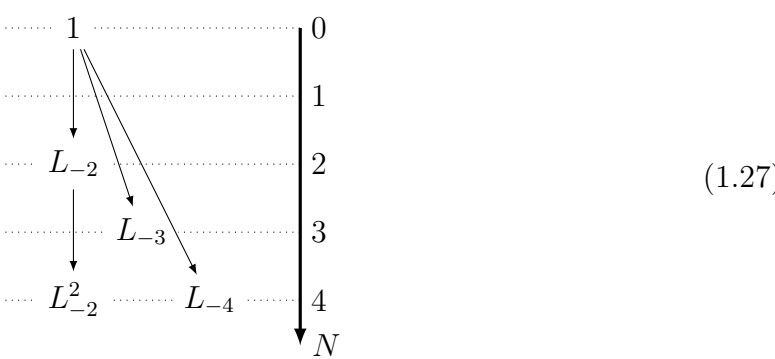

$$\tag{1.27}$$

### 1.1.5   The Shapovalov form

We will now introduce the Shapovalov form, which will be useful for computing operator product expansions and conformal blocks. We will see that the Shapovalov from has zeros related to null vectors, and this will lead to poles of the operator product expansions and conformal blocks.

Let us define an involution on the Virasoro algebra by

$$L_n^* = L_{-n} \ . \tag{1.28}$$

This involution is a Lie algebra automorphism, as it obeys $[L_m, L_n]^* = -[L_m^*, L_n^*]$. It can be extended to the universal enveloping algebra by $(LL')^* = L'^* L^*$. Using this involution, we define the **Shapovalov form** on any highest-weight representation as the symmetric bilinear form such that

$$\left(LV_\Delta \middle| L'V_\Delta\right) = \begin{cases} 0 & \text{if } |L| \neq |L'| \ , \\ S_{L,L'}(\Delta) & \text{if } |L| = |L'| \ , \quad \text{where} \quad L^* L' V_\Delta = S_{L,L'}(\Delta) V_\Delta \ . \end{cases} \tag{1.29}$$

At each level $N \in \mathbb{N}$, the Shapovalov form is described by a matrix that depends on $\Delta$ and $c$. In the cases $N = 0, 1, 2$, these matrices are

$$S_{1,1} = 1 \ , \quad S_{L_{-1},L_{-1}} = 2\Delta \ , \quad \begin{bmatrix} S_{L_{-1}^2,L_{-1}^2} & S_{L_{-1}^2,L_{-2}} \\ S_{L_{-1}^2,L_{-2}} & S_{L_{-2},L_{-2}} \end{bmatrix} = \begin{bmatrix} 4\Delta(2\Delta+1) & 6\Delta \\ 6\Delta & 4\Delta + \frac{c}{2} \end{bmatrix} \ . \tag{1.30}$$

From its definition, it follows that the Shapovalov form has zeros:

$$\forall L, L' \in \mathcal{L} \ , \quad S_{LL_{\langle r,s \rangle},L'}(\Delta_{(r,s)}) = 0 \ . \tag{1.31}$$

For generic values of the central charge, these are simple zeros. This implies that the inverse Shapovalov matrix at level $N$ has simple poles for $\Delta \in \left\{\Delta_{(r,s)}\right\}_{rs \leq N}$. In a basis of level-$rs$ creation operators that includes $L_{\langle r,s \rangle}$, the level-$rs$ inverse Shapovalov matrix behaves as

$$S_{L,L'}^{-1}(\Delta) \underset{\Delta \to \Delta_{(r,s)}}{=} \frac{1}{\Delta - \Delta_{(r,s)}} \frac{\delta_{L,L_{\langle r,s \rangle}} \delta_{L',L_{\langle r,s \rangle}}}{S'_{L_{\langle r,s \rangle},L_{\langle r,s \rangle}}(\Delta_{(r,s)})} + O(1) \ . \tag{1.32}$$

In fact, the level-$N$ inverse Shapovalov matrix can be computed explicitly in terms of $\left\{L_{\langle r,s \rangle}\right\}_{rs \leq N}$ [10].

Using the same coefficients (1.29), we can define a sesquilinear form instead of a bilinear form. This sesquilinear form is Hermitian if its coefficients are real, i.e. if $\Delta, c \in \mathbb{R}$. Our highest-weight representation is called **unitary** if that Hermitian form is positive definite. Unitarity plays an important role in some applications of CFT, but it has no bearing on exact solvability, so we will not elaborate further on that subject.

## 1.2   Fields and operator product expansions

The fundamental objects of conformal field theory are fields. In quantum field theory, all states can be obtained by acting on the vacuum state with fields: this is called the **state-field correspondence** or operator-state correspondence. In conformal field theory, the correspondence is bijective, and allows us to identify the space of fields with the **space of states** or **spectrum** — a vector space on which the Virasoro algebra acts. To a state $V$, the correspondence associates a field $V(z)$, which depends on a position $z \in \overline{\mathbb{C}}$ in the

Riemann sphere. Primary states, descendant states and degenerate states respectively give rise to **primary fields**, **descendant fields** and **degenerate fields**.

In the bootstrap approach, which is axiomatic rather than constructive, we need not define fields. Rather, we use fields as a convenient notation for stating properties of correlation functions, which we will introduce in Section 1.3.

### 1.2.1   The energy-momentum tensor

The dependence of fields on the coordinate $z$ is determined by the assumption that the Virasoro generator $L_{-1}$ generates translations, just like the Witt algebra generator $\ell_{-1}$ (1.2):

$$\boxed{\frac{\partial}{\partial z}V(z) = L_{-1}V(z)} . \tag{1.33}$$

Since fields depend on $z$, the Virasoro algebra that acts on fields also depends on $z$. This can be made explicit using the notation

$$L_n V(z) = L_n^{(z)}V(z) . \tag{1.34}$$

Applying Eq. (1.33) to $L_n^{(z)}V(z)$, we find how the Virasoro generator $L_n^{(z)}$ depends on $z$,

$$\frac{\partial}{\partial z}L_n^{(z)} = [L_{-1}^{(z)}, L_n^{(z)}] = -(n+1)L_{n-1}^{(z)} , \qquad (\forall n \in \mathbb{Z}) . \tag{1.35}$$

This can be rewritten as the $z$-independence of a formal Laurent series that combines all the generators $(L_n^{(z)})_{n\in\mathbb{Z}}$,

$$\frac{\partial}{\partial z}\sum_{n\in\mathbb{Z}}\frac{L_n^{(z)}}{(y-z)^{n+2}} = 0 , \qquad (\forall y \in \mathbb{C}\backslash\{z\}) . \tag{1.36}$$

The formal Laurent series is then called the **energy-momentum tensor**,

$$\boxed{T(y) = \sum_{n\in\mathbb{Z}}\frac{L_n^{(z)}}{(y-z)^{n+2}}} . \tag{1.37}$$

A priori, the energy-momentum tensor only makes sense when acting on a field $V(z)$,

$$T(y)V(z) = \sum_{n\in\mathbb{Z}}\frac{L_n V(z)}{(y-z)^{n+2}} \quad , \quad L_n V(z) = \frac{1}{2\pi i}\oint_z dy\,(y-z)^{n+1}T(y)V(z) . \tag{1.38}$$

However, we will shortly interpret $T(y)$ as a field, and $T(y)V(z)$ as an operator product expansion of 2 fields. The statement that $V_\Delta(z)$ is a primary field of dimension $\Delta$ is equivalent to the OPE

$$\boxed{T(y)V_\Delta(z) = \frac{\Delta}{(y-z)^2}V_\Delta(z) + \frac{1}{y-z}\frac{\partial}{\partial z}V_\Delta(z) + O(1)} , \tag{1.39}$$

where we used Eq. (1.33) and wrote the regular terms as $O(1)$.

### 1.2.2 Operator product expansions

The fundamental axiom of conformal field theory, which underlies the conformal bootstrap approach, is the existence of **operator product expansions (OPEs)**. This axiom states that there exists a bilinear product on the space of fields,

$$\boxed{V_1(z_1)V_2(z_2) = \sum_{k \in \mathcal{S}} C_{12}^k(z_1, z_2)V_k(z_2)} , \tag{1.40}$$

for some family $(V_k)_{k \in \mathcal{S}}$ of linearly independent states, and some functions $C_{12}^k(z_1, z_2)$ called **OPE coefficients**. Operator product expansions generally exist in quantum field theory, but they are particularly useful in conformal field theory because they are supposed to hold for any $z_1$ in a finite neighbourhood of $z_2$ (but $z_1 \neq z_2$), and not just asymptotically as $z_1 \to z_2$. The finite neighbourhood in question may depend on the correlation function in which we perform the OPE, and also on the basis $(V_k)_{k \in \mathcal{S}}$.

We further assume that the product of fields is commutative,

$$\boxed{V_1(z_1)V_2(z_2) = V_2(z_2)V_1(z_1)} . \tag{1.41}$$

This implies that OPEs are **commutative** and **associative**. However, the resulting constraints on the coefficients $C_{12}^i(z_1, z_2)$ are not particularly simple, because of having to choose a position ($z_2$ in our conventions) for the fields in the right-hand side of the OPE (1.40).

We also assume that there exists an **identity field** $I(z)$ whose OPE with any other field is trivial, namely $I(z_1)V(z_2) = V(z_2)$. (The corresponding state is called the vacuum state.) Let us consider the energy-momentum tensor $T$ as a field, then we have $T(y)I(z) = T(y) = T(z) + O(y - z)$. Comparing this with Eq. (1.38) applied to $I(z)$, we find

$$L_{n \geq -1}I(z) = 0 \quad , \quad T(z) = L_{-2}I(z) . \tag{1.42}$$

In particular, the identity field is a degenerate primary field of conformal dimension zero, with a vanishing null vector at level 1, i.e. $I(z) \propto V_{\langle 1,1 \rangle}^d(z)$. And from the expression of $T(z)$ as a descendant of that field, we deduce

$$T(y)T(z) = \frac{\frac{c}{2}}{(y - z)^4} + \frac{2T(z)}{(y - z)^2} + \frac{\partial T(z)}{y - z} + O(1) . \tag{1.43}$$

This OPE is equivalent to the commutation relations of the Virasoro algebra (1.5).

### 1.2.3 Conformal symmetry constraints on OPEs

For technical simplicity, we focus on an OPE of 2 primary fields. In that OPE, we single out the contributions of a primary field $V_\Delta$ and its descendants:

$$V_{\Delta_1}(z_1)V_{\Delta_2}(z_2) \supset \sum_{L \in \mathcal{L}} C_{\Delta_1, \Delta_2}^{\Delta, L}(z_1, z_2)LV_\Delta(z_2) , \tag{1.44}$$

where $\mathcal{L}$ (1.8) is the set of creation operators. On both sides of the OPE, let us insert $\frac{1}{2\pi i} \oint_{z_1, z_2} dy(y - z_2)^{n+1}T(y)$, where the integration contour encloses both $z_1$ and $z_2$. Using Eqs. (1.38) and (1.39), we find

$$\left(L_n^{(z_2)} + z_{12}^{n+1}\partial_{z_1} + (n + 1)z_{12}^n\Delta_1\right)V_{\Delta_1}(z_1)V_{\Delta_2}(z_2) \supset \sum_{L \in \mathcal{L}} C_{\Delta_1, \Delta_2}^{\Delta, L}(z_1, z_2)L_nLV_\Delta(z_2) . \tag{1.45}$$

We focus on 3 cases:

- $\boxed{n = -1}$: We use Eq. (1.33) for $V_{\Delta_2}(z_2)$ and $LV_\Delta(z_2)$, and perform the OPE again on the left-hand side:

$$(\partial_{z_1} + \partial_{z_2}) \sum_{L \in \mathcal{L}} C^{\Delta,L}_{\Delta_1,\Delta_2}(z_1, z_2) LV_\Delta(z_2) = \sum_{L \in \mathcal{L}} C^{\Delta,L}_{\Delta_1,\Delta_2}(z_1, z_2) \partial_{z_2} LV_\Delta(z_2) . \qquad (1.46)$$

This leads to $(\partial_{z_1} + \partial_{z_2}) C^{\Delta,L}_{\Delta_1,\Delta_2}(z_1, z_2) = 0$, i.e. translation invariance of the OPE coefficients.

- $\boxed{n = 0}$: We use $L_0 LV_\Delta(z_2) = (\Delta + |L|) V_\Delta(z_2)$, and perform the OPE again on the left-hand side. This leads to $(z_{12} \partial_{z_1} + \Delta_1 + \Delta_2 - \Delta - |L|) C^{\Delta,L}_{\Delta_1,\Delta_2}(z_1, z_2) = 0$, and we deduce the dependence of OPE coefficients on $z_1, z_2$:

$$C^{\Delta,L}_{\Delta_1,\Delta_2}(z_1, z_2) = z_{12}^{\Delta+|L|-\Delta_1-\Delta_2} C^{\Delta,L}_{\Delta_1,\Delta_2} . \qquad (1.47)$$

- $\boxed{n \geq 1}$: We use $L_n^{(z_2)} V_{\Delta_2}(z_2) = 0$, and perform the OPE again on the left-hand side. The action of the differential operator on the OPE coefficients is now known, and we find

$$\sum_{L \in \mathcal{L}} C^{\Delta,L}_{\Delta_1,\Delta_2} z_{12}^{|L|+n} (\Delta + |L| + n\Delta_1 - \Delta_2) LV_\Delta = \sum_{L \in \mathcal{L}} C^{\Delta,L}_{\Delta_1,\Delta_2} z_{12}^{|L|} L_n LV_\Delta . \qquad (1.48)$$

Extracting the coefficient of $z_{12}^N$ for some $N \in \mathbb{N}$, we find

$$L_n W_N = \theta_{N,n} W_{N-n} \quad \text{with} \quad \begin{cases} W_N = \sum_{|L|=N} C^{\Delta,L}_{\Delta_1,\Delta_2} LV_\Delta , \\ \theta_{N,n} = N - n + n\Delta_1 - \Delta_2 + \Delta . \end{cases} \qquad (1.49)$$

These linear equations for the OPE coefficients $C^{\Delta,L}_{\Delta_1,\Delta_2}$ are called **OPE Ward identities**. Let us look for a solution $\left( f^{\Delta,L}_{\Delta_1,\Delta_2} \right)_{L \in \mathcal{L}} = \left( f^L \right)_{L \in \mathcal{L}}$, normalized by setting $f^1 = 1$, equivalently $W_0 = V_\Delta$. Iterating Eq. (1.49), we can compute $L^* W_N$ for any creation operator $L \in \mathcal{L}_N$. On the other hand, $L^* W_N = \sum_{L' \in \mathcal{L}_N} S_{L,L'} f^{L'} V_\Delta$ where $S_{L,L'}$ is the Shapovalov form (1.29). This leads to the linear system

$$\forall L \in \mathcal{L}_N, \quad \sum_{L' \in \mathcal{L}_N} S_{L,L'} f^{L'} = g^L \quad \text{with} \quad g^{L_{-n_1} L_{-n_2} \cdots L_{-n_k}}_{\Delta,\Delta_2,\Delta_1} = \prod_{i=1}^{k} \theta_{\sum_{j=i}^{k} n_j, n_i} . \qquad (1.50)$$

Assuming the Shapovalov form is invertible at level $N$, the solution is

$$\boxed{ f^{\Delta,L}_{\Delta_1,\Delta_2} = \sum_{|L'|=|L|} S^{-1}_{L,L'}(\Delta) g^{L'}_{\Delta,\Delta_2,\Delta_1} } . \qquad (1.51)$$

For example, we have $g^{L_{-1}} = \theta_{1,1}$, $g^{L^2_{-1}} = \theta_{1,1}\theta_{2,1}$ and $g^{L_{-2}} = \theta_{2,2}$ with

$$\theta_{1,1} = \Delta + \Delta_1 - \Delta_2 \quad , \quad \theta_{2,1} = \theta_{1,1} + 1 \quad , \quad \theta_{2,2} = \Delta + 2\Delta_2 - \Delta_1 . \qquad (1.52)$$

Inverting the Shapovalov matrix (1.30), we deduce

$$f^{L_{-1}} = \frac{\Delta + \Delta_1 - \Delta_2}{2\Delta} , \qquad (1.53a)$$

$$\begin{bmatrix} f^{L^2_{-1}} \\ f^{L_{-2}} \end{bmatrix} = \frac{1}{16(\Delta - \Delta_{(2,1)})(\Delta - \Delta_{(1,2)})} \begin{bmatrix} 2 + \frac{c}{4\Delta} & -3 \\ -3 & 4\Delta + 2 \end{bmatrix} \begin{bmatrix} (\Delta+\Delta_1-\Delta_2)(\Delta+\Delta_1-\Delta_2+1) \\ \Delta+2\Delta_1-\Delta_2 \end{bmatrix} . \qquad (1.53b)$$

The solution $\left(f^L\right)_{L \in \mathcal{L}}$ of the OPE Ward identities is unique if and only if the Shapovalov matrix is invertible, i.e. if and only if there is no null vector in the Verma module $\mathcal{V}_\Delta$. A null vector $L_{\langle r,s \rangle} V_{\Delta_{(r,s)}}$ at level $rs$ manifests itself as a simple pole at $\Delta = \Delta_{(r,s)}$ of the vectors $(f^L)_{L \in \mathcal{L}_N}$ for any $N \geq rs$. To summarize, the contributions of $V_\Delta$ and its descendants in an OPE of 2 primary fields read

$$\boxed{V_{\Delta_1}(z_1)V_{\Delta_2}(z_2) \supset C^\Delta_{\Delta_1,\Delta_2} z_{12}^{\Delta-\Delta_1-\Delta_2}\left(V_\Delta(z_2) + \sum_{L \in \mathcal{L}\backslash\{1\}} z_{12}^{|L|} f^{\Delta,L}_{\Delta_1,\Delta_2} L V_\Delta(z_2)\right)}, \quad (1.54)$$

where the **OPE structure constant** $C^\Delta_{\Delta_1,\Delta_2}$ is left undetermined by conformal symmetry, while the dependence on $z_1, z_2$ and the coefficients $f^{\Delta,L}_{\Delta_1,\Delta_2}$ are universal quantities that are in principle known.

In the case of the OPE coefficients $C^{\Delta,L}_{\Delta_1,\Delta_2}(z_1, z_2)$, we have seen that the primary field determines the contributions of its descendant fields. This will also be true in correlation functions, as a result of Virasoro symmetry. However, this would not necessarily hold if we had a larger chiral symmetry algebra such as a W-algebra.

The form of the dependence on $z_1, z_2$ provides an a posteriori justification for our assumption in Section 1.1.2 that $L_0$-eigenvalues be bounded from below. We have indeed seen that an $L_0$-eigenvector $LV_\Delta$ of dimension $\Delta + |L|$ comes with a factor $z_{12}^{\Delta+|L|}$. If $L_0$-eigenvalues were not bounded from below, the sum over states could not converge in a neighbourhood of $z_1 = z_2$.

### 1.2.4 Fusion rules and fusion products

Because of the poles in OPE coefficients, there are constraints on the fields that can appear in OPEs. Let us focus on the pole at $\Delta_{(1,1)} = 0$. There are 2 types of primary fields of dimension 0: the non-degenerate field $V_0$, which generates the Verma module $\mathcal{V}_0$, and the degenerate field $V^d_{\langle 1,1 \rangle}$, which obeys the null vector equation $L_{-1}V^d_{\langle 1,1 \rangle} = \partial V^d_{\langle 1,1 \rangle} = 0$ and generates the degenerate representation $\mathcal{R}^d_{\langle 1,1 \rangle}$ (1.27).

- Can $V_0$ appear in the OPE $V_{\Delta_1}V_{\Delta_2}$? The pole of $f^{L_{-1}}$ (1.53a) means we cannot impose the normalization condition $f^1 = 1$: we in fact have $f^1 = 0$ while $f^{L_{-1}} \neq 0$. Therefore, the non-vanishing null vector $L_{-1}V_0$ may appear in the OPE, together with its descendants, while $V_0$ itself and its other descendants give vanishing contributions. This situation is better interpreted as the appearance of $V_1$, not $V_0$.

- In the case of $V^d_{\langle 1,1 \rangle}$, the null vector vanishes, and no non-vanishing descendant of $V^d_{\langle 1,1 \rangle}$ may appear — unless $g^{L_{-1}} = 0$ in Eq. (1.50), which happens if $\Delta_1 = \Delta_2$.

On the other hand, the OPE $V_0 V_{\Delta_2}$ is just a special case of the OPE (1.54). But the OPE $V^d_{\langle 1,1 \rangle} V_{\Delta_2}$ must obey

$$0 = \frac{\partial}{\partial z_1} V^d_{\langle 1,1 \rangle}(z_1)V_{\Delta_2}(z_2) \supset C^\Delta_{\langle 1,1 \rangle,\Delta_2} \frac{\partial}{\partial z_1} z_{12}^{\Delta-\Delta_2}\big(V_\Delta(z_2) + \cdots\big), \quad (1.55)$$

therefore $V_\Delta$ may appear only provided $\Delta = \Delta_2$. This agrees with the remark (in Section 1.2.2) that $V^d_{\langle 1,1 \rangle}$ is proportional to the identity field.

Let us introduce the **fusion rules** as a convenient notation for writing constraints on OPEs, by omitting everything but the primary fields:

$$V^d_{\langle 1,1 \rangle}V_\Delta = V_\Delta \quad , \quad V_{\Delta_1}V_{\Delta_2} \ni V^d_{\langle 1,1 \rangle} \implies \Delta_1 = \Delta_2 . \quad (1.56)$$

This may be reformulated in terms of the **fusion product** of representations of the Virasoro algebra:

$$\mathcal{R}^d_{\langle 1,1 \rangle} \times \mathcal{V}_\Delta = \mathcal{V}_\Delta \quad , \quad \dim \mathrm{Hom}\left(\mathcal{V}_{\Delta_1} \times \mathcal{V}_{\Delta_2}, \mathcal{R}^d_{\langle 1,1 \rangle}\right) = \delta_{\Delta_1, \Delta_2} \ , \tag{1.57}$$

where $\mathrm{Hom}(\mathcal{R}, \mathcal{R}')$ is the space of linear maps from $\mathcal{R}$ to $\mathcal{R}'$ that commute with the action of the Virasoro algebra. The fusion product can be algebraically defined [11]. It differs from the tensor product, in particular its definition involves the fields' positions, although the resulting product does not. Just like OPEs, the fusion product is commutative and associative. Moreover, as we just saw in the case of Verma modules, we have an isomorphism

$$\mathrm{Hom}\left(\mathcal{R}, \mathcal{R}'\right) \simeq \mathrm{Hom}\left(\mathcal{R} \times \mathcal{R}', \mathcal{R}^d_{\langle 1,1 \rangle}\right) \ , \tag{1.58}$$

which together with associativity implies

$$\mathrm{Hom}\left(\mathcal{R} \times \mathcal{R}', \mathcal{R}''\right) \simeq \mathrm{Hom}\left(\mathcal{R}' \times \mathcal{R}'', \mathcal{R}\right) \ . \tag{1.59}$$

Given 3 representations $\mathcal{R}, \mathcal{R}', \mathcal{R}''$, the number $\dim \mathrm{Hom}\left(\mathcal{R} \times \mathcal{R}', \mathcal{R}''\right) \in \mathbb{N}$ is called the **fusion multiplicity** of $\mathcal{R}''$ in the fusion product $\mathcal{R} \times \mathcal{R}'$. A fusion multiplicity is called nontrivial if it is $\geq 2$.

### 1.2.5 Fusion products of degenerate representations

According to the isomorphism (1.59), $V^d_{\langle 2,1 \rangle} V_{\Delta_1} \ni V_{\Delta_2} \iff V_{\Delta_1} V_{\Delta_2} \ni V^d_{\langle 2,1 \rangle} \iff$ the 2-dimensional linear system (1.50) at $N = 2$ with $\Delta = \Delta_{(2,1)}$ has a solution, even though it is of rank 1. Explicitly, this condition reads

$$\left(4\Delta_{(2,1)} + 2\right)\left(\Delta_{(2,1)} + 2\Delta_1 - \Delta_2\right) = 3\left(\Delta_{(2,1)} + \Delta_1 - \Delta_2\right)\left(\Delta_{(2,1)} + 1 + \Delta_1 - \Delta_2\right) \ . \tag{1.60}$$

Using the expression (1.16) for $\Delta_{(2,1)}$, and replacing conformal dimensions with momentums (1.17), this equation reduces to

$$\prod_{\pm, \pm} \left(\tfrac{\beta}{2} \pm P_1 \pm P_2\right) = 0 \ . \tag{1.61}$$

This leads to the fusion rules of $V^d_{\langle 2,1 \rangle}$, from which the fusion rules of $V^d_{\langle 1,2 \rangle}$ are deduced by $\beta \to \beta^{-1}$. We write these fusion rules in terms of fusion products, where we index Verma modules by their momentums:

$$\boxed{\mathcal{R}^d_{\langle 2,1 \rangle} \times \mathcal{V}_P = \sum_\pm \mathcal{V}_{P \pm \frac{\beta}{2}}} \quad , \quad \boxed{\mathcal{R}^d_{\langle 1,2 \rangle} \times \mathcal{V}_P = \sum_\pm \mathcal{V}_{P \pm \frac{1}{2\beta}}} \ . \tag{1.62}$$

From these simple formulas, all the discrete spectrums of Section 2 will follow. For the moment, let us deduce the fusion products of higher degenerate representations $\mathcal{R}^d_{\langle r,s \rangle}$, using the associativity of the fusion product. We start with the remark that a highest-weight representation $\mathcal{R}$ is degenerate if and only if its fusion product with any Verma module $\mathcal{R} \times \mathcal{V}_P$ has finitely many terms:

- If $\mathcal{R} = \mathcal{R}^d_{\langle r,s \rangle}$ is degenerate, the condition that $V^d_{\langle r,s \rangle}$ may appear in the OPE $V_{\Delta_1}(z_1) V_{\Delta_2}(z_2)$ leads to polynomial equations of order $rs$ on $\Delta_1, \Delta_2$, just as we saw in the case $(r, s) = (2, 1)$. These equations may be interpreted as conditions on residues of OPE coefficients,

$$V_{\Delta_1}(z_1) V_{\Delta_2}(z_2) \supset V^d_{\langle r,s \rangle} \quad \Longrightarrow \quad \forall L \in \mathcal{L}, \ \operatorname*{Res}_{\Delta = \Delta_{(r,s)}} f^{\Delta, L}_{\Delta_1, \Delta_2} = 0 \ . \tag{1.63}$$

(If the residues vanish for $|L| = rs$, then they vanish for all $L \in \mathcal{L}$.)

- If $\mathcal{R}$ is not degenerate, the corresponding primary field may appear in any OPE of the type $V_{\Delta_1}(z_1)V_{\Delta_2}(z_2)$, because the OPE Ward identities have a solution.

It follows that a fusion product of degenerate representations is itself a finite sum of degenerate representations. For example, according to Eq. (1.62), the product $\mathcal{R}^d_{\langle 2,1 \rangle} \times \mathcal{R}^d_{\langle 2,1 \rangle}$ is the sum of 2 representations of momentums $\{P_{(2,1)} \pm \frac{\beta}{2}\} = \{P_{(1,1)}, P_{(3,1)}\}$. These representations must be degenerate, therefore $\mathcal{R}^d_{\langle 2,1 \rangle} \times \mathcal{R}^d_{\langle 2,1 \rangle} = \mathcal{R}^d_{\langle 1,1 \rangle} + \mathcal{R}^d_{\langle 3,1 \rangle}$. By associativity of $\mathcal{R}^d_{\langle 2,1 \rangle} \times \mathcal{R}^d_{\langle 2,1 \rangle} \times \mathcal{V}_P$, we then deduce $\mathcal{R}^d_{\langle 3,1 \rangle} \times \mathcal{V}_P = \mathcal{V}_{P-\beta} + \mathcal{V}_P + \mathcal{V}_{P+\beta}$. Iterating this reasoning over $r$ or $s$, we obtain

$$\mathcal{R}^d_{\langle 2,1 \rangle} \times \mathcal{R}^d_{\langle r,s \rangle} \underset{r \geq 2}{=} \sum_{\pm} \mathcal{R}^d_{\langle r\pm 1,s \rangle} \quad , \quad \mathcal{R}^d_{\langle 1,2 \rangle} \times \mathcal{R}^d_{\langle r,s \rangle} \underset{s \geq 2}{=} \sum_{\pm} \mathcal{R}^d_{\langle r,s\pm 1 \rangle} \ . \tag{1.64}$$

This allows us to find the fusion product of any degenerate representation with a Verma module, by iteration over $r$ and $s$:

$$\boxed{\mathcal{R}^d_{\langle r,s \rangle} \times \mathcal{V}_P = \sum_{i=-\frac{r-1}{2}}^{\frac{r-1}{2}} \sum_{j=-\frac{s-1}{2}}^{\frac{s-1}{2}} \mathcal{V}_{P+i\beta+j\beta^{-1}}} \ , \tag{1.65}$$

where the sums run by increments of 1. Since this sum has $rs$ terms, the Verma module $\mathcal{V}_{\Delta_{(r,s)}}$ must have a null vector at level $rs$, a result that we stated in Section 1.1.3, and which we now derived using fusion products. Fusion can even be used to compute the null vectors themselves [1].

We can then find the fusion product $\mathcal{R}^d_{\langle r_1,s_1 \rangle} \times \mathcal{R}^d_{\langle r_2,s_2 \rangle}$, either by again iterating over $r_1$ and $s_1$, or by summing the degenerate representations whose momentums appear in both $\mathcal{R}^d_{\langle r_1,s_1 \rangle} \times \mathcal{V}_{P_{(r_2,s_2)}}$ and $\mathcal{R}^d_{\langle r_2,s_2 \rangle} \times \mathcal{V}_{P_{(r_1,s_1)}}$:

$$\boxed{\mathcal{R}^d_{\langle r_1,s_1 \rangle} \times \mathcal{R}^d_{\langle r_2,s_2 \rangle} = \sum_{r \overset{2}{=} |r_1-r_2|+1}^{r_1+r_2-1} \sum_{s \overset{2}{=} |s_1-s_2|+1}^{s_1+s_2-1} \mathcal{R}^d_{\langle r,s \rangle}} \ , \tag{1.66}$$

where the sums now run by increments of 2. The number of terms is $\min(r_1, r_2) \min(s_1, s_2)$. For example,

$$\mathcal{R}^d_{\langle 5,2 \rangle} \times \mathcal{R}^d_{\langle 3,3 \rangle} = \mathcal{R}^d_{\langle 3,2 \rangle} + \mathcal{R}^d_{\langle 5,2 \rangle} + \mathcal{R}^d_{\langle 7,2 \rangle} + \mathcal{R}^d_{\langle 3,4 \rangle} + \mathcal{R}^d_{\langle 5,4 \rangle} + \mathcal{R}^d_{\langle 7,4 \rangle} \ . \tag{1.67}$$

## 1.3 Correlation functions

Correlation functions are observables of quantum field theory: quantities that we can in principle compute, and compare with experimental data. We will focus on $N$-point functions on the Riemann sphere: functions of $N$ distinct positions $z_1, z_2, \dots, z_N \in \overline{\mathbb{C}}$, which are written in terms of $N$ fields as

$$\left\langle V_1(z_1)V_2(z_2)\cdots V_N(z_N) \right\rangle \ . \tag{1.68}$$

Correlation functions are assumed to depend linearly on fields: in particular, this implies $\partial_{z_1} \left\langle V_1(z_1)\cdots \right\rangle = \left\langle \partial_{z_1} V_1(z_1)\cdots \right\rangle$. All the assumptions and results of Section 1.2 may be understood as constraints on correlation functions. In 2d CFT, there are other observables that we will not consider, such as:

- Correlation functions on higher-genus Riemann surfaces, starting with the torus.

- Correlation functions on Riemann surfaces with boundaries, starting with the disc.

- Correlation functions in the presence of defect lines.

- Entanglement entropy.

### 1.3.1 Conformal symmetry constraints

In order to derive the conformal symmetry constraints for an $N$-point function of primary fields $Z$, it is convenient to consider the $N + 1$-point function $Z(y)$ that also includes the energy-momentum tensor $T(y)$:

$$
Z = \left\langle \prod_{i=1}^{N} V_{\Delta_i}(z_i) \right\rangle \quad , \quad Z(y) = \left\langle T(y) \prod_{i=1}^{N} V_{\Delta_i}(z_i) \right\rangle . \tag{1.69}
$$

The function $Z(y)$ is meromorphic, with $N$ poles at $y = z_i$, where its behaviour is constrained by the OPE $T(y)V_{\Delta_i}(z_i)$ (1.39). To fully determine $Z(y)$, it remains to determine its behaviour at $y = \infty$. Because our fields live on the Riemann sphere, they must be smooth at $y = \infty$: in the case of $T(y)$, we assume that this means

$$
T(y) \underset{y\to\infty}{=} O\left(\frac{1}{y^4}\right) . \tag{1.70}
$$

This implies first of all that $\lim_{y\to\infty} Z(y) = 0$, therefore $Z(y)$ is completely determined by its residues at $y = z_i$:

$$
Z(y) = \sum_{i=1}^{N} \left( \frac{\Delta_i}{(y - z_i)^2} + \frac{1}{y - z_i}\frac{\partial}{\partial z_i} \right) Z . \tag{1.71}
$$

Furthermore, Eq. (1.70) implies the vanishing of the coefficients of $y^{-1}, y^{-2}, y^{-3}$ in the Laurent expansion of $Z(y)$ near $y = \infty$, leading to the 3 **global Ward identities**

$$
\sum_{i=1}^{N} \partial_{z_i} Z = \sum_{i=1}^{N} \left( z_i \partial_{z_i} + \Delta_i \right) Z = \sum_{i=1}^{N} \left( z_i^2 \partial_{z_i} + 2\Delta_i z_i \right) Z = 0 . \tag{1.72}
$$

Equivalently, the behaviour of $Z$ under global conformal transformations (1.4) must be

$$
\boxed{ \left\langle \prod_{i=1}^{N} V_{\Delta_i} \left( \frac{az_i + b}{cz_i + d} \right) \right\rangle = \prod_{i=1}^{N}(cz_i + d)^{2\Delta_i} \left\langle \prod_{i=1}^{N} V_{\Delta_i}(z_i) \right\rangle . } \tag{1.73}
$$

The global Ward identities are valid not only for primary fields, but also for quasi-primary fields: $L_0$-eigenvectors that are annihilated by $L_1$ but not necessarily by $L_{n\geq 2}$. In particular, applying Eq. (1.73) to the transformation $z \mapsto \frac{1}{z}$, we find $V_\Delta(z) \underset{z\to\infty}{=} O(z^{-2\Delta})$, which allows us to define

$$
V_\Delta(\infty) = \lim_{z\to\infty} z^{2\Delta} V_\Delta(z) . \tag{1.74}
$$

Furthermore, according to the OPE $T(y)T(z)$ (1.43), the energy-momentum tensor $T$ is a quasi-primary field of dimension $\Delta = 2$, whose assumed behaviour at $\infty$ (1.70) is consistent with Eq. (1.74). Using Eq. (1.73) for a correlation function that involves $T$, we deduce a definition of descendant fields at $\infty$,

$$
L_n V_\Delta(\infty) = \frac{1}{2\pi i} \oint_\infty dy \, y^{1-n} T(y) V_\Delta(\infty) . \tag{1.75}
$$

(Compare with $L_n V_\Delta(z)$ (1.38).) We could alternatively use the transformation $z \mapsto \frac{1}{z-a}$ for any $a \in \mathbb{C}$, and obtain an $a$-dependent definition $\frac{1}{2\pi i} \oint_\infty dy \, (y - a)^{1-n} T(y) V_\Delta(\infty)$. We interpret $\frac{1}{z-a}$ as a **local coordinate** near $z = \infty$. Correlation functions on general

Riemann surfaces depend on choices of local coordinates, see [12, Section 3]. In the case of the Riemann sphere, there is a canonical local coordinate near any point $z \in \mathbb{C}$, but not near $z = \infty$. So we chose the local coordinate $\frac{1}{z}$ near $z = \infty$.

Let us solve the global Ward identities in the cases $N = 1, 2, 3, 4$. We have 3 linear differential equations in the $N$ variables $z_i$, and we write their solutions up to $z_i$-independent prefactors:

- $\boxed{N = 1}$ : The identities amount to $\partial_{z_1} Z = \Delta_1 Z = 0$, and their solution is

$$\left\langle V_{\Delta_1}(z_1) \right\rangle \propto \delta_{\Delta_1, 0} . \tag{1.76a}$$

- $\boxed{N = 2}$ : Again, there are more Ward identities than variables, leading to $Z \neq 0 \implies \Delta_1 = \Delta_2$, and to the solution

$$\left\langle V_{\Delta_1}(z_1) V_{\Delta_2}(z_2) \right\rangle \propto \delta_{\Delta_1, \Delta_2} z_{12}^{-2\Delta_1} . \tag{1.76b}$$

- $\boxed{N = 3}$ : With as many Ward identities as variables, we have a unique solution with no constraints on the conformal dimensions:

$$\left\langle \prod_{i=1}^{3} V_{\Delta_i}(z_i) \right\rangle \propto z_{12}^{\Delta_3 - \Delta_1 - \Delta_2} z_{13}^{\Delta_2 - \Delta_1 - \Delta_3} z_{23}^{\Delta_1 - \Delta_2 - \Delta_3} . \tag{1.76c}$$

- $\boxed{N = 4}$ : The solution involves an arbitrary function $G(z)$ of the **cross-ratio** $z = \frac{z_{12} z_{34}}{z_{13} z_{24}}$:

$$\left\langle \prod_{i=1}^{4} V_{\Delta_i}(z_i) \right\rangle \propto z_{13}^{-2\Delta_1} z_{23}^{\Delta_1 - \Delta_2 - \Delta_3 + \Delta_4} z_{24}^{-\Delta_1 - \Delta_2 + \Delta_3 - \Delta_4} z_{34}^{\Delta_1 + \Delta_2 - \Delta_3 - \Delta_4} G\left( \frac{z_{12} z_{34}}{z_{13} z_{24}} \right) . \tag{1.76d}$$

A 4-point function is therefore completely determined by its dependence on 1 of its 4 variables. In particular, it can be convenient to set 3 positions to the fixed values $(z_2, z_3, z_4) = (0, \infty, 1)$, in which case

$$\left\langle V_{\Delta_1}(z) V_{\Delta_2}(0) V_{\Delta_3}(\infty) V_{\Delta_4}(1) \right\rangle \propto G(z) . \tag{1.77}$$

In addition to global Ward identities, there are **local Ward identities** , which follow from $\oint_\infty dy\, \epsilon(y) Z(y) = 0$, where $\epsilon(y)$ is a meromorphic function such that $\epsilon(y) \underset{y \to \infty}{=} O(y^2)$, with poles at $y = z_i$. Local Ward identities allow us to determine correlation functions of descendant fields from the corresponding correlation functions of primary fields. In the case $\epsilon(y) = \frac{1}{(y - z_1)^{n-1}}$ with $n \geq -1$, using the expressions (1.71) for $Z(y)$ and (1.38) for $L_{-n} V_{\Delta_1}(z_1)$, we obtain the Ward identity

$$\left\langle L_{-n} V_{\Delta_1}(z_1) \prod_{j=2}^{N} V_{\Delta_j}(z_j) \right\rangle = \sum_{j=2}^{N} \left( -\frac{1}{z_{j1}^{n-1}} \frac{\partial}{\partial z_j} + \frac{n-1}{z_{j1}^{n}} \Delta_j \right) \left\langle \prod_{j=1}^{N} V_{\Delta_j}(z_j) \right\rangle . \tag{1.78}$$

If $n \geq 2$ this is a local Ward identity, while for $n = -1, 0, 1$ we recover global Ward identities. In particular, for any creation operator $L \in \mathcal{L}$ we can compute the ratio

$$g_{\Delta_1, \Delta_2, \Delta_3}^{L} = \frac{\langle L V_{\Delta_1}(0) V_{\Delta_2}(\infty) V_{\Delta_3}(1) \rangle}{\langle V_{\Delta_1}(0) V_{\Delta_2}(\infty) V_{\Delta_3}(1) \rangle} . \tag{1.79}$$

Let us show that this coincides with the quantity that appeared in OPE Ward identities (1.50). In the 3-point function $\langle L'V_\Delta(0)V_{\Delta_2}(\infty)V_{\Delta_1}(1)\rangle \propto g^{L'}_{\Delta,\Delta_2,\Delta_1}$, we write the OPE $V_{\Delta_1}(1)V_{\Delta_2}(\infty)$ as a linear combination of descendant fields $LV_\Delta(\infty)$. Since our local coordinate near $\infty$ is $y = \frac{1}{z}$, we obtain a series in powers of $y_1 - y_2 = 1$, which may seem large until we remember that the field $L'V_\Delta(0)$ is infinitely far away at $y = \infty$. Assuming for simplicity $\langle V_\Delta(0)V_\Delta(\infty)\rangle = 1$, we obtain

$$g^{L'}_{\Delta,\Delta_2,\Delta_1} = \sum_{L\in\mathcal{L}} f^{\Delta,L}_{\Delta_1,\Delta_2}\left\langle L'V_\Delta(0)LV_\Delta(\infty)\right\rangle . \tag{1.80}$$

By the definition (1.75) of descendant fields at $\infty$, the 2-point function coincides with the Shapovalov form,

$$\boxed{S_{L,L'}(\Delta) = \left\langle L'V_\Delta(0)LV_\Delta(\infty)\right\rangle = \left\langle L^*L'V_\Delta(0)V_\Delta(\infty)\right\rangle} , \tag{1.81}$$

which vanishes if $|L| > |L'|$ because $V_\Delta$ is primary, and also if $|L| < |L'|$ because $LV_\Delta$ is at $\infty$. So we recover the expression (1.50) of $g^L$ in terms of OPE coefficients and the Shapovalov form.

### 1.3.2 Single-valuedness and chiral factorization

We assume that correlation functions are **single-valued** on the Riemann sphere. Equivalently, we assume that any 2 fields are **mutually local**, i.e. that they have trivial monodromies around each other: $V_1(ze^{2\pi i})V_2(0) = V_1(z)V_2(0)$.

The correlation functions of Section 1.3.1 are of the form $z^\Delta$, which is multivalued unless $\Delta \in \mathbb{Z}$. Single-valued functions like $|z|^{2\Delta} = z^\Delta\bar{z}^\Delta$ are not written in terms of powers of $z$, but also involve $\bar{z}$. In the literature, the notation $f(z,\bar{z})$ is sometimes used for arbitrary functions of $z$, with $f(z)$ reserved to locally holomorphic functions, i.e. functions such that $\frac{\partial}{\partial\bar{z}}f(z) = 0$. We will not use the notation $f(z,\bar{z})$, which is redundant since $\bar{z}$ is itself a function of $z$.

The generators $\ell_n$ (1.2) of local conformal transformations are only valid when acting on locally holomorphic functions: for more general functions, the generators are

$$\ell_n + \bar{\ell}_n \quad , \quad i(\ell_n - \bar{\ell}_n) . \tag{1.82}$$

In fact we want the algebra of conformal transformations to act on a complex vector space: the space of complex-valued functions on $\overline{\mathbb{C}}$ for the Witt algebra, the space of states of the CFT for the Virasoro algebra. We should therefore complexify our symmetry algebra, and we obtain 2 copies of the Virasoro algebra: a **left-moving** Virasoro algebra $\mathfrak{V}$ with generators $L_n$, and a **right-moving** Virasoro algebra $\bar{\mathfrak{V}}$ with generators $\bar{L}_n$. In the case of the algebra of global conformal transformations, we start with the 6-dimensional real Lie algebra $\mathfrak{sl}_2(\mathbb{C})$, with generators (1.82) with $n = -1, 0, 1$, and the complexification yields $\mathfrak{sl}_2(\mathbb{C})^{\mathbb{C}} = \mathfrak{sl}_2(\mathbb{C}) \times \mathfrak{sl}_2(\mathbb{C})$.

In Sections 1.1 and 1.2, we have been doing **chiral CFT**, which means considering $\mathfrak{V}$ only. In a full CFT, the symmetry algebra is the **conformal algebra** $\mathfrak{V} \times \bar{\mathfrak{V}}$, where for simplicity we assume that the left and right central charges are equal. When it comes to representations, Verma modules of $\mathfrak{V} \times \bar{\mathfrak{V}}$ are simply $\mathcal{V}_\Delta \otimes \bar{\mathcal{V}}_{\bar\Delta}$, where we write a bar over representations of $\bar{\mathfrak{V}}$. There also exist representations of $\mathfrak{V} \times \bar{\mathfrak{V}}$ that are not factorized, in particular logarithmic representations [13].

Fields now belong to representations of $\mathfrak{V} \times \bar{\mathfrak{V}}$, and they obey $\frac{\partial}{\partial\bar{z}}V(z) = \bar{L}_{-1}V(z)$, which is the antiholomorphic version of Eq. (1.33). This allows us to define a right-moving energy-momentum tensor $\bar{T}(y) = \sum_{n\in\mathbb{Z}} \frac{\bar{L}_n^{(z)}}{(\bar{y}-\bar{z})^{n+2}}$. While $T$ is locally holomorphic,

$\bar{T}$ is locally antiholomorphic:

$$\partial_{\bar{z}} T(z) = 0 \quad , \quad \partial_z \bar{T}(z) = 0 \ . \tag{1.83}$$

Let $V_{\Delta,\bar{\Delta}}(z)$ be a field that is primary with respect to the conformal algebra, with the left dimension $\Delta$ and the right dimension $\bar{\Delta}$. The difference of these dimensions is called the **conformal spin**

$$S = \Delta - \bar{\Delta} \ . \tag{1.84}$$

According to Eq. (1.73), under a rotation $z \mapsto e^{i\theta} z$, corresponding to a global conformal transformation $\left( \begin{smallmatrix} a & b \\ c & d \end{smallmatrix} \right) = \left( \begin{smallmatrix} e^{\frac{i}{2}\theta} & 0 \\ 0 & e^{-\frac{i}{2}\theta} \end{smallmatrix} \right)$, a primary field transforms as

$$V_{\Delta,\bar{\Delta}}(z) \to e^{iS\theta} V_{\Delta,\bar{\Delta}} \left( e^{i\theta} z \right) \ . \tag{1.85}$$

In order to write correlation functions, let us define the **modulus squared notation** for functions of the central charge, conformal dimensions, and positions,

$$|f(c, \Delta, z)|^2 = f(c, \Delta, z) f(c, \bar{\Delta}, \bar{z}) \ , \tag{1.86}$$

where $z, \bar{z}$ are complex conjugates, while $\Delta, \bar{\Delta}$ are generally not. For simplicity we write $V_i$ for a primary field of dimensions $\Delta_i, \bar{\Delta}_i$. The solution of the left- and right-moving global Ward identities for 2-point functions is the modulus squared of Eq. (1.76b),

$$\boxed{\left\langle V_1(z_1) V_2(z_2) \right\rangle = B_{12} \left| \delta_{\Delta_1, \Delta_2} z_{12}^{-2\Delta_1} \right|^2} \ , \tag{1.87}$$

where the constant, $z_i$-independent coefficient $B_{12}$ is now called the **2-point structure constant**. Similarly, 3-point functions are given by the modulus squared of Eq. (1.76c),

$$\boxed{\left\langle V_1(z_1) V_2(z_2) V_3(z_3) \right\rangle = C_{123} \left| z_{12}^{\Delta_3 - \Delta_1 - \Delta_2} z_{13}^{\Delta_2 - \Delta_1 - \Delta_3} z_{23}^{\Delta_1 - \Delta_2 - \Delta_3} \right|^2} \ , \tag{1.88}$$

where $C_{123}$ is the **3-point structure constant**. Single-valuedness of the 2-point and 3-point functions implies that spins obey

$$S_i \in \frac{1}{2}\mathbb{Z} \quad , \quad S_1 + S_2 + S_3 \in \mathbb{Z} \ . \tag{1.89}$$

Now we have assumed that fields commute (1.41), but the $z_i$-dependent factors of the 2-point and 3-point functions are not invariant under permutations. This implies that structure constants behave as

$$B_{21} = (-)^{2S_1} B_{12} \quad , \quad C_{\sigma(1)\sigma(2)\sigma(3)} = \text{sign}(\sigma)^{S_1+S_2+S_3} C_{123} \ , \tag{1.90}$$

where $\sigma$ is a permutation of $\{1, 2, 3\}$. In particular, for the 2-point function of a field with itself, we have $B_{12} = B_{21}$, which must vanish if the spin is half-integer $S_1 \in \frac{1}{2} + \mathbb{Z}$. Fields with half-integer spins should rather anti-commute with themselves, i.e. they should be fermionic. Unless stated otherwise, we will assume that spins are integer:

$$\boxed{S \in \mathbb{Z}} \ . \tag{1.91}$$

Finally, notice that $\langle V_1 V_2 \rangle = \langle I V_1 V_2 \rangle$, where $I$ is the identity field. Assuming for simplicity

$$B_{ij} = \delta_{ij} B_i \ , \tag{1.92}$$

this implies

$$\boxed{B_i = C_{Iii}} \ . \tag{1.93}$$

### 1.3.3 Crossing symmetry and conformal blocks

Let us systematically use OPEs for simplifying correlation functions. Knowing the contribution (1.54) of left-moving descendants of primary fields, the complete OPE of 2 primary fields reads

$$V_1(z_1)V_2(z_2) = \sum_{k\in\mathcal{S}^{12}} C_{12}^k \left| z_{12}^{\Delta_k-\Delta_1-\Delta_2} \sum_{L\in\mathcal{L}} z_{12}^{|L|} f_{\Delta_1,\Delta_2}^{\Delta_k,L} \right|^2 L\bar{L}V_k(z_2) , \qquad (1.94)$$

where the basis $\mathcal{L}$ of creation operators was defined in Eq. (1.8), we assume that all fields are primaries or descendants thereof, and the **OPE spectrum** $\mathcal{S}^{12}$ is the set of primary fields that appear in our OPE. Let us insert this OPE in a 3-point function, and focus on the leading terms as $z_1 \to z_2$. Using the 2-point function (1.87), we obtain

$$\left\langle V_1(z_1)V_2(z_2)V_3(z_3) \right\rangle \underset{z_1\to z_2}{\sim} \left| z_{12}^{\Delta_3-\Delta_1-\Delta_2} z_{23}^{-2\Delta_3} \right|^2 \sum_{k\in\mathcal{S}} C_{12}^k B_{k3} . \qquad (1.95)$$

Comparing with the 3-point function (1.88), this implies a relation between the 2-point, 3-point and OPE structure constants,

$$\boxed{C_{123} = C_{12}^3 B_3} . \qquad (1.96)$$

If we now insert the OPE in the 4-point function $\langle V_1(z)V_2(0)V_3(\infty)V_4(1)\rangle$, we obtain a linear combination of 3-point functions of the type

$$\left\langle L\bar{L}V_k(0)V_3(\infty)V_4(1) \right\rangle = \left| g_{\Delta_k,\Delta_3,\Delta_4}^L \right|^2 \left\langle V_k(0)V_3(\infty)V_4(1) \right\rangle , \qquad (1.97)$$

where the action of $L, \bar{L} \in \mathcal{L}$ gives rise to the factors $g^L$ (1.79), which are determined by conformal symmetry. This allows us to write our 4-point function as

$$\boxed{\left\langle V_1(z)V_2(0)V_3(\infty)V_4(1) \right\rangle = \sum_{k\in\mathcal{S}^{(s)}} C_{12}^k C_{k34} \left| \mathcal{F}_{\Delta_k}^{(s)}(c|\Delta_1,\Delta_2,\Delta_3,\Delta_4|z) \right|^2} , \qquad (1.98)$$

where we introduce the $s$-**channel spectrum** $\mathcal{S}^{(s)} = \mathcal{S}^{12} \cap \mathcal{S}^{34}$, and the $s$-channel **Virasoro blocks**, which are known (in principle) functions of $z$ as well as the central charge and conformal dimensions, of the type

$$\mathcal{F}_{\Delta}^{(s)}(c|\Delta_1,\Delta_2,\Delta_3,\Delta_4|z) = z^{\Delta-\Delta_1-\Delta_2} \sum_{N=0}^{\infty} f_N(\Delta)z^N . \qquad (1.99)$$

The OPE that we have taken leads to coefficients $f_N(\Delta)$ that are combinations of $f^L$ and $g^L$. Using the relation (1.51), these coefficients can alternatively be rewritten in terms of $g^L$ alone, at the price of introducing the inverse Shapovalov form:

$$f_N(\Delta) = \sum_{\substack{L\in\mathcal{L}\\|L|=N}} f_{\Delta_1,\Delta_2}^{\Delta,L} g_{\Delta,\Delta_3,\Delta_4}^L = \sum_{\substack{L,L'\in\mathcal{L}\\|L|=|L'|=N}} g_{\Delta,\Delta_2,\Delta_1}^{L'} S_{L,L'}^{-1}(\Delta) g_{\Delta,\Delta_3,\Delta_4}^L , \qquad (1.100)$$

The first few coefficients may be assembled from Eqs. (1.53), (1.50):

$$f_0(\Delta) = 1 \quad , \quad f_1(\Delta) = \frac{(\Delta+\Delta_1-\Delta_2)(\Delta+\Delta_4-\Delta_3)}{2\Delta} , \qquad (1.101a)$$

$$f_2(\Delta) = \frac{\left[ \begin{smallmatrix} (\Delta+\Delta_4-\Delta_3)(\Delta+\Delta_4-\Delta_3+1) \\ \Delta+2\Delta_4-\Delta_3 \end{smallmatrix} \right]^T \left[ \begin{smallmatrix} 2+\frac{c}{4\Delta} & -3 \\ -3 & 4\Delta+2 \end{smallmatrix} \right] \left[ \begin{smallmatrix} (\Delta+\Delta_1-\Delta_2)(\Delta+\Delta_1-\Delta_2+1) \\ \Delta+2\Delta_1-\Delta_2 \end{smallmatrix} \right]}{16(\Delta-\Delta_{(2,1)})(\Delta-\Delta_{(1,2)})} . \qquad (1.101b)$$

The $s$-channel decomposition (1.98) reduces the 4-point function to a combination of model-dependent structure constants, and universal Virasoro blocks. Repeatedly using OPEs, any $N$-point function can likewise be decomposed into $N$-point Virasoro blocks. However, these decompositions are not unique. In the case $N = 4$, there are 3 possible decompositions, called the **s-channel**, **t-channel** and **u-channel** decompositions, depending on which OPE we perform. For each channel, let us write the relevant OPE, the **4-point structure constants** that appear as coefficients, the asymptotics of the Virasoro blocks, and a diagram that represents the decomposition:

| Channel | $s$ | $t$ | $u$ |
|---|---|---|---|
| OPE | $V_1(z)V_2(0)$ | $V_1(z)V_4(1)$ | $V_1(z)V_3(\infty)$ |
| 4-point | $D_k^{(s)} = C_{12}^k C_{k34}$ | $D_k^{(t)} = C_{41}^k C_{k23}$ | $D_k^{(u)} = C_{31}^k C_{k42}$ |
| Asymptotics | $\mathcal{F}_\Delta^{(s)}(z) \underset{z\to 0}{\sim} z^{\Delta-\Delta_1-\Delta_2}$ | $\mathcal{F}_\Delta^{(t)}(z) \underset{z\to 1}{\sim} (1-z)^{\Delta-\Delta_1-\Delta_4}$ | $\mathcal{F}_\Delta^{(u)}(z) \underset{z\to\infty}{\sim} \left(\frac{1}{z}\right)^{\Delta+\Delta_1-\Delta_3}$ |
| Diagram |  |  |  |

$$(1.102)$$

The commutativity of the product of fields implies that the 4-point function (1.76d) is invariant under permutations of the 4 fields, and this determines how Virasoro blocks behave under permutations. In particular, $t$-channel and $u$-channel blocks can be deduced from $s$-channel blocks via the relations

$$\mathcal{F}_\Delta^{(t)}(z) = \mathcal{F}_\Delta^{(s)}(\Delta_1, \Delta_4, \Delta_3, \Delta_2 | 1 - z) \quad , \quad \mathcal{F}_\Delta^{(u)}(z) = z^{-2\Delta_1} \mathcal{F}_\Delta^{(s)}(\Delta_1, \Delta_3, \Delta_2, \Delta_4 | \tfrac{1}{z}) \ .$$
$$(1.103)$$

The decomposition (1.98) of a 4-point function involves $s$-channel **conformal blocks** of the form

$$\mathcal{G}_{\Delta,\bar{\Delta}}^{(x)}(z) = \left| \mathcal{F}_\Delta^{(x)}(z) \right|^2 \ , \qquad (x = s, t, u) \ . \tag{1.104}$$

These conformal blocks are chirally factorized into Virasoro blocks because of our assumption that all fields are primaries or descendants. However, representations of the conformal algebra that are not factorized give rise to conformal blocks that are also not factorized: this is the case with the logarithmic blocks of Section 5.3.2. Let us now write the **crossing symmetry equations** as the equality between the 3 decompositions of a 4-point function into structure constants and conformal blocks:

$$\boxed{\forall z \in \overline{\mathbb{C}} \backslash \{0, 1, \infty\} \ , \quad \sum_{k\in\mathcal{S}^{(s)}} D_k^{(s)} \mathcal{G}_{\Delta_k,\bar{\Delta}_k}^{(s)}(z) = \sum_{k\in\mathcal{S}^{(t)}} D_k^{(t)} \mathcal{G}_{\Delta_k,\bar{\Delta}_k}^{(t)}(z) = \sum_{k\in\mathcal{S}^{(u)}} D_k^{(u)} \mathcal{G}_{\Delta_k,\bar{\Delta}_k}^{(u)}(z)} \ .$$
$$(1.105)$$

Given 4 fields $V_1, V_2, V_3, V_4$, this is a system of infinitely many equations (parametrized by the cross-ratio $z$) for the spectrum $\mathcal{S}$ and the 4-point structure constants $D_k^{(x)}$. Alternatively, we can take the unknowns to be the **CFT data**: the spectrum and the 2-point and 3-point structure constants. CFT data determine OPEs, and crossing symmetry of all 4-point functions is equivalent to the associativity of OPEs, and therefore to the consistency of the CFT on the sphere.

# 2 Sketching exactly solvable CFTs

In this section we derive the spectrums and fusion rules of exactly solvable CFTs. We begin with a general discussion of CFTs, their spectrums, and the consequences of having degenerate fields. Historically, the CFTs we will discuss have been discovered using a variety of techniques:

- the modular bootstrap for minimal models [1],

- the Lagrangian approach for Liouville theory with $c \geq 25$ [14],

- lattice constructions for the $O(n)$ and Potts models [15],

- the numerical bootstrap for Liouville theory with $c \leq 1$ [16].

Here we recover their spectrums from assumptions about Virasoro representations, their fusion rules, and single-valuedness of correlation functions. Although our assumptions are simple, they are not unique or inevitable, and it can be interesting to relax or modify them. This approach can also be used in CFTs with extended symmetry algebras, see for example [17].

## 2.1 Spectrum and degenerate fields

### 2.1.1 What is a CFT? What is a spectrum?

We define a **conformal field theory** as a set of correlation functions that obey the assumptions of Section 1, most importantly Virasoro symmetry, the commutativity of fields, the existence of OPEs, and single-valuedness. We already know that on the sphere, a conformal field theory is characterized by a spectrum and 2-point and 3-point structure constants such that crossing symmetry is obeyed.

This definition works very well in **rational CFTs**, i.e. CFTs whose spectrums are made of finitely many primary fields and their descendants. However, in more general CFTs, the notion of the spectrum becomes fuzzy. To say why, let us compare various spectrums associated to various objects. By an abuse of terminology, we call spectrum a basis of primary fields:

| Object | Spectrum | Definition or constraint |
|--------|----------|--------------------------|
| $V_1 V_2$ | OPE spectrum $\mathcal{S}^{12}$ | $V_1 V_2 \sim \sum_{k \in \mathcal{S}^{12}} V_k$ |
| $\langle V_1 V_2 V_3 V_4 \rangle$ | Channel spectrum $\mathcal{S}^{(s)}$ | $\mathcal{S}^{(s)} = \mathcal{S}^{12} \cap \mathcal{S}^{34}$ |
| CFT | Physical spectrum $\mathcal{S}$ | $i, j \in \mathcal{S} \implies \mathcal{S}^{ij} \subset \mathcal{S}$ |
| $\widehat{\text{CFT}}$ | Extended spectrum $\widehat{\mathcal{S}}$ | $\mathcal{S} \subset \widehat{\mathcal{S}}, \quad i, j \in \widehat{\mathcal{S}} \implies \mathcal{S}^{ij} \subset \widehat{\mathcal{S}}$ |

(2.1)

The **physical spectrum** is made of the fields whose correlation functions we are interested in. Our constraint on the physical spectrum is called **closure under OPE**. This constraint means that if a field appears in an OPE (or a fortiori in a channel spectrum), then its correlation functions exist. We also introduce **extended spectrums** as larger spectrums that are also closed under OPE. There are various reasons for considering extended spectrums:

- In Section 1.3.1, we have used the energy-momentum tensor $T$ for deriving Ward identities. However, in Liouville theory, $T$ does not belong to the physical spectrum. Nevertheless, it always makes sense to insert $T$ in correlation functions, because it is a descendant of the identity field.

- In the $O(n)$ model and in Liouville theory, we will use the degenerate field $V^d_{\langle 1,2 \rangle}$ for constraining correlation functions, even though it does not belong to the physical spectrum. Correlation functions of the $O(n)$ model that involve $V^d_{\langle 1,2 \rangle}$ are not single-valued: they belong to an extended CFT that obeys weaker assumptions.

- The spectrum of Liouville theory is continuous, so that the OPE gives rise to an integral over the real momentums of primary fields. Since the integrand is meromorphic, the integration contour can be deformed in the complex plane. This leads to an extended spectrum of complex momentums.

We will say that **a field exists** if it belongs to some extended spectrum, i.e. if it can be consistently added to our CFT.

### 2.1.2  Diagonal and non-diagonal fields

Motivated by the single-valuedness of correlation functions, we have made the assumption (1.91) that conformal spins are integer. In extended spectrums, we may relax this assumption to the half-integer spin condition (1.89):

$$i \in \mathcal{S} \implies S_i \in \mathbb{Z} \quad , \quad i \in \widehat{\mathcal{S}} \implies S_i \in \frac{1}{2}\mathbb{Z} . \tag{2.2}$$

A simple way to satisfy these assumptions is to have fields of spin zero, i.e. **spinless fields**. However, whenever degenerate fields exist, we will use the slightly different notion of a **diagonal field**: a spinless field whose OPEs with degenerate fields only produces spinless fields. Conversely, a **non-diagonal field** is a field that either has nonzero spin, or is related to fields with nonzero spins by taking OPEs with degenerate fields. A **diagonal CFT** is a CFT whose spectrum only involves diagonal fields.

Let us summarize the primary fields that we will consider, together with their momentums, and the representations of the conformal algebra that they generate.

| Name | Notation | Conditions | $(P, \bar{P})$ | Representation |
|---|---|---|---|---|
| Degenerate | $V^d_{\langle r,s \rangle}$ | $r, s \in \mathbb{N}^*$ | $\left( P_{(r,s)}, P_{(r,s)} \right)$ | $\mathcal{R}^d_{\langle r,s \rangle} \otimes \bar{\mathcal{R}}^d_{\langle r,s \rangle}$ |
| Fully degenerate | $V^f_{\langle r,s \rangle}$ | $r, s \in \mathbb{N}^*$ $\beta^2 \in \mathbb{Q}$ | $\left( P_{(r,s)}, P_{(r,s)} \right)$ | $\mathcal{R}^f_{\langle r,s \rangle} \otimes \bar{\mathcal{R}}^f_{\langle r,s \rangle}$ |
| Diagonal | $V_P$ | $P \in \mathbb{C}$ | $(P, P)$ | $\mathcal{V}_P \otimes \bar{\mathcal{V}}_P$ |
| Non-diagonal | $V_{(r,s)}$ | $r, s \in \mathbb{C}$ $rs \in \frac{1}{2}\mathbb{Z}$ | $\left( P_{(r,s)}, P_{(r,-s)} \right)$ | $\mathcal{V}_{P_{(r,s)}} \otimes \bar{\mathcal{V}}_{P_{(r,-s)}}$ (in general) |

$$\tag{2.3}$$

We parametrize non-diagonal primary fields $V_{(r,s)}$ using the Kac indices defined by Eq. (1.18), instead of the left and right dimensions. According to Eq. (1.19), the conformal spin of $V_{(r,s)}$ is

$$\boxed{S_{(r,s)} = -rs} . \tag{2.4}$$

Then $V_{(r,s)}$ generates the Verma module $\mathcal{V}_{P_{(r,s)}} \otimes \bar{\mathcal{V}}_{P_{(r,-s)}}$, unless that module has null vectors, which occurs if $r, s \in \mathbb{Z}^*$, and also for other values of $r, s$ if $\beta^2 \in \mathbb{Q}$. If there are null vectors, $V_{(r,s)}$ may generate a degenerate representation, or belong to a logarithmic representation.

Our labelling of fields is redundant, because the reflection $P \to -P$ leaved the conformal dimension (1.17) invariant. We therefore assume the **reflection relations**

$$V_P = V_{-P} \quad , \quad V_{(r,s)} = V_{(-r,-s)} \ . \tag{2.5}$$

In some cases we will allow a nontrivial **reflection coefficient** $R_P$ such that $V_P = R_P V_{-P}$.

### 2.1.3 Constraints from degenerate fields

We will now study how the existence of diagonal degenerate fields constrains the spectrum. To begin with, let us discuss which sets of degenerate fields can occur. We assume that the OPE of 2 degenerate fields yields all the degenerate fields that are allowed by the fusion rules (1.66). If for example we repeatedly fuse $V_{\langle 5,2 \rangle}^d$ with itself, we obtain all $V_{\langle r,s \rangle}^d$ with $(r,s) \in (2\mathbb{N}+1) \times \mathbb{N}^*$. For each one of the 2 Kac indices $r, s$, allowing 1 even value implies allowing all values in $\mathbb{N}^*$, and allowing 1 odd value except 1 implies allowing all odd values. Modulo $r \leftrightarrow s$, this leads to 7 possible sets of degenerate fields, which we characterize by whether or not they contain $V_{\langle 1,1 \rangle}^d, V_{\langle 2,1 \rangle}^d, V_{\langle 3,1 \rangle}^d, V_{\langle 1,2 \rangle}^d$ and $V_{\langle 1,3 \rangle}^d$:

$$\emptyset \quad \{V_{\langle 1,1 \rangle}^d\} \quad \{V_{\langle 1,3 \rangle}^d\} \quad \{V_{\langle 1,2 \rangle}^d\} \quad \{V_{\langle 3,1 \rangle}^d, V_{\langle 1,3 \rangle}^d\} \quad \{V_{\langle 3,1 \rangle}^d, V_{\langle 1,2 \rangle}^d\} \quad \{V_{\langle 2,1 \rangle}^d, V_{\langle 1,2 \rangle}^d\} \tag{2.6}$$

In particular, the set that is generated by $\{V_{\langle 2,1 \rangle}^d, V_{\langle 1,2 \rangle}^d\}$ is the full set of degenerate fields $\{V_{\langle r,s \rangle}^d\}_{r,s \in \mathbb{N}^*}$. In order to solve CFTs, the existence of degenerate fields matters much more than their presence in the spectrum. We will focus on 2 types of CFTs:

- **CFTs with 2 degenerate fields**, in the sense that both $V_{\langle 2,1 \rangle}^d$ and $V_{\langle 1,2 \rangle}^d$ exist.

- **CFTs with 1 degenerate field** $V_{\langle 1,2 \rangle}^d$. Because the invariance under $\beta \to \beta^{-1}$ is broken by choosing $V_{\langle 1,2 \rangle}^d$ rather than $V_{\langle 2,1 \rangle}^d$, such CFTs depend on $\beta^2$ (1.15) rather than on the central charge $c$.

Next, let us determine the OPEs of degenerate fields with other primary fields. The OPE of $V_{\langle 2,1 \rangle}^d$ or $V_{\langle 1,2 \rangle}^d$ with a given primary field is constrained by the fusion rules (1.62), which allow 2 left-moving momentums and 2 right-moving momentums, for a total of 4 representations. In the case of a diagonal field $V_P$, the requirement that only spinless primary fields appear single out 2 of the 4 possibilities:

$$\boxed{V_{\langle 2,1 \rangle}^d V_P \sim \sum_{\pm} V_{P \pm \frac{\beta}{2}}} \quad , \quad \boxed{V_{\langle 1,2 \rangle}^d V_P \sim \sum_{\pm} V_{P \pm \frac{1}{2\beta}}} \ . \tag{2.7}$$

In the case of a non-diagonal field $V_{(r,s)}$, the 4 fields that can a priori appear are

$$V_{\langle 2,1 \rangle}^d V_{(r,s)} \subset \sum_{\pm} V_{(r\pm 1,s)} + \sum_{\pm} V_{(r,s\pm\beta^2)} \ , \tag{2.8a}$$

$$V_{\langle 1,2 \rangle}^d V_{(r,s)} \subset \sum_{\pm} V_{(r\pm\beta^{-2},s)} + \sum_{\pm} V_{(r,s\pm 1)} \ . \tag{2.8b}$$

Since all fields have half-integer spins, the following spin differences must also be half-integer whenever the corresponding fields exist:

$$S_{(r\pm 1,s)} - S_{(r,s)} = \mp s \quad , \quad S_{(r,s\pm\beta^2)} - S_{(r,s)} = \mp r\beta^2 \ , \tag{2.9a}$$

$$S_{(r\pm\beta^{-2},s)} - S_{(r,s)} = \mp s\beta^{-2} \quad , \quad S_{(r,s\pm 1)} - S_{(r,s)} = \mp r \ . \tag{2.9b}$$

We now assume that $\beta^2 \notin \mathbb{Q}$, and remember $rs \in \frac{1}{2}\mathbb{Z}$. It follows that $V_{(r\pm 1,s)}$ and $V_{(r,s\pm 1)}$ can coexist neither with $V_{(r,s\pm\beta^2)}$ nor with $V_{(r\pm\beta^{-2},s)}$. Therefore, the OPEs $V_{\langle 2,1 \rangle}^d V_{(r,s)}$ and

$V_{\langle 1,2\rangle}^d V_{(r,s)}$ each contain at most 2 primary fields, out of the 4 that are allowed by fusion rules. For notational simplicity, we choose these 2 primary fields as follows:

$$\boxed{V_{\langle 2,1\rangle}^d V_{(r,s)} \sim \sum_\pm V_{(r\pm 1,s)}} \quad , \quad \boxed{V_{\langle 1,2\rangle}^d V_{(r,s)} \sim \sum_\pm V_{(r,s\pm 1)}} \; . \tag{2.10}$$

While they were derived under the assumption $\beta^2 \notin \mathbb{Q}$, we will assume that these OPEs are valid for any value of the central charge.

The OPEs (2.10) and (2.7) indicate which primary fields can appear: we now assume that in each one of these OPEs, the 2 fields that can appear do appear, i.e. $V_{\langle 1,2\rangle}^d, V_{(r,s)} \in \widehat{\mathcal{S}} \implies V_{(r,s\pm 1)} \in \widehat{\mathcal{S}}$. So the conformal spin (2.4) must remain half-integer under $s \to s\pm 1$, therefore $r \in \frac{1}{2}\mathbb{Z}$. Moreover, the OPE (1.94) implies that the monodromy of $V_{\langle 1,2\rangle}^d$ around $V_{(r,s)}$ is $e^{2\pi i r} \in \{1,-1\}$. In the $N+1$-point function $\left\langle V_{\langle 1,2\rangle}^d \prod_{i=1}^N V_{(r_i,s_i)} \right\rangle$, the product of the monodromies of $V_{\langle 1,2\rangle}^d$ must be 1, therefore $\sum_{i=1}^N r_i \in \mathbb{Z}$. This conclusion also holds for $N$-point functions of the type $\left\langle \prod_{i=1}^N V_{(r_i,s_i)} \right\rangle$, which may be deduced from our $N+1$-point function using an OPE. Therefore, if $V_{\langle 1,2\rangle}^d$ exists, then the first Kac index of non-diagonal fields $V_{(r,s)}$ is half-integer and conserved modulo integers:

$$\boxed{V_{\langle 1,2\rangle}^d \in \widehat{\mathcal{S}} \quad \implies \quad r \in \frac{1}{2}\mathbb{Z} \quad , \quad \sum_{i=1}^N r_i \in \mathbb{Z}} \; . \tag{2.11a}$$

Similarly, if $V_{\langle 2,1\rangle}^d$ exists, then the same is true for the second Kac index,

$$\boxed{V_{\langle 2,1\rangle}^d \in \widehat{\mathcal{S}} \quad \implies \quad s \in \frac{1}{2}\mathbb{Z} \quad , \quad \sum_{i=1}^N s_i \in \mathbb{Z}} \; . \tag{2.11b}$$

These constraints also apply in the presence of diagonal fields, if we set $r = s = 0$ for any diagonal field.

Finally, a necessary condition for an OPE $V_1 V_2 \sim \sum_{k \in \mathcal{S}^{12}} V_k$ (1.94) to converge is that the total conformal dimension be bounded from below over the OPE spectrum $\mathcal{S}^{12}$,

$$\boxed{\inf_{k \in \mathcal{S}^{12}} \Re\left(\Delta_k + \bar{\Delta}_k\right) > -\infty} \; . \tag{2.12}$$

The total conformal dimension of a non-diagonal field $V_{(r,s)}$ is

$$\Delta_{(r,s)} + \Delta_{(r,-s)} = \frac{1}{2}\left[r^2\beta^2 + s^2\beta^{-2} - \left(\beta - \beta^{-1}\right)^2\right] \; . \tag{2.13}$$

If $V_{\langle 1,2\rangle}^d$ exists, then the OPE (2.10) increments $s$ by 1, therefore $s$ is unbounded in real part i.e. $\sup_{V_{(r,s)} \in \widehat{\mathcal{S}}}(\Re s) = \infty$. Assuming that this holds not only for the extended spectrum $\widehat{\mathcal{S}}$ but also for some OPE spectrum $\mathcal{S}^{12}$, we deduce $\Re\beta^{-2} > 0$, or equivalently

$$\boxed{\Re\beta^2 > 0 \quad \text{i.e.} \quad \Re c < 13} \; . \tag{2.14}$$

This constraint holds for all known non-diagonal CFTs with at least 1 degenerate field.

## 2.2 Diagonal CFTs with 2 degenerate fields

### 2.2.1 Generalized minimal models

For $\beta^2 \in \mathbb{C}^* \backslash \mathbb{Q}$, we define the **generalized minimal model (GMM)** as the CFT whose spectrum of primary fields is made of all diagonal degenerate fields:

$$\boxed{\mathcal{S}^{\mathrm{GMM}} = \left\{ V_{\langle r,s \rangle}^d \right\}_{r,s \in \mathbb{N}^*}} . \tag{2.15}$$

Thanks to the degenerate fusion rules (1.66), any OPE spectrum is finite, and therefore respects the convergence condition (2.12).

For $\beta^2 \in \mathbb{Q}$, the degenerate dimensions $\Delta_{(r,s)}$ are not all different, and there is no GMM. However, there exist other CFTs whose primary fields are diagonal and (partly or fully) degenerate, starting with A-series minimal models. One way to explore this issue is to take limits of GMM correlation functions as $\beta^2 \to \beta_0^2 \in \mathbb{Q}$. This can give rise not only to minimal model correlation functions, but also to logarithmic correlation functions [18], suggesting the existence of non-minimal CFTs.

Generalized minimal models exist only on the sphere. In particular, their torus partition functions are infinite, because $\lim_{r,s \to \infty} \Delta_{(r,s)} \neq +\infty$. Nevertheless, GMMs are interesting because they are very simple and natural CFTs, which give rise to various other CFTs in certain limits — starting with Liouville theory.

### 2.2.2 Liouville theory

Let us consider the spectrum of a generalized minimal model with $\beta^2 \in \mathbb{R}_{>0} \backslash \mathbb{Q}$. The momentums $P_{(r,s)}$ (1.18) are dense in the real line, i.e. $\mathbb{R} = \mathrm{closure}\left( \left\{ P_{(r,s)} \right\}_{r,s \in \mathbb{N}^*} \right)$. Any diagonal field with a real momentum can therefore be obtained as a limit of degenerate fields,

$$\forall P \in \mathbb{R} , \quad V_P = \lim_{\substack{r,s \to \infty \\ P_{(r,s)} \to P}} V_{\langle r,s \rangle}^d . \tag{2.16}$$

We assume that the resulting fields are only characterized by their conformal dimensions. This implies that $V_P$ does not depend on how exactly we take the limit, and also that $V_P = V_{-P}$. The study of correlation functions in Section 3 will justify this assumption.

If we apply the limit to both fields in the OPE $V_{\langle r_1,s_1 \rangle}^d V_{\langle r_2,s_2 \rangle}^d$, the resulting OPE is formally given by the degenerate fusion rules (1.66) with $r_1, s_1, r_2, s_2 \to \infty$, and we obtain

$$\boxed{V_{P_1} V_{P_2} \sim \frac{1}{2} \int_{\mathbb{R}} dP \, V_P} , \tag{2.17}$$

where the factor $\frac{1}{2}$ is because $V_P = V_{-P}$. Now, remember from Section 1.2.3 that OPE coefficients have poles when the momentum $P$ becomes degenerate, and these poles are dense in the real line. The integral in the OPE $V_{P_1} V_{P_2}$ is therefore formally divergent. But it can be regularized by shifting the integration line into the complex plane:

$$V_{P_1} V_{P_2} \sim \frac{1}{2} \int_{\mathbb{R}+i\epsilon} dP \, V_P . \tag{2.18}$$

Let us call **Liouville theory** the resulting CFT. The spectrum of primary fields is formally

$$\boxed{\mathcal{S}^{\mathrm{Liouville}} = \frac{1}{2} \left\{ V_P \right\}_{P \in \mathbb{R}}} . \tag{2.19}$$

The regularized OPE (2.18) requires that we extend the spectrum to complex momentums. This will turn out to be possible, because correlation functions are analytic in $P$.

For the moment, we have defined Liouville theory for $\beta^2 \in \mathbb{R}_{>0}\backslash\mathbb{Q}$. By taking limits, we can extend it to the half-line $c \leq 1$ i.e. $\beta^2 \in \mathbb{R}_{>0}$. But we cannot do an analytic continuation to the rest of the complex $c$-plane, because of the poles of OPE coefficients. These poles lie on the real $P$-line if $c \leq 1$, but as soon as $c$ strays from this half-line, the poles spread out in the complex $P$-plane, and cross the integration line $\mathbb{R} + i\epsilon$ for any $\epsilon > 0$. In the following figures, we draw the regions where poles are found in blue, and the poles $\pm P_{(1,1)}$ as blue circles; the integration line is in red:

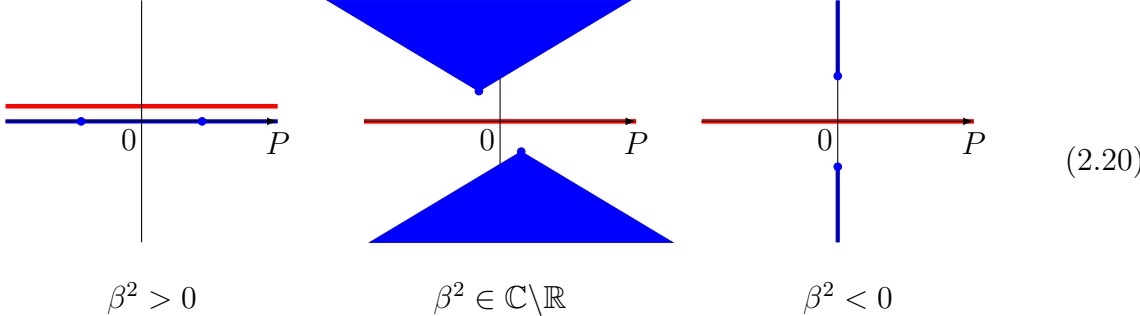

$$\beta^2 > 0 \qquad\qquad \beta^2 \in \mathbb{C}\backslash\mathbb{R} \qquad\qquad \beta^2 < 0 \qquad\qquad (2.20)$$

We nevertheless define Liouville theory for $\beta^2 \in \mathbb{C}\backslash\mathbb{R}_{\geq 0}$ by the spectrum (2.19) and OPE (2.17), where we no longer need to shift the integration line. We also assume that correlation functions are analytic in momentums and in the central charge, and that degenerate fields exist, so that we have the extended spectrum

$$\boxed{\widehat{\mathcal{S}}^{\text{Liouville}} = \frac{1}{2}\left\{V_P\right\}_{P\in\mathbb{C}} \cup \left\{V^d_{\langle r,s\rangle}\right\}_{r,s\in\mathbb{N}^*}} \ . \qquad (2.21)$$

The resulting extended CFT therefore includes a generalized minimal model.

### 2.2.3  A-series minimal models

Let us look for rational, diagonal CFTs. To any degenerate representation of the Virasoro algebra, we can associate a diagonal primary field. We conjecture that this leads to a bijection

$$\text{Finite sets of degenerate representations that are closed under fusion}$$

$$\Longleftrightarrow \qquad\qquad (2.22)$$

$$\text{Rational, diagonal CFTs on the sphere}$$

To the best of our knowledge, there is no known counterexample to this conjecture (and no proof either). From the fusion rules (1.66), no nontrivial finite set of simply degenerate representations can be closed under fusion. We therefore need multiply degenerate representations, which implies $\beta^2 \in \mathbb{Q}$.

In the case $\beta^2 = -\frac{q}{p} < 0$, with $p, q \in \mathbb{N}^*$ coprime, there are no finite sets of degenerate fields that are closed under OPEs. To look for such sets, we could focus on the simplest doubly degenerate field,

$$V^f_{\langle p+1,1\rangle} = V^f_{\langle 1,q+1\rangle} \ . \qquad (2.23)$$

Repeated OPEs of this field with itself generate infinitely many fields of the type $V^f_{\langle kp+1,1\rangle}$ for any $k \in \mathbb{N}$, starting with

$$V^f_{\langle p+1,1\rangle} V^f_{\langle p+1,1\rangle} \sim V^f_{\langle 1,1\rangle} + V^f_{\langle 2p+1,1\rangle} \ . \qquad (2.24)$$

To show that we obtain fully degenerate fields, we can use OPE associativity, as well as OPEs with generic diagonal fields (1.65), starting with

$$V_{\langle p+1,1\rangle}^f V_P \sim \sum_{\pm} V_{P\pm\frac{1}{2}\sqrt{-pq}} \ , \tag{2.25}$$

where $\{P \pm \frac{1}{2}\sqrt{-pq}\}_{\pm} = \{P + i\beta\}_{i=-\frac{p}{2},\cdots,\frac{p}{2}} \cap \{P + j\beta^{-1}\}_{j=-\frac{q}{2},\cdots,\frac{q}{2}}$.

We therefore focus on the case $\beta^2 = \frac{q}{p} > 0$ with $p, q \in \mathbb{N}^*$ coprime. For simplicity, we will only study the representations that are generated by $\mathcal{R}_{\langle 2,1\rangle}^f$ and $\mathcal{R}_{\langle 1,2\rangle}^f$. This is enough for recovering the known A-series minimal models, but not for investigating other diagonal, rational CFTs that might exist on the sphere. Let us show that $\mathcal{R}_{\langle 2,1\rangle}^f$ and $\mathcal{R}_{\langle 1,2\rangle}^f$ generate the representations $\mathcal{R}_{\langle r,s\rangle}^f$ whose indices belong to the **Kac table**

$$\boxed{K_{p,q} = \big((0,p)\times(0,q)\big)\bigcap\big(\mathbb{N}\times\mathbb{N}\big)} \ . \tag{2.26}$$

We will in fact show that $\left\{\mathcal{R}_{\langle r,s\rangle}^f\right\}_{(r,s)\in K_{p,q}}$ is stable under fusion. Representations in this set are doubly degenerate,

$$\mathcal{R}_{\langle r,s\rangle}^f = \mathcal{R}_{\langle p-r,q-s\rangle}^f \ , \tag{2.27}$$

and this leads to the following constraints on fusion rules (1.66)

$$\mathcal{R}_{\langle r_1,s_1\rangle}^f \times \mathcal{R}_{\langle r_2,s_2\rangle}^f \subset \sum_{r\overset{2}{=}|r_1-r_2|+1}^{r_1+r_2-1}\sum_{s\overset{2}{=}|s_1-s_2|+1}^{s_1+s_2-1} \mathcal{R}_{\langle r,s\rangle}^d \ , \tag{2.28a}$$

$$\mathcal{R}_{\langle r_1,s_1\rangle}^f \times \mathcal{R}_{\langle r_2,s_2\rangle}^f \subset \sum_{r'\overset{2}{=}|r_1-r_2|+1}^{2p-r_1-r_2-1}\sum_{s'\overset{2}{=}|s_1-s_2|+1}^{2q-s_1-s_2-1} \mathcal{R}_{\langle r',s'\rangle}^d \ , \tag{2.28b}$$

where $\mathcal{R}_{\langle r,s\rangle}^d$ is a representation that may or may not be fully degenerate. Now, $\mathcal{R}_{\langle r,s\rangle}^d = \mathcal{R}_{\langle r',s'\rangle}^d$ implies $(r,s) = (r',s')$ or $(r,s) = (p-r',q-s')$, but the second equality is impossible due to parity: since $p$ and $q$ are coprime, one of them must be odd, say $p$, but $r \equiv r' \equiv |r_1 - r_2| + 1 \bmod 2$. We therefore obtain the fusion rules

$$\boxed{\mathcal{R}_{\langle r_1,s_1\rangle}^f \times \mathcal{R}_{\langle r_2,s_2\rangle}^f = \sum_{r\overset{2}{=}|r_1-r_2|+1}^{\min(r_1+r_2,2p-r_1-r_2)-1}\sum_{s\overset{2}{=}|s_1-s_2|+1}^{\min(s_1+s_2,2q-s_1-s_2)-1} \mathcal{R}_{\langle r,s\rangle}^f} \ , \tag{2.29}$$

where we only generate Kac indices in the Kac table $(r,s) \in K_{p,q}$ as advertised. The number of terms is $\min(r_1,r_2,p-r_1,p-r_2)\min(s_1,s_2,q-s_1,q-s_2)$. To show that $\mathcal{R}_{\langle r,s\rangle}^f$ is fully degenerate, it is enough to realize that a Kac table representation is fully degenerate if and only if its fusion with a generic Verma module $\mathcal{V}_P$ is zero. From Eq. (1.65),

$$\mathcal{R}_{\langle r,s\rangle}^f \times \mathcal{V}_P \subset \sum_{i=-\frac{r-1}{2}}^{\frac{r-1}{2}}\sum_{j=-\frac{s-1}{2}}^{\frac{s-1}{2}} \mathcal{V}_{P+i\beta+j\beta^{-1}} \cap \sum_{i'=-\frac{p-r-1}{2}}^{\frac{p-r-1}{2}}\sum_{j'=-\frac{q-s-1}{2}}^{\frac{q-s-1}{2}} \mathcal{V}_{P+i'\beta+j'\beta^{-1}} \ . \tag{2.30}$$

The modules $\mathcal{V}_{P+i\beta+j\beta^{-1}}$ and $\mathcal{V}_{P+i'\beta+j'\beta^{-1}}$ coincide if and only if their momentums are opposite or equal. If they are opposite, $P = -P_{(i+i',j+j')}$ must belong to the (reflected) Kac table, so it is not generic. If they are equal, we have $i\beta + j\beta^{-1} = i'\beta + j'\beta^{-1}$ i.e.

$2(i - i')q = 2(j' - j)p$. But parity forbids $(i, j) = (i', j')$, since $p$ or $q$ is odd. And we have $2|i - i'| < p$ and $2|j - j'| < q$, so our equality cannot be satisfied, and

$$\mathcal{R}^f_{\langle r,s\rangle} \times \mathcal{V}_P = 0 \ . \tag{2.31}$$

Therefore, by associativity of the fusion product, the product of 2 fully degenerate representations is again a combination of fully degenerate representations.

We therefore define the **A-series minimal model** $\mathrm{AMM}_{p,q}$ by its spectrum, made of fully degenerate fields in the Kac table:

$$\boxed{\mathcal{S}^{\mathrm{AMM}_{p,q}} = \frac{1}{2} \left\{ V^f_{\langle r,s\rangle} \right\}_{(r,s) \in K_{p,q}} \quad \text{with} \quad 2 \leq q < p \quad \text{and} \quad p, q \text{ coprime}} \ . \tag{2.32}$$

The factor $\frac{1}{2}$ accounts for the $\mathbb{Z}_2$ symmetry (2.27). We assume $2 \leq q$ for the Kac table to be non-empty, and $q < p$ because $\mathrm{AMM}_{p,q} = \mathrm{AMM}_{q,p}$.

For example, $\mathrm{AMM}_{4,3}$ has the central charge $c = \frac{1}{2}$ and describes some of the observables of the critical Ising model. Let us describe its 3 primary fields, and display their dimensions in the Kac table:

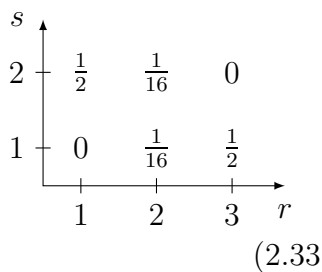

| Field | $V^f_{\langle 1,1\rangle} = V^f_{\langle 3,2\rangle}$ | $V^f_{\langle 1,2\rangle} = V^f_{\langle 3,1\rangle}$ | $V^f_{\langle 2,1\rangle} = V^f_{\langle 2,2\rangle}$ |
|---|---|---|---|
| Notation | $I$ | $\epsilon$ | $\sigma$ |
| Name | Identity | Energy | Spin |
| Dimension | $0$ | $\frac{1}{2}$ | $\frac{1}{16}$ |

$$\tag{2.33}$$

Having only 3 primary fields, we can afford to write the complete fusion rules:

$$
\begin{aligned}
I \times I &= I \ , & \epsilon \times \epsilon &= I \ , \\
I \times \epsilon &= \epsilon \ , & \epsilon \times \sigma &= \sigma \ , \\
I \times \sigma &= \sigma \ , & \sigma \times \sigma &= I + \epsilon \ .
\end{aligned}
\tag{2.34}
$$

For example, $\mathrm{AMM}_{7,4}$ has the central charges $c = -\frac{13}{14}$. Its 9 primary fields have the following dimensions:

$$\tag{2.35}$$

### 2.2.4 Runkel–Watts-type CFTs

Let us consider a limit of A-series minimal models where $\frac{q}{p} \to \beta_0^2 > 0$. We first take the Kac table indices to be fixed. The Kac table becomes infinite, which suggests

$$\lim_{\substack{\frac{q}{p} \to \beta_0^2 \\ r,s \text{ fixed}}} \mathrm{AMM}_{p,q} = \mathrm{GMM} \ . \tag{2.36}$$

Since we obtain arbitrary degenerate fields $V^d_{\langle r,s \rangle}$ in the limit, it is also possible to send the indices $r, s$ to $\infty$, and therefore to send the momentums $P_{(r,s)}$ to arbitrary values $P \in \mathbb{R}$, just as in Eq. (2.16). Therefore, Liouville theory with $c \leq 1$ is not only a limit of generalized minimal models, but also a limit of minimal models:

$$\lim_{\substack{\frac{q}{p} \to \beta_0^2 \\ P_{(r,s)} \to P}} \mathrm{AMM}_{p,q} = \text{Liouville} . \tag{2.37}$$

However, if $\beta_0^2 \in \mathbb{Q}$, it is also possible to fine-tune the behaviour of $p, q, r, s$ such that constraints from the minimal model's fusion rules survive in the limit. As a result, we can obtain not only Liouville theory with its trivial OPE (2.17), but also another diagonal CFT with the same continuous spectrum but a non-trivial OPE, called a **Runkel–Watts-type theory** after the original example at $c = 1$, discovered by Runkel and Watts [19].

To begin with, we write $\beta_0^2 = \frac{q_0}{p_0}$ with $p_0$ and $q_0$ coprime integers, equivalently $\mathbb{Z}\beta_0 + \mathbb{Z}\beta_0^{-1} = \frac{1}{\sqrt{p_0 q_0}}\mathbb{Z}$. There exist infinitely many pairs $(p, q) \in \mathbb{N}^2$ such that

$$qp_0 - pq_0 = 1 . \tag{2.38}$$

Choosing such pairs amounts to fine-tuning the values of $p, q$, because there are many other ways to satisfy $\lim_{p,q \to \infty} \frac{q}{p} = \frac{q_0}{p_0}$, which is equivalent to $qp_0 - pq_0 \ll p$. For fixed $r, s$, we compute the momentum

$$P_{(r,s)} = \frac{1}{2\sqrt{p_0 q_0}} \left[ rq_0 - sp_0 + \frac{1}{2}\left(\frac{r}{p} + \frac{s}{q}\right) + O\left(\frac{1}{p^2}\right) \right] . \tag{2.39}$$

We now fine-tune $r, s$ such that

$$\lim(rq_0 - sp_0), \lim\frac{r}{p}, \lim\frac{s}{q} < \infty \quad , \quad \lim\frac{r}{p} = \lim\frac{s}{q} . \tag{2.40}$$

To achieve this fine-tuning, it is enough to set

$$\begin{cases} r = r_0 + kp_0 , \\ s = s_0 + kq_0 , \end{cases} \quad \text{with} \quad k = O(p) \in \mathbb{N} . \tag{2.41}$$

This leads to

$$\lim_{p,q \to \infty} P_{(r,s)} = \frac{1}{2\sqrt{p_0 q_0}} (n + x) \quad \text{with} \quad \begin{cases} n = r_0 q_0 - s_0 p_0 \in \mathbb{Z} , \\ x = \lim\frac{r}{p} = \lim\frac{s}{q} \in (0, 1) . \end{cases} \tag{2.42}$$

Now what happens to the minimal model fusion rules $\mathcal{R}^f_{\langle r_1,s_1 \rangle} \times \mathcal{R}^f_{\langle r_2,s_2 \rangle}$ (2.29) when we apply our limit to both representations? Let us examine the momentums (2.39) of the resulting representations $\mathcal{R}^f_{\langle r,s \rangle}$. The combination $n = rq_0 - sp_0$ takes integer values such that $n \equiv n_1 + n_2 + p_0 + q_0 \bmod 2$, while both $\frac{r}{p}$ and $\frac{s}{q}$, and therefore also $x = \frac{1}{2}\left(\frac{r}{p} + \frac{s}{q}\right)$, are evenly distributed in the interval $(|x_1 - x_2|, \min(x_1 + x_2, 2 - x_1 - x_2))$. Writing the primary fields as $V_{n,x} = \lim V^f_{(r,s)}$, we thus find the OPE

$$\boxed{V_{n_1,x_1} V_{n_2,x_2} \sim \sum_{n \in n_1 + n_2 + p_0 + q_0 + 2\mathbb{Z}} \int_{|x_1 - x_2|}^{\min(x_1 + x_2, 2 - x_1 - x_2)} dx \, V_{n,x}} . \tag{2.43}$$

Of course, $V_{n,x}$ is just another notation for the diagonal field $V_P$, with $n, x$ the integer and fractional parts of $2\sqrt{p_0 q_0}P$:

$$P = \frac{n+x}{2\sqrt{p_0 q_0}} \quad , \quad n = \left\lfloor 2\sqrt{p_0 q_0}P \right\rfloor \quad , \quad x = \left\{ 2\sqrt{p_0 q_0}P \right\} . \tag{2.44}$$

The reflection relation $V_P = V_{-P}$ is now $V_{n,x} = V_{-n-1,1-x}$. The resulting Runkel–Watts-type CFT has the same spectrum as Liouville theory (2.19) $\mathcal{S}^{\text{RWT}_{p_0,q_0}} = \frac{1}{2}\{V_P\}_{P \in \mathbb{R}}$. In contrast to Liouville theory, the momentums cannot be continued to complex values, where the OPE would no longer make sense. Notice however that if $x_1 \neq x_2$, the OPE does not produce fields with degenerate momentums, i.e. fields of the type $V_{n,0}$. So there is no need to regularize the integral over $x$. Alternatively, the OPE may be written in a manifestly reflection-invariant way using the momentums $P_i$ [16],

$$\left\langle \prod_{i=1}^{3} V_{P_i} \right\rangle \neq 0 \implies \prod_{\pm,\pm} \sin \pi \left( \tfrac{p_0+q_0}{2} + \sqrt{p_0 q_0}(P_1 \pm P_2 \pm P_3) \right) < 0 . \tag{2.45}$$

It turns out that Runkel–Watts-type theories can also be obtained as limits of Liouville theory with $\beta^2 \in \mathbb{C}\backslash\mathbb{R}$ [20, 21]. The limit of Liouville theory for $\beta^2 \to \beta_0^2 > 0$ generally does not make sense, but it exists if $\beta_0^2$ is rational and the momentums are real. In this limit, the condition (2.45) emerges from the 3-point structure constants of Liouville theory.

## 2.3 Non-diagonal CFTs with 2 degenerate fields

### 2.3.1 D-series minimal models

Let us look for rational, non-diagonal CFTs, built from the same representations as A-series minimal models: fully degenerate representations with indices in the Kac table (2.26), with $\beta^2 = \frac{q}{p} \in \mathbb{Q}_{>0}$. For any non-diagonal field $V_{(r,s)}$, we require that the left and right dimensions $\Delta_{(r,\pm s)}$ both coincide with dimensions from the Kac table. To achieve this, a simple ansatz is

$$(r,s) \in \left[ \left(-\tfrac{p}{2}, \tfrac{p}{2}\right) \cap \left(\mathbb{Z} + \tfrac{p}{2}\right) \right] \times \left[ \left(-\tfrac{q}{2}, \tfrac{q}{2}\right) \cap \left(\mathbb{Z} + \tfrac{q}{2}\right) \right] . \tag{2.46}$$

This amounts to taking advantage of the identity $\Delta_{(r,s)} = \Delta_{(r+\frac{p}{2}, s+\frac{q}{2})}$ for centering the Kac table at $(r,s) = (0,0)$, which makes the table invariant under $(r,s) \to (r,-s)$. It remains to impose the integer spin condition $rs \in \mathbb{Z}$. If $p, q$ are both odd, no field has integer spin. Without loss of generality, we therefore assume $p \in 2\mathbb{N}^*$ and $q \in 2\mathbb{N}+1$, and the integer spin condition reduces to $r \in 2\mathbb{Z}$. This completes the determination of the non-diagonal sector.

Since $q$ is odd, all non-diagonal fields have $s \in \mathbb{Z} + \frac{1}{2}$, and the conservation of $s \bmod \mathbb{Z}$ (2.11b) (with $s = 0$ for diagonal fields) reduces to the **conservation of diagonality**:

$$D \times D = D \quad , \quad D \times N = N \quad , \quad N \times N = D , \tag{2.47}$$

where $D$ stands for diagonal and $N$ for non-diagonal fields. Moreover, we assume that the diagonal sector is generated by taking OPEs of non-diagonal fields. By the degenerate fusion rules (1.66), the diagonal sector involves representations $\mathcal{R}^f_{\langle r,s \rangle}$ with $r \in 2\mathbb{N}+1$. We therefore define a **D-series minimal model** $\text{DMM}_{p,q}$ by the spectrum

$$\boxed{\mathcal{S}^{\text{DMM}_{p,q}} = \frac{1}{2}\left\{ V^f_{\langle r,s \rangle} \right\}_{\substack{(r,s) \in K_{p,q} \\ r \equiv 1 \bmod 2}} \bigcup \frac{1}{2}\left\{ V_{(r,s)} \right\}_{\substack{(r,s) \in (-\frac{p}{2}, \frac{p}{2}) \times (-\frac{q}{2}, \frac{q}{2}) \\ (r,s) \in 2\mathbb{Z} \times (\mathbb{Z} + \frac{1}{2})}}} , \tag{2.48}$$

where the non-diagonal fields $V_{(r,s)}$ are fully degenerate. This is valid for

$$\boxed{p \in 2\mathbb{N} + 6 \quad , \quad q \in 2\mathbb{N} + 3 \quad , \quad p, q \text{ coprime}} , \tag{2.49}$$

where the lower bounds $p \geq 6$ and $q \geq 3$ ensure that the non-diagonal sector contains fields with nonzero spins. If we allow spins to be half-integer, we obtain the extended spectrum

$$\widehat{\mathcal{S}}^{\mathrm{DMM}_{p,q}} = \frac{1}{2} \left\{ V_{\langle r,s \rangle}^f \right\}_{\substack{(r,s) \in (0,p) \times (0,q) \\ (r,s) \in \mathbb{N} \times \mathbb{N}}} \bigcup \frac{1}{2} \left\{ V_{(r,s)} \right\}_{\substack{(r,s) \in (-\frac{p}{2}, \frac{p}{2}) \times (-\frac{q}{2}, \frac{q}{2}) \\ (r,s) \in \mathbb{Z} \times (\mathbb{Z} + \frac{1}{2})}} . \tag{2.50}$$

The corresponding CFT may be called a **fermionic minimal model** $\widehat{\mathrm{DMM}}_{p,q}$ [22]. The numbers of primary fields in these minimal models are:

| Model | $\mathrm{AMM}_{p,q}$ | $\mathrm{DMM}_{p,q}$ | | $\widehat{\mathrm{DMM}}_{p,q}$ |
|---|---|---|---|---|
| | | $p \equiv 0 \bmod 4$ | $p \equiv 2 \bmod 4$ | |
| Diagonal | $\frac{1}{2}(p-1)(q-1)$ | $\frac{1}{4}p(q-1)$ | | $\frac{1}{2}(p-1)(q-1)$ |
| Non-diagonal | $0$ | $\frac{1}{4}(p-2)(q-1)$ | $\frac{1}{4}p(q-1)$ | $\frac{1}{2}(p-1)(q-1)$ |
| Total | $\frac{1}{2}(p-1)(q-1)$ | $\frac{1}{2}(p-1)(q-1)$ | $\frac{1}{2}p(q-1)$ | $(p-1)(q-1)$ |

$$\tag{2.51}$$

The fusion rules of D-series and fermionic minimal models are obtained by combining the fully degenerate fusion rules (2.29) with the conservation of diagonality:

$$V_{\langle r_1, s_1 \rangle}^f V_{(r_2, s_2)} \sim \sum_{r \overset{2}{=} \max(r_2-r_1, r_1-r_2-p)+1}^{\min(r_1+r_2, p-r_1-r_2)-1} \sum_{s \overset{2}{=} \max(s_2-s_1, s_1-s_2-q)+1}^{\min(s_1+s_2, q-s_1-s_2)-1} V_{(r,s)} , \tag{2.52a}$$

$$V_{(r_1, s_1)} V_{(r_2, s_2)} \sim \sum_{r \overset{2}{=} |r_1-r_2|+1}^{p-|r_1+r_2|-1} \sum_{s \overset{2}{=} |s_1-s_2|+1}^{q-|s_1+s_2|-1} V_{\langle r,s \rangle}^f . \tag{2.52b}$$

Curiously, the diagonal field $V_{\langle \frac{p}{2}, s \rangle}^f$ has the same conformal dimensions as the non-diagonal field $V_{(0, s-\frac{q}{2})}$: this an instance of a nontrivial **field multiplicity**. However, these 2 fields have different OPEs.

For example, let us compare the diagonal and non-diagonal minimal models $\mathrm{AMM}_{6,5}$ and $\mathrm{DMM}_{6,5}$, which both have $c = \frac{4}{5}$, and respectively describe observables of the tetracritical Ising model and of the critical 3-state Potts model. The Virasoro representations that appear in the spectrums of both models belong to the Kac table $K_{6,5}$. Each entry in the table correspond to a primary field of $\mathrm{AMM}_{6,5}$. Since $p = 6 \equiv 2 \bmod 4$, both sectors of $\mathrm{DMM}_{6,5}$ are built from representations $\mathcal{R}_{\langle r,s \rangle}^f$ with $r \in 2\mathbb{N} + 1$. We indicate this in the Kac table by having the $r \in 2\mathbb{N} + 1$ columns both boxed (for the diagonal sector) and

highlighted (for the non-diagonal sector):

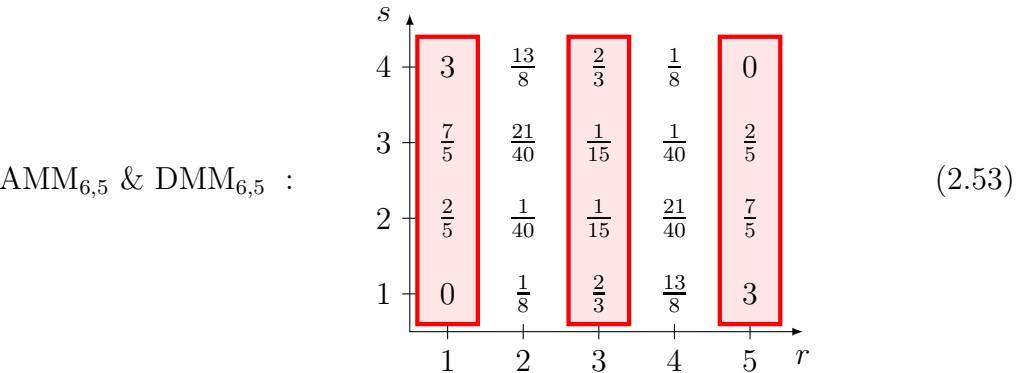

$$\text{AMM}_{6,5} \text{ \& } \text{DMM}_{6,5} \; : \tag{2.53}$$

The 10 primary fields of $\text{AMM}_{6,5}$ are

$$V^f_{\langle 1,1\rangle}, V^f_{\langle 2,1\rangle}, V^f_{\langle 3,1\rangle}, V^f_{\langle 4,1\rangle}, V^f_{\langle 5,1\rangle}, V^f_{\langle 1,2\rangle}, V^f_{\langle 2,2\rangle}, V^f_{\langle 3,2\rangle}, V^f_{\langle 4,2\rangle}, V^f_{\langle 5,2\rangle} \; . \tag{2.54}$$

In $\text{DMM}_{6,5}$, only 6 of these fields are present, and they are complemented by 6 non-diagonal fields. For each one of these 12 fields, let us indicate the left and right Kac indices of the corresponding degenerate representations:

| Field | $V^f_{\langle 1,1\rangle}$ | $V^f_{\langle 3,1\rangle}$ | $V^f_{\langle 5,1\rangle}$ | $V^f_{\langle 1,2\rangle}$ | $V^f_{\langle 3,2\rangle}$ | $V^f_{\langle 5,2\rangle}$ |
|---|---|---|---|---|---|---|
| Indices | $(1,1)(1,1)$ | $(3,1)(3,1)$ | $(5,1)(5,1)$ | $(1,2)(1,2)$ | $(3,2)(3,2)$ | $(5,2)(5,2)$ |

| Field | $V_{(0,\frac{1}{2})}$ | $V_{(0,\frac{3}{2})}$ | $V_{(2,-\frac{3}{2})}$ | $V_{(2,-\frac{1}{2})}$ | $V_{(2,\frac{1}{2})}$ | $V_{(2,\frac{3}{2})}$ |
|---|---|---|---|---|---|---|
| Indices | $(3,2)(3,2)$ | $(3,1)(3,1)$ | $(5,1)(1,1)$ | $(5,2)(1,2)$ | $(1,2)(5,2)$ | $(1,1)(5,1)$ |

$$\tag{2.55}$$

We therefore have the coincidences of conformal dimensions $V_{(0,\frac{1}{2})} \simeq V^f_{\langle 3,2\rangle}$ and $V_{(0,\frac{3}{2})} \simeq V^f_{\langle 3,1\rangle}$. The OPEs however do not coincide, for example

$$V^f_{\langle 3,1\rangle} V^f_{\langle 3,1\rangle} \sim V^f_{\langle 1,1\rangle} + V^f_{\langle 3,1\rangle} + V^f_{\langle 5,1\rangle} \quad , \quad V^f_{\langle 3,1\rangle} V_{(0,\frac{3}{2})} \sim V_{(2,-\frac{3}{2})} + V_{(0,\frac{3}{2})} + V_{(2,\frac{3}{2})} \; . \tag{2.56}$$

As another example, let us compare the diagonal and non-diagonal minimal models $\text{AMM}_{8,5}$ and $\text{DMM}_{8,5}$, which both have $c = -\frac{7}{20}$. The Virasoro representations that appear in the spectrums of both models belong to the Kac table $K_{8,5}$. Each entry in the table correspond to a primary field of $\text{AMM}_{8,5}$. Since $p = 8 \equiv 0 \bmod 4$, the diagonal sector of $\text{DMM}_{8,5}$ is built from representations $\mathcal{R}^f_{\langle r,s\rangle}$ with $r \in 2\mathbb{N}+1$ (boxed), while the non-diagonal sector is built from $\mathcal{R}^f_{\langle r,s\rangle}$ with $r \in 2\mathbb{N}$ (highlighted):

$$\text{AMM}_{8,5} \text{ \& } \text{DMM}_{8,5} \; : \tag{2.57}$$

The diagonal model $\text{AMM}_{8,5}$ has 14 primary fields $V^f_{\langle r,s \rangle}$ with $r = 1, 2, \ldots, 7$ and $s = 1, 2$. The non-diagonal model $\text{DMM}_{8,5}$ also has 14 primary fields: the 8 diagonal fields $V^f_{\langle 1,1 \rangle}, V^f_{\langle 3,1 \rangle}, V^f_{\langle 5,1 \rangle}, V^f_{\langle 7,1 \rangle}, V^f_{\langle 1,2 \rangle}, V^f_{\langle 3,2 \rangle}, V^f_{\langle 5,2 \rangle}, V^f_{\langle 7,2 \rangle}$, and 6 non-diagonal fields:

| Field | $V_{(0,\frac{1}{2})}$ | $V_{(0,\frac{3}{2})}$ | $V_{(2,-\frac{3}{2})}$ | $V_{(2,-\frac{1}{2})}$ | $V_{(2,\frac{1}{2})}$ | $V_{(2,\frac{3}{2})}$ |
|---|---|---|---|---|---|---|
| Indices | $\langle 4,2 \rangle \langle 4,2 \rangle$ | $\langle 4,1 \rangle \langle 4,1 \rangle$ | $\langle 6,1 \rangle \langle 2,1 \rangle$ | $\langle 6,2 \rangle \langle 2,2 \rangle$ | $\langle 2,2 \rangle \langle 6,2 \rangle$ | $\langle 2,1 \rangle \langle 6,1 \rangle$ |

$$(2.58)$$

### 2.3.2 Generalized D-series minimal models

Let us consider a limit of D-series minimal models where $\frac{q}{p} \to \beta^2 > 0$. There is a natural way of taking the limit of the non-diagonal sector, which is to assume that non-diagonal fields $V_{(r,s)}$ have fixed indices:

$$\lim_{p,q \to \infty} \mathcal{S}^{\text{DMM}_{p,q}}\big|_{\text{non-diagonal}} = \frac{1}{2} \left\{ V_{(r,s)} \right\}_{(r,s) \in 2\mathbb{Z} \times (\mathbb{Z}+\frac{1}{2})} . \tag{2.59}$$

The diagonal sector only differs from the spectrum $\mathcal{S}^{\text{AMM}_{p,q}}$ (2.32) by the restriction $r \in 2\mathbb{N} + 1$. As we saw in Section 2.2.4, there are several possible limits of A-series minimal models, including Liouville theory and generalized minimal models, and this also applies to our diagonal sector $\mathcal{S}^{\text{DMM}_{p,q}}\big|_{\text{diagonal}}$. However, we will now lift this ambiguity by deriving the diagonal sector from the non-diagonal sector, using the fusion rule (2.52b). At first sight, this fusion rule has the limit

$$\lim_{p,q \to \infty} V_{(r_1,s_1)} V_{(r_2,s_2)} \overset{?}{\sim} \sum_{r \overset{2}{=} |r_1-r_2|+1}^{\infty} \sum_{s \overset{2}{=} |s_1-s_2|+1}^{\infty} V^d_{\langle r,s \rangle} . \tag{2.60}$$

This is a sum over infinitely many degenerate fields, whose momentums $P_{(r,s)}$ are dense in the real line if $\beta^2 \notin \mathbb{Q}$. The sum is therefore divergent. To obtain a finite limit, we need to compute

$$\lim_{p,q \to \infty} \frac{1}{q} V_{(r_1,s_1)} V_{(r_2,s_2)} \sim \frac{1}{2} \int_{\mathbb{R}} dP \, V_P . \tag{2.61}$$

This shows that in the limit CFT, the OPE of 2 non-diagonal fields is actually

$$V_{(r_1,s_1)} V_{(r_2,s_2)} \sim \frac{1}{2} \int_{\mathbb{R}+i\epsilon} dP \, V_P , \tag{2.62}$$

where the imaginary shift $i\epsilon$ is here for avoiding poles of OPE coefficients, just like in $c \le 1$ Liouville theory (2.18). This leads to the spectrum

$$\boxed{\mathcal{S}^{\text{GDMM}_{\beta^2}} = \frac{1}{2} \left\{ V_P \right\}_{P \in \mathbb{R}} \bigcup \frac{1}{2} \left\{ V_{(r,s)} \right\}_{(r,s) \in 2\mathbb{Z} \times (\mathbb{Z}+\frac{1}{2})}} , \tag{2.63}$$

and we call the corresponding CFT a **generalized D-series minimal model**. Due to the lack of $r \leftrightarrow s$ symmetry in the non-diagonal sector, this theory depends on $\beta^2$ rather than the central charge, i.e. it is not invariant under $\beta \to \beta^{-1}$. The OPEs of this CFT are determined by the conservation of diagonality: in addition to the OPEs of 2 non-diagonal fields (2.62) and the OPE of 2 diagonal fields (2.17), we have the mixed OPE

$$V_{(r_1,s_1)} V_P \sim \frac{1}{2} \sum_{r \in 2\mathbb{Z}} \sum_{s \in \mathbb{Z}+\frac{1}{2}} V_{(r,s)} . \tag{2.64}$$

Like Liouville theory, generalized D-series minimal models cannot be analytically continued beyond $\beta^2 \in \mathbb{R}_{>0}$. However, we can define these models by their spectrums in the range $\Re\beta^2 > 0$ (2.14), assuming that they exist. There is indeed evidence that they exist [23], although they have not been fully solved, and the OPEs (2.62) may include discrete terms in addition to the integral [24].

## 2.4 Loop CFTs

Critical loop models were originally defined as critical limits of statistical models that describe non-intersecting loops in a 2d space. Starting with the Ising model, many statistical models can be reformulated in terms of loops, or directly constructed from loops. But there is a lot of choice about which observables to include in the model. In the Ising model, if we consider only local observables, we obtain the minimal model $\mathrm{AMM}_{4,3}$ in the critical limit. If we also consider non-local observables such as cluster connectivities, we obtain a much richer CFT. For applications to statistical physics, what matters is less to define a consistent CFT than to compute correlation functions, which are then interpreted in terms of probabilities. We will sketch a few consistent CFTs in Section 2.4.3, but our main focus will be on correlation functions, which will be the subject of Section 5.

To begin with, we will now propose a simple extended spectrum that will include all the known loop CFTs, based on the assumption that there exists 1 degenerate field.

### 2.4.1 Extended spectrum

Let the extended spectrum $\widehat{\mathcal{S}}^{\mathrm{loop}}$ be made of all primary fields with half-integer spins that are compatible with the degenerate field $V^d_{\langle 1,2\rangle}$:

$$\boxed{\widehat{\mathcal{S}}^{\mathrm{loop}} = \left\{V^d_{\langle 1,s\rangle}\right\}_{s\in\mathbb{N}^*} \bigcup \left\{V_{(r,s)}\right\}_{\substack{r\in\frac{1}{2}\mathbb{N}^* \\ s\in\frac{1}{2r}\mathbb{Z}}} \bigcup \left\{V_P\right\}_{P\in\mathbb{C}}}. \tag{2.65}$$

We include all degenerate fields that are generated from $V^d_{\langle 1,2\rangle}$ by fusion, and all possible diagonal fields. When it comes to non-diagonal fields, the first index $r$ must be half-integer according to Eq. (2.11a). The reflection relation $V_{(r,s)} = V_{(-r,-s)}$ (2.5) allows us to assume $r \geq 0$, and the sector $r = 0$ is redundant with the diagonal sector, since

$$\boxed{V_{(0,s)} = V_{\frac{1}{2}\beta^{-1}s} \quad \text{i.e.} \quad V_P = V_{(0,2\beta P)}}. \tag{2.66}$$

On the other hand, $V_{(r,0)}$ is a genuinely non-diagonal field, whose fusion with $V^d_{\langle 1,2\rangle}$ differs from that of the diagonal field with the same dimensions $V_{\frac{1}{2}\beta r}$. Non-diagonal fields with integer spins include:

$$V_{(\frac{1}{2},0)}, V_{(\frac{1}{2},\pm2)}, V_{(\frac{1}{2},\pm4)}, \cdots \tag{2.67a}$$

$$V_{(1,0)}, V_{(1,\pm1)}, V_{(1,\pm2)}, \cdots \tag{2.67b}$$

$$V_{(\frac{3}{2},0)}, V_{(\frac{3}{2},\pm\frac{2}{3})}, V_{(\frac{3}{2},\pm\frac{4}{3})}, V_{(\frac{3}{2},\pm2)}, \cdots \tag{2.67c}$$

$$V_{(2,0)}, V_{(2,\pm\frac{1}{2})}, V_{(2,\pm1)}, V_{(2,\pm\frac{3}{2})}, \cdots \tag{2.67d}$$

Our simple extended spectrum hides important subtleties:

- We characterize fields by their conformal dimensions, but there could be several independent fields with the same dimensions, in other words there could be non-trivial field multiplicities. This is known to occur in the loop CFTs of Section 2.4.3, where the multiplicities are described in terms of representations of global symmetry groups.

- Knowing which primary fields belong to $\widehat{S}^{\text{loop}}$ is not enough for determining how the conformal algebra acts on $\widehat{S}^{\text{loop}}$. For $(r, s) \in \mathbb{N}^* \times \mathbb{Z}^*$, the primary field $V_{(r,s)}$ has a null vector, and the Verma module $\mathcal{V}_{\Delta_{(r,s)}} \otimes \bar{\mathcal{V}}_{\Delta_{(r,-s)}}$ is not the only representation that includes $V_{(r,s)}$. The correct representation can be determined by starting with $V_{(r,0)}$ and repeatedly taking OPEs with $V_{\langle 1,2 \rangle}^d$. This leads to a logarithmic inde-composable representation $\mathcal{W}_{(r,|s|)}$ that contains both $V_{(r,s)}$ and $V_{(r,-s)}$, and is not left-right factorized [13]. By convention we label the resulting family $\left( \mathcal{W}_{(r,s)} \right)_{\substack{r \in \mathbb{N}^* \\ s \in \mathbb{N}}}$ with natural integers $s$, where we define $\mathcal{W}_{(r,0)} = \mathcal{V}_{\Delta_{(r,0)}} \otimes \bar{\mathcal{V}}_{\Delta_{(r,0)}}$.

### 2.4.2 Dependence on the central charge and fusion rules

In loop CFTs, the existence of non-diagonal fields together with $V_{\langle 1,2 \rangle}^d$ leads to the half-plane constraint $\Re c < 13$ (2.14) on the central charge. And the asymmetry between the 2 Kac indices $r, s$ means that loop CFTs really depend on $\beta^2$ (1.15), not on $c$.

We now make an assumption that has 3 different formulations:

1. Any OPE spectrum is discrete.

2. Loop CFTs do not include Liouville theory.

3. Correlation functions are meromorphic in $\beta$ over the whole domain $\Re c < 13$.

These fomulations are equivalent because a continous OPE spectrum would produce non-analyticities on the line $c \leq 1$, as we saw in the context of Liouville theory in Section 2.2.2. Admittedly, it is only a conjecture that discreteness of the OPEs implies that correlation functions are meromorphic.

In addition to our assumption that OPEs are discrete, the known constraints on fusion rules are:

- OPEs that involve the degenerate field $V_{\langle 1,2 \rangle}^d$ are given by Eqs. (2.7) and (2.10),

- the first Kac index is conserved modulo integers (2.11a).

### 2.4.3 $O(n)$ CFT, $PSU(n)$ CFT, Potts CFT

Three known models give rise to integer-spin subsets of the extended spectrum $\widehat{S}^{\text{loop}}$. Their spectrums are

$$\mathcal{S}^{O(n)} = \left\{ V_{\langle 1,s \rangle}^d \right\}_{s \in 2\mathbb{N}+1} \bigcup \left\{ V_{(r,s)} \right\}_{\substack{r \in \frac{1}{2}\mathbb{N}^* \\ s \in \frac{1}{r}\mathbb{Z}}} , \tag{2.68}$$

$$\mathcal{S}^{PSU(n)} = \left\{ V_{\langle 1,s \rangle}^d \right\}_{s \in \mathbb{N}^*} \bigcup \left\{ V_{(r,s)} \right\}_{\substack{r \in \mathbb{N}^* \\ s \in \frac{1}{r}\mathbb{Z}}} , \tag{2.69}$$

$$\mathcal{S}^{\text{Potts}} = \left\{ V_{\langle 1,s \rangle}^d \right\}_{s \in \mathbb{N}^*} \bigcup \left\{ V_{(r,s)} \right\}_{\substack{r \in \mathbb{N}+2 \\ s \in \frac{1}{r}\mathbb{Z}}} \bigcup \left\{ V_{P_{(0,s)}} \right\}_{s \in \mathbb{N}+\frac{1}{2}} . \tag{2.70}$$

The spectrums of the $O(n)$ and $PSU(n)$ models are natural constructions: we start with a degenerate field $V^d \in \{ V_{\langle 1,3 \rangle}^d, V_{\langle 1,2 \rangle}^d \}$, and add non-diagonal fields $V_{(r,s)} \in \widehat{S}^{\text{loop}}$ such that $V_{(r,s)}$ and the fields that appear in $V^d V_{(r,s)}$ have integer spins. On the other hand, the Potts spectrum has no simple explanation in our formalism. We can only remark that it is obtained from the $PSU(n)$ spectrum by removing the non-diagonal fields $\left\{ V_{(1,s)} \right\}_{s \in \mathbb{Z}}$ and adding the diagonal fields $\left\{ V_{P_{(0,s)}} \right\}_{s \in \mathbb{N}+\frac{1}{2}}$, see Section 5.1.4 for a justification from statistical physics.

Each one of these CFTs has

- a modular-invariant torus partition function,

- a group of global symmetries that commutes with conformal symmetry: the orthogonal group $O(n)$, the unitary group $PSU(n)$, or the symmetric group $S_Q$ in the case of the Potts CFT. The parameters of these groups can take arbitrary complex values [25, 26]

$$\boxed{n = -2\cos\left(\pi\beta^2\right)} \quad , \quad \boxed{Q = 4\cos^2\left(\pi\beta^2\right)} \ . \tag{2.71}$$

Degenerate fields are invariant under global symmetries, while other fields can transform in non-trivial representations [27, 28].

For example, in the $O(n)$ CFT, the fields $V_{(\frac{1}{2},0)}, V_{(1,0)}$ and $V_{(1,1)}$ transform under $O(n)$ as a vector, a symmetric 2-tensor and an antisymmetric 2-tensor respectively. There are 2 fields of the type $V_{(\frac{3}{2},0)}$, which transform as symmetric and antisymmetric 3-tensors. Let us diplay the behaviour of these fields and a few more, while writing finite-dimensional irreducible representations of $O(n)$ as Young diagrams:

$$V_{(\frac{1}{2},0)} : [1] \ , \tag{2.72a}$$

$$V_{(1,0)} : [2] \ , \tag{2.72b}$$

$$V_{(1,1)} : [11] \ , \tag{2.72c}$$

$$V_{(\frac{3}{2},0)} : [3] + [111] \ , \tag{2.72d}$$

$$V_{(\frac{3}{2},\frac{2}{3})} : [21] \ , \tag{2.72e}$$

$$V_{(2,0)} : [4] + [22] + [211] + [2] + [] \ , \tag{2.72f}$$

$$V_{(2,\frac{1}{2})} : [31] + [211] + [11] \ , \tag{2.72g}$$

$$V_{(2,1)} : [31] + [22] + [1111] + [2] \ , \tag{2.72h}$$

$$V_{(\frac{5}{2},0)} : [5] + [32] + 2[311] + [221] + [11111] + [3] + 2[21] + [111] + [1] \ . \tag{2.72i}$$

In particular, there are 2 fields of type $V_{(\frac{5}{2},0)}$ that transform in the irreducible representation [311], and also 2 fields that transform in [21]. This is a first hint that field multiplicities become larger as the first index of $V_{(r,s)}$ increases.

The situation is similar in the $PSU(n)$ and Potts CFTs: let us only mention that the diagonal non-degenerate fields $V_{P_{(0,s)}}$ of the Potts CFT transform in the standard representation of $S_Q$, whose dimension is $Q - 1$.

## 2.5 Completing the picture

### 2.5.1 Summary of spectrums

For each CFT, we indicate the values of $\beta^2$, of the Kac indices of degenerate fields, of the indices of non-diagonal fields, and of the momentums of diagonal fields:

| CFT | $\beta^2$ | $V^{d/f}_{\langle r,s\rangle}$ | $V_{(r,s)}$ | $V_P$ |
|---|---|---|---|---|
| GMM | $\mathbb{C}\backslash\mathbb{Q}$ | $\mathbb{N}^*\times\mathbb{N}^*$ | – | – |
| Liouville | $\mathbb{C}^*$ | – | – | $\mathbb{R}$ |
| $\widehat{\text{Liouville}}$ | $\mathbb{C}^*$ | $\mathbb{N}^*\times\mathbb{N}^*$ | – | $\mathbb{C}$ |
| AMM | $\frac{q}{p}>0$ | $\mathbb{N}\times\mathbb{N}$ $(0,p)\times(0,q)$ | – | – |
| RWT | $\frac{q}{p}>0$ | – | – | $\mathbb{R}$ |
| DMM | $\frac{q}{p}>0$ $p\in 2\mathbb{N}$ | $(2\mathbb{N}+1)\times\mathbb{N}$ $(0,p)\times(0,q)$ | $2\mathbb{Z}\times(\mathbb{Z}+\frac{1}{2})$ $(-\frac{p}{2},\frac{p}{2})\times(-\frac{q}{2},\frac{q}{2})$ | – |
| GDMM | $\{\Re\beta^2>0\}$ | – | $2\mathbb{Z}\times(\mathbb{Z}+\frac{1}{2})$ | $\mathbb{R}$ |
| $\widehat{\text{Loop}}$ | $\{\Re\beta^2>0\}$ | $\{1\}\times\mathbb{N}^*$ | $\frac{1}{2}\mathbb{N}^*\times\frac{1}{2r}\mathbb{Z}$ | $\mathbb{C}$ |
| $O(n)$ | $\{\Re\beta^2>0\}$ | $\{1\}\times(2\mathbb{N}+1)$ | $\frac{1}{2}\mathbb{N}^*\times\frac{1}{r}\mathbb{Z}$ | – |
| $PSU(n)$ | $\{\Re\beta^2>0\}$ | $\{1\}\times\mathbb{N}^*$ | $\mathbb{N}^*\times\frac{1}{r}\mathbb{Z}$ | – |
| Potts | $\{\Re\beta^2>0\}$ | $\{1\}\times\mathbb{N}^*$ | $(\mathbb{N}+2)\times\frac{1}{r}\mathbb{Z}$ | $\{P_{(0,s)}\}_{s\in\mathbb{N}+\frac{1}{2}}$ |

$$(2.73)$$

In the case of loop models, this picture is most probably not complete. In addition to the extended spectrum of $\widehat{\text{Loop}}$, it is possible to defined a backbone field: a diagonal primary field whose conformal dimension does not have a simple expression in terms of Kac indices [29]. In fact, from its dimension, it looks like this field may not be compatible with the existence of any degenerate field. Moreover, when studying correlation functions, we will see that the degenerate field $V^d_{\langle 2,1\rangle}$, which is absent from our extended spectrum, seems to play a role in the CFT, see Section 5.4.1.

### 2.5.2 Limits of diagonal CFTs

Taking limits played an important role in deriving some of the CFTs of Section 2.2. Let us summarize the limits that relate Liouville theory, (generalized) minimal models, and Runkel–Watts-type theories:

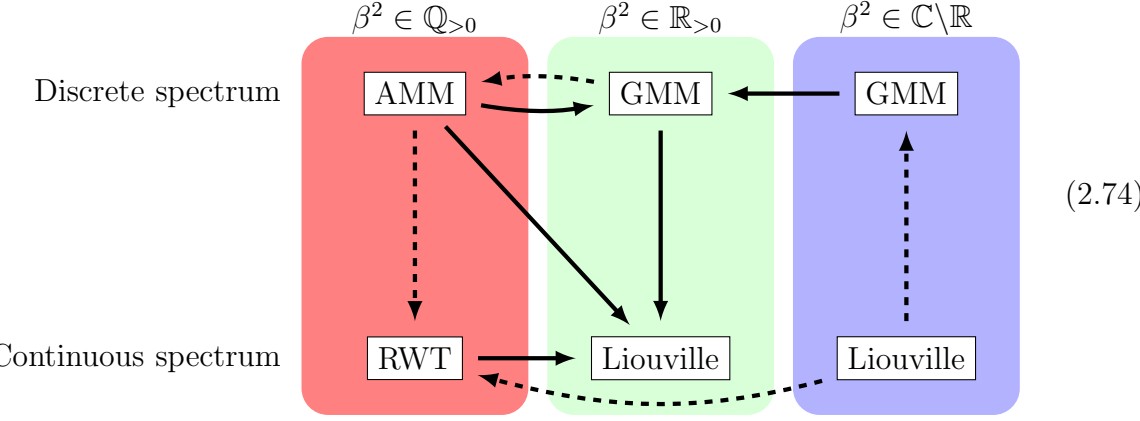

$$(2.74)$$

Somewhat subjectively, we have distinguished limits that are straightforward (full arrows) from limits that are subtle (dashed arrows). The subtle limits are:

- $\boxed{\text{AMM} \to \text{RWT}}$ (Section 2.2.4): This involves fine-tuning the parameters $p, q$ of the minimal models, as well as the parameters $r, s$ of the primary fields.

- $\boxed{\text{Liouville} \to \text{RWT}}$ (Section 2.2.4): This requires a detailed analysis of structure constants.

- $\boxed{\text{GMM} \to \text{AMM}}$ ([18]): This requires a detailed analysis of correlation functions, and only some of the GMM correlation functions tend to AMM correlation functions.

- $\boxed{\text{Liouville} \to \text{GMM}}$ (Section 3.3.1): This requires a detailed analysis of correlation functions.

### 2.5.3 Other exactly solvable CFTs

Let us mention a few other solvable CFTs:

- **E-series minimal models** exist for $\beta^2 = \frac{q}{p}$ with $p = 12, 18, 30$ [1]. They have finitely many primary fields, which belong to fully degenerate representations from the Kac table. They have 2 independent degenerate fields $\left\{ V^f_{\langle r,1 \rangle}, V^f_{\langle 1,2 \rangle} \right\}$ with $r = 4, 5, 7$. However, we do not know how to derive their known spectrums from the study of fusion rules. Solving E-series minimal models is apparently still an open problem.

- **Logarithmic minimal models** are believed to exist for $\beta^2 \in \mathbb{Q}_{>0}$ [30]. Their primary fields belong to the extended Kac table, i.e. they have integer Kac indices. Their spectrums include logarithmic representations. Their spectrums and fusion rules are not fully known. It is in fact not clear how many such CFTs exist at any given central charge. Presumably, they can be obtained as limits of generalized minimal models.

- At $c = 1$, the **compactified free boson** and its $\mathbb{Z}_2$ orbifold the **Ashkin–Teller model** are better understood in terms of an extended symmetry algebra (an abelian affine Lie algebra), although they can also be solved in terms of Virasoro symmetry only [31]. They have a degenerate field $V^f_{\langle 1,3 \rangle} = V^f_{\langle 3,1 \rangle}$, which however behaves as the identity field under fusion with non-diagonal fields. As a result, the constraints (2.11a) are not obeyed. These models have a continuous parameter: the compactification radius $R \in \mathbb{C}^*$. For a special value of $R$, the compactified free boson is a $c \to 1$ limit of the $O(n)$ CFT, and the Ashkin–Teller model is a $c \to 1$ limit of the Potts CFT. Compactified free bosons also exist for $c \in \mathbb{C}$, but then $R$ is quantized [7].

## 3 Analytic bootstrap

In Section 1.2.3, we have worked out how local conformal symmetry constrains operator product expansions. This was done by using the associativity of operator products of the type $TVV$, where $T$ is the energy-momentum tensor. Similarly, we will now study how degenerate fields constrain structure constants, by using the associativity of operator products of the type $V^d_{\langle 2,1 \rangle} VV$ or $V^d_{\langle 1,2 \rangle} VV$. This will determine how structure constants behave under momentum shifts of the type $P \to P + \beta$ or $P \to P + \beta^{-1}$.

## 3.1 Degenerate 4-point functions

Associativity of OPEs is equivalent to crossing symmetry of 4-point functions, so let us consider a 4-point function that involves a degenerate field. Since fusion products of degenerate representations (1.65) are finite, our 4-point function is a combination of finitely many conformal blocks in any one of the 3 channels $s, t, u$. This makes crossing symmetry equations particularly simple, and allows them to be solved analytically. And since any degenerate field $V_{\langle r,s \rangle}^d$ can be obtained from the 2 basic degenerate fields $V_{\langle 2,1 \rangle}^d$ and $V_{\langle 1,2 \rangle}^d$ by fusion, the crossing symmetry equations for $\left\langle V_{\langle r,s \rangle}^d V_1 V_2 V_3 \right\rangle$ follow from the 2 cases $(r,s) = (2,1)$ and $(r,s) = (1,2)$, which are related to one another by $\beta \to \beta^{-1}$.

### 3.1.1 BPZ equations

As we have seen in Section 1.3, for any creation operator $L \in \mathcal{L}$, local conformal symmetry determines descendant correlation functions $\langle L V_0 \cdots \rangle$ in terms of the corresponding primary correlation functions $\langle V_0 \cdots \rangle$. In the case of 4-point functions, we have

$$\left\langle L V_0(z) V_1(0) V_2(\infty) V_3(1) \right\rangle = P_L(z, \partial_z) \left\langle V_0(z) V_1(0) V_2(\infty) V_3(1) \right\rangle , \tag{3.1}$$

where $P_L(z, \partial_z)$ is a differential polynomial whose degree in $\partial_z$ is the level $|L|$ of the descendant field. If now our field is degenerate $V_0 = V_{\langle r,s \rangle}^d$, then the null vector equation $L_{\langle r,s \rangle} V_{\langle r,s \rangle}^d = 0$ (1.26) leads to a differential equation of order $rs$,

$$P_{L_{\langle r,s \rangle}}(z, \partial_z) \left\langle V_{\langle r,s \rangle}^d(z) V_1(0) V_2(\infty) V_3(1) \right\rangle = 0 . \tag{3.2}$$

This is called a Belavin–Polyakov–Zamolodchikov equation or **BPZ equation**. Let us derive the BPZ equation for a 4-point function of the degenerate field $V_{\langle 2,1 \rangle}$, together with 3 other primary fields:

$$G(z) = \left\langle V_{\langle 2,1 \rangle}^d(z) V_1(0) V_2(\infty) V_3(1) \right\rangle . \tag{3.3}$$

The relevant creation operator is $L_{\langle 2,1 \rangle} = L_{-1}^2 - \beta^2 L_{-2}$ (1.16), and the insertion of $L_{-2}$ is determined by the local Ward identity (1.78). However, this identity involves derivatives in the positions of $V_i$, which are now fixed to $0, \infty, 1$. We can eliminate these derivatives using global Ward identities. A technical trick to do this is to rewrite the local Ward identity as $G_\infty(z) = G_z(z) + G_0(z) + G_1(z)$, where we define

$$G_{z_0}(z) = \frac{1}{2\pi i} \oint_{z_0} \frac{y(y-1)}{y-z} \left\langle T(y) V_{\langle 2,1 \rangle}(z) V_1(0) V_2(\infty) V_3(1) \right\rangle . \tag{3.4}$$

We then compute $G_{z_0}(z)$ using the OPE $T(y) V_\Delta(z_0)$ (1.39). The case $T(y) V_\Delta(\infty) = \frac{\Delta}{y^2} V_\Delta(\infty) + O(\frac{1}{y})$ is obtained from the case $z_0 \in \mathbb{C}$ using the global conformal transformation $z \to \frac{1}{z}$ and the definition (1.74) of $V_\Delta(\infty)$. We find

$$G_\infty(z) = \Delta_2 G(z) \quad , \quad G_0(z) = \frac{\Delta_1}{z} G(z) \quad , \quad G_1(z) = \frac{\Delta_3}{1-z} G(z) , \tag{3.5a}$$

$$G_z(z) = \left( \Delta_{(2,1)} + (2z-1)\partial_z \right) G(z) + z(z-1) \left\langle L_{-2} V_{\langle 2,1 \rangle}(z) V_1(0) V_2(\infty) V_3(1) \right\rangle . \tag{3.5b}$$

Combining our rewritten local Ward identity with the null vector equation $L_{\langle 2,1 \rangle} V_{\langle 2,1 \rangle}^d = 0$, and using $L_{-1}^2 V_{\langle 2,1 \rangle}^d(z) = \partial_z^2 V_{\langle 2,1 \rangle}^d(z)$ (1.33), we obtain the BPZ equation

$$\left( \frac{z(z-1)}{\beta^2} \frac{\partial^2}{\partial z^2} + (2z-1)\frac{\partial}{\partial z} + \frac{\Delta_1}{z} + \Delta_{(2,1)} - \Delta_2 + \frac{\Delta_3}{1-z} \right) G(z) = 0 . \tag{3.6}$$

### 3.1.2 Hypergeometric conformal blocks

According to the degenerate fusion rule (1.62), the decomposition of $G(z)$ into $s$-channel left-moving conformal blocks is a combination of 2 blocks with momentums $P_1 \pm \frac{\beta}{2}$. The situation is similar in the $t$- and $u$-channels, and we write the resulting blocks as $\mathcal{F}_\pm^{(x)}(z)$ with $x \in \{s, t, u\}$:

$$
\mathcal{F}_\pm^{(s)} = \begin{array}{c} 1 \\ \langle 2, 1 \rangle \end{array}\!\!\!\!\Big\rangle\!\!\! \begin{array}{c} P_1 \pm \frac{\beta}{2} \end{array}\!\!\!\Big\langle\!\!\!\begin{array}{c} 2 \\ 3 \end{array} \qquad
\mathcal{F}_\pm^{(t)} = \begin{array}{c} 1 \quad 2 \\ P_3 \pm \frac{\beta}{2} \\ \langle 2, 1 \rangle \quad 3 \end{array} \qquad
\mathcal{F}_\pm^{(u)} = \begin{array}{c} 1 \quad 2 \\ P_2 \pm \frac{\beta}{2} \\ \langle 2, 1 \rangle \quad 3 \end{array} \tag{3.7}
$$

The conformal blocks provide 3 bases $\left\{ \mathcal{F}_\pm^{(x)}(z) \right\}_\pm$ of solutions of the BPZ equation. Each solution is characterized by its asymptotic behaviour (1.102) in a limit $z \to z_0$ with $z_0 \in \{0, \infty, 1\}$. For $z_0 = 0, 1$ this behaviour is of the type $z^{\delta_1}$ and $(1 - z)^{\delta_3}$, with exponents that we compute using Eq. (1.17):

$$
\delta_i = \Delta \left( P_i + \tfrac{\beta}{2} \right) - \Delta(P_i) - \Delta_{(2,1)} = \beta P_i + \tfrac{1 - \beta^2}{2} . \tag{3.8}
$$

Introducing $\varphi(z) = z^{\delta_1}(1 - z)^{\delta_3}$, we deduce that the differential equation for $\varphi^{-1}(z)G(z)$ has holomorphic solutions near $z = 0$ and near $z = 1$, namely $\varphi(z)\mathcal{F}_+^{(s)}(z)$ and $\varphi(z)\mathcal{F}_+^{(t)}(z)$ respectively. By a tedious but straightforward calculation, we find that the equation for $\varphi^{-1}(z)G(z)$ is a **hypergeometric differential equation**

$$
\left( z(1 - z)\frac{\partial^2}{\partial z^2} + [C - (A + B + 1)z]\frac{\partial}{\partial z} - AB \right) \varphi^{-1}(z)G(z) = 0 , \tag{3.9}
$$

with the parameters

$$
\{A, B\} = \left\{ \tfrac{1}{2} + \beta(P_1 \pm P_2 + P_3) \right\}_\pm \quad , \quad C = 1 + 2\beta P_1 . \tag{3.10}
$$

This allows us to write our conformal blocks in terms of the **hypergeometric function** $_2F_1$,

$$
\mathcal{F}_+^{(s)}(z) = z^{\beta P_1 + \frac{1-\beta^2}{2}}(1 - z)^{\beta P_3 + \frac{1-\beta^2}{2}} {}_2F_1\left( \tfrac{1}{2} + \beta P_{123}, \tfrac{1}{2} + \beta P_{13}^2, 1 + 2\beta P_1, z \right) , \tag{3.11a}
$$

$$
\mathcal{F}_+^{(t)}(z) = z^{\beta P_1 + \frac{1-\beta^2}{2}}(1 - z)^{\beta P_3 + \frac{1-\beta^2}{2}} {}_2F_1\left( \tfrac{1}{2} + \beta P_{123}, \tfrac{1}{2} + \beta P_{13}^2, 1 + 2\beta P_3, 1 - z \right) , \tag{3.11b}
$$

$$
\mathcal{F}_+^{(u)}(z) = z^{-\beta P_{23} - \frac{\beta^2}{2}}(z - 1)^{\beta P_3 + \frac{1-\beta^2}{2}} {}_2F_1\left( \tfrac{1}{2} + \beta P_{123}, \tfrac{1}{2} + \beta P_{23}^1, 1 + 2\beta P_2, \tfrac{1}{z} \right) , \tag{3.11c}
$$

where we used the notations $P_{123} = P_1 + P_2 + P_3$ and $P_{12}^3 = P_1 + P_2 - P_3$. The block $\mathcal{F}_-^{(x)}$ is obtained from $\mathcal{F}_+^{(x)}$ by $P_i \to -P_i$ where $i = 1, 3, 2$ for $x = s, t, u$. Let us write the change of basis that relates the $s$- and $t$-channel conformal blocks as

$$
\mathcal{F}_{\epsilon_1}^{(s)} = \sum_{\epsilon_3 \pm} F_{\epsilon_1, \epsilon_3} \mathcal{F}_{\epsilon_3}^{(t)} , \tag{3.12}
$$

where the size 2 matrix $(F_{\epsilon_1, \epsilon_3})_{\epsilon_1, \epsilon_3 = \pm}$ is a **degenerate fusing matrix**. This matrix has the coefficients

$$
\boxed{ F_{\epsilon_1, \epsilon_3} = \frac{\Gamma(1 + 2\beta\epsilon_1 P_1)\Gamma(-2\beta\epsilon_3 P_3)}{\prod_\pm \Gamma(\tfrac{1}{2} + \beta\epsilon_1 P_1 \pm \beta P_2 - \beta\epsilon_3 P_3)} } , \tag{3.13}
$$

and the determinant

$$\det F = -\frac{P_1}{P_3} \ . \tag{3.14}$$

The degenerate fusing matrix is built from Euler's **Gamma function** $\Gamma(x)$: a meromorphic function of $x \in \mathbb{C}$ with simple poles for $x \in -\mathbb{N}$ and no zeros, which obeys $\Gamma(x+1) = x\Gamma(x)$ and $\Gamma(x)\Gamma(1-x) = \frac{\pi}{\sin(\pi x)}$.

### 3.1.3 Crossing symmetry

We will now assemble hypergeometric conformal blocks into crossing-symmetric 4-point functions. We assume that fusion with $V_{\langle 2,1 \rangle}^d$ acts diagonally on $V_1, V_2, V_3$, i.e. the left and right momentums $P_i, \bar{P}_i$ are shifted by the same amount $\frac{\beta}{2}$ or $-\frac{\beta}{2}$. (The case where they are shifted by opposite amounts is equivalent, and can be obtained by $\bar{P}_i \to -\bar{P}_i$.) This assumption is fulfilled by diagonal fields $V_P$, but also by non-diagonal fields $V_{(r,s)}$ with momentums of the type $(P, \bar{P}) = (P_{(r,s)}, P_{(r,-s)})$. Then the degenerate 4-point function (3.3) decomposes into degenerate $x$-channel conformal blocks as

$$\left\langle V_{\langle 2,1 \rangle}^d V_1 V_2 V_3 \right\rangle = \sum_{\pm} d_{\pm}^{(x)} \left| \mathcal{F}_{\pm}^{(x)} \right|^2 \ , \tag{3.15}$$

where we use the modulus squared notation (1.86). Crossing symmetry is the equality of the 3 decompositions $x = s, t, u$, and leads to relations between the degenerate 4-point structure constants $d_{\pm}^{(x)}$. In particular, applying the change of basis (3.12) to the $s$-channel decomposition, and comparing with the $t$-channel decomposition, we obtain

$$\forall \epsilon, \bar{\epsilon} \in \{+, -\} \ , \qquad \sum_{\pm} d_{\pm}^{(s)} F_{\pm,\epsilon} \bar{F}_{\pm,\bar{\epsilon}} = \delta_{\epsilon,\bar{\epsilon}} d_{\epsilon}^{(t)} \ . \tag{3.16}$$

The 2 cases $\epsilon = -\bar{\epsilon}$ give us 2 linear relations between $d_+^{(s)}$ and $d_-^{(s)}$, whose compatibility amounts to

$$\frac{F_{++}F_{--}}{F_{+-}F_{-+}} = \frac{\bar{F}_{++}\bar{F}_{--}}{\bar{F}_{+-}\bar{F}_{-+}} \ . \tag{3.17}$$

With our degenerate fusing matrix (3.13), we find

$$\frac{F_{++}F_{--}}{F_{+-}F_{-+}} = \prod_{\pm} \frac{\cos \pi\beta(P_1 \pm P_2 - P_3)}{\cos \pi\beta(P_1 \pm P_2 + P_3)} \ . \tag{3.18}$$

Thanks to the conservation of the second Kac index $s = \beta(\bar{P} - P)$ modulo integers (2.11b), this expression satisfies the compatibility condition (3.17). Then the crossing symmetry relations (3.16) determine all the structure constants $d_{\pm}^{(s)}, d_{\pm}^{(t)}$ in terms of one of them,

$$\frac{d_-^{(s)}}{d_+^{(s)}} = -\frac{F_{++}\bar{F}_{+-}}{F_{-+}\bar{F}_{--}} \quad , \quad \frac{d_+^{(t)}}{d_+^{(s)}} = \frac{F_{++}}{\bar{F}_{--}}\det \bar{F} \quad , \quad \frac{d_-^{(t)}}{d_+^{(s)}} = -\frac{\bar{F}_{+-}}{F_{-+}}\det F \ . \tag{3.19}$$

Explicitly, these relations read

$$\frac{d_-^{(s)}}{d_+^{(s)}} = -\frac{\Gamma(1 + 2\beta P_1)}{\Gamma(1 - 2\beta P_1)} \frac{\Gamma(1 + 2\beta \bar{P}_1)}{\Gamma(1 - 2\beta \bar{P}_1)} (-)^{2s_2} \frac{\prod_{\pm,\pm} \Gamma\left(\frac{1}{2} - \beta\bar{P}_1 \pm \beta\bar{P}_2 \pm \beta\bar{P}_3\right)}{\prod_{\pm,\pm} \Gamma\left(\frac{1}{2} + \beta P_1 \pm \beta P_2 \pm \beta P_3\right)} \ , \tag{3.20a}$$

$$\frac{d_+^{(t)}}{d_+^{(s)}} = \frac{\Gamma\left(1+2\beta P_1\right)}{\Gamma\left(-2\beta\bar{P}_1\right)} \frac{\Gamma\left(-2\beta P_3\right)}{\Gamma\left(1+2\beta\bar{P}_3\right)} \frac{\prod_\pm \Gamma\left(\frac{1}{2}-\beta\bar{P}_1 \pm \beta\bar{P}_2 + \beta\bar{P}_3\right)}{\prod_\pm \Gamma\left(\frac{1}{2}+\beta P_1 \pm \beta P_2 - \beta P_3\right)} \ , \tag{3.20b}$$

$$\frac{d_-^{(t)}}{d_+^{(s)}} = \frac{\Gamma\left(1+2\beta\bar{P}_1\right)}{\Gamma\left(-2\beta P_1\right)} \frac{\Gamma\left(2\beta\bar{P}_3\right)}{\Gamma\left(1-2\beta P_3\right)} \frac{\prod_\pm \Gamma\left(\frac{1}{2}-\beta P_1 \pm \beta P_2 - \beta P_3\right)}{\prod_\pm \Gamma\left(\frac{1}{2}+\beta\bar{P}_1 \pm \beta\bar{P}_2 + \beta\bar{P}_3\right)} \ . \tag{3.20c}$$

We will also need Eq. (3.20a) in the special case $\left\langle V_{\langle 2,1\rangle}^d V_1 V_{\langle 2,1\rangle}^d V_1 \right\rangle$, where 2 of the fields are degenerate and the remaining 2 are equal:

$$\frac{d_-^{(s)}}{d_+^{(s)}} = -\frac{\Gamma\left(1+2\beta P_1\right)}{\Gamma\left(1-2\beta P_1\right)} \frac{\Gamma\left(1+2\beta\bar{P}_1\right)}{\Gamma\left(1-2\beta\bar{P}_1\right)} \frac{\Gamma\left(\beta^2-2\beta P_1\right)}{\Gamma\left(\beta^2+2\beta\bar{P}_1\right)} \frac{\Gamma\left(1-\beta^2-2\beta P_1\right)}{\Gamma\left(1-\beta^2+2\beta\bar{P}_1\right)} \ . \tag{3.21}$$

Therefore, crossing symmetry allows us to determine degenerate 4-point functions up to an overall constant factor, by imposing relations on 4-point structure constants. Next we will deduce shift equations for the 2-point and 3-point structure constants.

## 3.2 Shift equations for structure constants

We will now derive shift equations for 3-point structure constants $C_{ijk}$. When it comes to 2-point structure constants, shift equations and their solutions can be deduced using Eq. (1.93).

### 3.2.1 Degenerate OPE structure constants

Let us introduce **degenerate OPE structure constants** $c_i^\pm$ such that

$$V_{\langle 2,1\rangle}^d V_i \sim c_i^+ V_{i^+} + c_i^- V_{i^-} \ , \tag{3.22}$$

where we use the notation $i^\pm$ for the labels of the primary fields that appear in the degenerate OPE. The reflection relation (2.5) determines $c_i^-$ from $c_i^+$:

$$c_i^- = c_{-i}^+ \ , \tag{3.23}$$

where we use the notation $-i$ for the label of a reflected field. With these notations, the degenerate 4-point structure constants are:

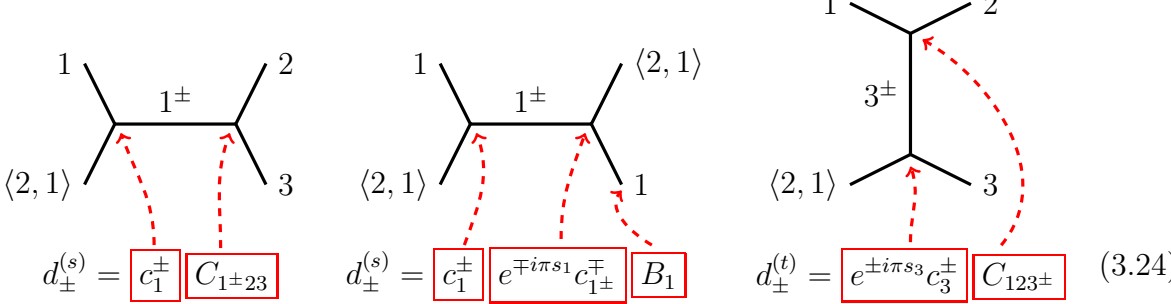

Notice that the OPEs $V_{\langle 2,1\rangle}^d V_i$ and $V_i V_{\langle 2,1\rangle}^d$ are related by a permutation, under which the OPE structure constant picks up a factor $e^{i\pi(S_i - S_{i^\pm})} = e^{i\pi s_i}$, using Eq. (2.9a). This explains the factors $e^{\mp i\pi s_1}$ and $e^{\pm i\pi s_3}$, which we do not write as $(-)^{s_i}$ in order to allow $s_i \in \frac{1}{2}\mathbb{Z}$ and not just $s_i \in \mathbb{Z}$. In the case of the 4-point function $\left\langle V_{\langle 2,1\rangle}^d V_i V_{\langle 2,1\rangle}^d V_i \right\rangle$, we deduce the ratio

$$\frac{d_-^{(s)}}{d_+^{(s)}} = (-)^{2s_i} \frac{c_i^-}{c_{i^+}^-} \frac{c_{i^-}^+}{c_i^+} \ . \tag{3.25}$$

Since its left-hand side is explicitly known by Eq. (3.21), this equation is a constraint on the degenerate OPE structure constants. It can be checked that the following expressions solves this constraint, and also obey the reflection relation (3.23):

$$c_i^+ = \beta^{2\beta(P_i+\bar{P}_i)}\frac{\Gamma(-2\beta P_i)}{\Gamma(1+2\beta\bar{P}_i)} \quad , \quad c_i^- = \beta^{-2\beta(P_i+\bar{P}_i)}\frac{\Gamma(2\beta P_i)}{\Gamma(1-2\beta\bar{P}_i)} \ , \tag{3.26}$$

This solution is far from unique, because in the bootstrap approach we are always free to perform **field renormalizations** using arbitrary constant coefficients $\lambda_i$,

$$V_i(z) \to \lambda_i V_i(z) \quad \implies \quad c_i^\pm \to \lambda_{\langle 2,1\rangle}\frac{\lambda_i}{\lambda_{i^\pm}}c_i^\pm \ , \ B_i \to \lambda_i^2 B_i \ , \ C_{ijk} \to \lambda_i\lambda_j\lambda_k C_{ijk} \ . \tag{3.27}$$

Instead of choosing a solution for $c_i^\pm$, and therefore a particular field normalization, we could work with renormalization-invariant quantities, which cannot involve $c_i^\pm$. For example, the $s$-channel 4-point structure constant $D_k^{(s)} = \frac{C_{12k}C_{k34}}{B_k}$ is invariant under renormalization of the channel field $V_k$. Using the identity $c_i^- B_{i^-} = (-)^{s_i}c_{i^-}^+ B_i$, which follows from Eqs. (1.90) and (1.96), it is possible to write $D_{k^+}^{(s)}$ as a combination of degenerate 4-point structure constants (3.24).

In a CFT with 2 degenerate fields, the associativity of OPEs of the type $V_{\langle 2,1\rangle}^d V_{\langle 1,2\rangle}^d V_i$ leads to relations between the degenerate OPE coefficients for the 2 degenerate fields. (See [7, Exercise 3.2] for the case of Liouville theory.) These relations are equivalent to the compatibility of the shift equations from $V_{\langle 2,1\rangle}^d$ and $V_{\langle 1,2\rangle}^d$ for $B_i, C_{ijk}$. We will show their compatibility by finding solutions.

### 3.2.2 Explicit shift equations

Combining the crossing symmetry equation (3.20a), and (3.21) with the determination (3.24) of degenerate 4pt structure constants and the choice (3.26) of degenerate OPE structure constants, we obtain

$$\boxed{\frac{C_{1-23}}{C_{1+23}} = (-)^{2s_2}\beta^{4\beta^2 r_1}\frac{\prod_{\pm,\pm}\Gamma\left(\frac{1}{2}-\beta\bar{P}_1\pm\beta\bar{P}_2\pm\beta\bar{P}_3\right)}{\prod_{\pm,\pm}\Gamma\left(\frac{1}{2}+\beta P_1\pm\beta P_2\pm\beta P_3\right)}} \ , \tag{3.28}$$

This is called a **shift equation**, because it determines how structure constants behave when the momentums $P_1, \bar{P}_1$ of a field are shifted by $\beta$, under the condition $s_i \in \frac{1}{2}\mathbb{Z}$ (2.11b). If $s_i \in \mathbb{Z}$, then not only $S_{i^-} - S_{i^+} \in \mathbb{Z}$ but also $S_i - S_{i^+} \in \mathbb{Z}$, and we also have the more refined shift equation, deduced from Eq. (3.20b),

$$\boxed{\frac{C_{123+}}{C_{1+23}} = (-)^{s_3}\beta^{2\beta^2(r_1-r_3)}\frac{\prod_\pm\Gamma\left(\frac{1}{2}-\beta\bar{P}_1\pm\beta\bar{P}_2+\beta\bar{P}_3\right)}{\prod_\pm\Gamma\left(\frac{1}{2}+\beta P_1\pm\beta P_2-\beta P_3\right)}} \ . \tag{3.29}$$

Our 2 shift equations were associated to the degenerate field $V_{\langle 2,1\rangle}^d$. To obtain the shift equations associated to $V_{\langle 1,2\rangle}^d$ instead, we may perform the substitutions $\beta \to \beta^{-1}$ and $r \leftrightarrow s$, which imply $(P, \bar{P}) \to (-P, \bar{P})$. We obtain

$$\frac{C_{1-23}}{C_{1+23}} = (-)^{2r_2}\beta^{-4\beta^{-2}s_1}\frac{\prod_{\pm,\pm}\Gamma\left(\frac{1}{2}-\beta\bar{P}_1\pm\beta\bar{P}_2\pm\beta\bar{P}_3\right)}{\prod_{\pm,\pm}\Gamma\left(\frac{1}{2}-\beta P_1\pm\beta P_2\pm\beta P_3\right)} \ , \tag{3.30}$$

$$\frac{C_{123+}}{C_{1+23}} = (-)^{s_3}\beta^{-2\beta^2(s_1-s_3)}\frac{\prod_\pm\Gamma\left(\frac{1}{2}-\beta\bar{P}_1\pm\beta\bar{P}_2+\beta\bar{P}_3\right)}{\prod_\pm\Gamma\left(\frac{1}{2}-\beta P_1\pm\beta P_2-\beta P_3\right)} \ . \tag{3.31}$$

## 3.3 Solutions of shift equations

Since shift equations produce Gamma functions, their solutions should be written in terms of Barnes' **double Gamma function** $\Gamma_\beta$, which obeys

$$\frac{\Gamma_\beta(x+\beta)}{\Gamma_\beta(x)} = \sqrt{2\pi}\frac{\beta^{\beta x - \frac{1}{2}}}{\Gamma(\beta x)} \quad , \quad \frac{\Gamma_\beta(x+\beta^{-1})}{\Gamma_\beta(x)} = \sqrt{2\pi}\frac{\beta^{\frac{1}{2}-\beta^{-1}x}}{\Gamma(\beta^{-1}x)} \ . \tag{3.32}$$

The double Gamma function also obeys $\Gamma_\beta(x) = \Gamma_{\beta^{-1}}(x)$. It is a meromorphic function of $x \in \mathbb{C}$, with simple poles for $x \in -\beta\mathbb{N} - \beta^{-1}\mathbb{N}$ and no zeros. It is also a meromorphic function of $\beta \in \mathbb{C}\backslash i\mathbb{R}$. It can be convenient to use the following consequence of Eq. (3.32):

$$\frac{\prod_\pm \Gamma_\beta\left(x\pm(y-\frac{\beta}{2})\right)}{\prod_\pm \Gamma_\beta\left(x\pm(y+\frac{\beta}{2})\right)} = \beta^{-2\beta y}\frac{\Gamma\left(\beta x + \beta y - \frac{1}{2}\beta^2\right)}{\Gamma\left(\beta x - \beta y - \frac{1}{2}\beta^2\right)} \ . \tag{3.33}$$

Using the double Gamma function, we will write solutions of the shift equations for 3-point structure constants. We will deduce solutions for 2-point structure constants using $B_i \propto C_{Iii}$ (1.93), where the shift equations allow an $i$-independent factor.

### 3.3.1 Diagonal CFTs with 2 degenerate fields

Let us start with Liouville theory for $c \in (-\infty, 1]$ i.e. $\beta^2 \in \mathbb{R}_{>0}$. We assume that the 3-point structure constant $C_{P_1,P_2,P_3}$ is a meromorphic function of the momentums, and a continuous function of $\beta$. The shift equations determine $C_{P_1,P_2,P_3}$ modulo a factor that is invariant under $P_i \to P_i + \beta$ and $P_i \to P_i + \beta^{-1}$. If $\beta^2 \notin \mathbb{Q}$, this factor must be constant on the complex $P_i$-plane. By continuity in $\beta$, the structure constant is unique on $c \in (-\infty, 1]$ modulo an unimportant $P_i$-independent factor. The unique solution turns out to be a holomorphic function of $P_i \in \mathbb{C}$ and $\beta \in \mathbb{C}\backslash i\mathbb{R}$. We then deduce $B_{P_1} \propto C_{P_{(1,1)},P_1,P_1}$,

$$\boxed{B_{P_1} = \prod_{\pm,\pm} \Gamma_\beta^{-1}\left(\beta^{\pm 1} \pm 2P_1\right)} \quad , \quad \boxed{C_{P_1,P_2,P_3} = \prod_{\pm,\pm,\pm} \Gamma_\beta^{-1}\left(\frac{\beta+\beta^{-1}}{2} \pm P_1 \pm P_2 \pm P_3\right)} \ . \tag{3.34}$$

The expression for $C_{P_1,P_2,P_3}$ obeys the shift equation (3.28) with $s_i = 0$ and $P_i = \bar{P}_i$. It is invariant under $P_i \to -P_i$, i.e. it obeys the reflection relation (2.5). And it is invariant under $\beta \to \beta^{-1}$, so it also obeys the shift equation (3.30). Remarkably, the 2-point and 3-point functions can be analytically continued beyond the half-line $c \in (-\infty, 1]$, while $N$-point functions with $N \geq 4$ cannot, as follows from our discussion of the OPE in Section 2.2.2.

The expressions (3.34) for 2-point and 3-point structure constants are also valid in generalized minimal models, and in A-series minimal models. In such models, solutions of shift equations are unique, up to a field-independent factor. In the case of $C_{P_{(r_1,s_1)},P_{(r_2,s_2)},P_{(r_3,s_3)}}$, our shift equations (3.28) and (3.29), together with the invariance under permutations of the 3 fields, determine all integer shifts that preserve $r_1 + r_2 + r_3 \bmod 2$. But the fusion rules (1.66) imply that the 3-point structure constant exists only if $r_1 + r_2 + r_3 \equiv 1 \bmod 2$, and can therefore be deduced from $C_{P_{(1,1)},P_{(1,1)},P_{(1,1)}}$ using shift equations. This implies in particular that the double Gamma functions can be replaced with products of Gamma functions. (See [7, Section 3.2.1] for explicit expressions.) This makes the 3-point structure constants well-defined for any $\beta \in \mathbb{C}^*$, including in the regime $\beta \in i\mathbb{R}$ where $\Gamma_\beta$ is not defined.

In (generalized) minimal models, the structure constants are always non-vanishing, even though the function $\Gamma_\beta^{-1}$ has zeros. For the 3-point structure constant, this is again because of the fusion rules. The factor $\Gamma_\beta^{-1}\left(\frac{1+\sum_i \epsilon_i r_i}{2}\beta + \frac{1-\sum_i \epsilon_i s_i}{2}\beta^{-1}\right)$ (with $\epsilon_i \in$

$\{+,-\}$) of $C_{P_{(r_1,s_1)},P_{(r_2,s_2)},P_{(r_3,s_3)}}$ would indeed vanish if the 2 integers $\frac{1+\sum_i \epsilon_i r_i}{2}$ and $\frac{1-\sum_i \epsilon_i s_i}{2}$ were both negative, but the fusion rules imply that the 3 integers $\sum_i \epsilon_i, \sum_i \epsilon_i r_i, \sum_i \epsilon_i s_i$ all have the same (nonzero) sign. Conversely, if the fusion rules are violated, then $C_{P_{(r_1,s_1)},P_{(r_2,s_2)},P_{(r_3,s_3)}}$ vanishes in some cases, but not in some others. This has been a source of puzzlement [32]. In the bootstrap approach, there is no puzzle, because structure constants make sense only provided fusion rules are obeyed.

We have chosen normalizations such that the structure constants (3.34) have simple expressions, and are holomorphic functions of $\beta, P_i$. In (generalized) minimal models, it is common to choose normalizations such that $V_{\langle 1,1 \rangle}^d$ is the identity field, and such that 2-point functions are trivial. Calling $\widetilde{B}_{\langle r_1,s_1 \rangle}, \widetilde{C}_{\langle r_1,s_1 \rangle \langle r_2,s_2 \rangle \langle r_3,s_3 \rangle}$ the corresponding structure constants, these conditions amount to

$$\widetilde{B}_{\langle r_1,s_1 \rangle} = \widetilde{C}_{\langle 1,1 \rangle \langle r_2,s_2 \rangle \langle r_2,s_2 \rangle} = 1 . \tag{3.35}$$

These conditions are fulfilled by

$$\widetilde{C}_{\langle r_1,s_1 \rangle \langle r_2,s_2 \rangle \langle r_3,s_3 \rangle} = \frac{C_{P_{(r_1,s_1)},P_{(r_2,s_2)},P_{(r_3,s_3)}}}{C_{P_{(1,1)},P_{(1,1)},P_{(1,1)}}} \frac{B_{P_{(1,1)}}^{\frac{3}{2}}}{\sqrt{B_{P_{(r_1,s_1)}} B_{P_{(r_2,s_2)}} B_{P_{(r_3,s_3)}}}} . \tag{3.36}$$

For example, in the Ising minimal model at $\beta^2 = \frac{3}{4}$, the nontrivial structure constant is $\widetilde{C}_{\langle 2,1 \rangle \langle 2,1 \rangle \langle 1,2 \rangle} = \frac{1}{2}$. In this case, the double Gamma functions not only combine into Gamma functions, but further simplify into a rational number.

Finally, let us consider Liouville theory with $c \in \mathbb{C} \backslash (-\infty, 1]$. This includes the half-line $c \in [25, \infty)$ i.e. $\beta \in i\mathbb{R}^*$, where the function $\Gamma_\beta$ is not defined. Instead, we should use the function $\hat{\Gamma}_\beta(x) = \frac{1}{\Gamma_{i\beta}(i\beta - ix)}$, which obeys shift equations that are very similar to the equations (3.32) for $\Gamma_\beta(x)$:

$$\frac{\hat{\Gamma}_\beta(x+\beta)}{\hat{\Gamma}_\beta(x)} = \sqrt{2\pi} \frac{(i\beta)^{\beta x - \frac{1}{2}}}{\Gamma(\beta x)} \quad , \quad \frac{\hat{\Gamma}_\beta(x+\beta^{-1})}{\hat{\Gamma}_\beta(x)} = \sqrt{2\pi}(i\beta)^{\frac{1}{2} - \beta^{-1}x} \Gamma(1 - \beta^{-1}x) . \tag{3.37}$$

Since we now have powers of $i\beta$ instead of $\beta$, we need to slightly modify the degenerate OPE structure constants (3.26), for example $c_i^+ = \beta^{4\beta P_i} \frac{\Gamma(-2\beta P_i)}{\Gamma(1+2\beta P_i)} \to (i\beta)^{4\beta P_i} \frac{\Gamma(-2\beta P_i)}{\Gamma(1+2\beta P_i)}$. The shift equation for structure constants (3.28) is similarly modified, i.e. $\beta^y \to (i\beta)^y$. Its solutions $\hat{B}_{P_1}, \hat{C}_{P_1,P_2,P_3}$ are obtained from $B_{P_1}, C_{P_1,P_2,P_3}$ (3.34) by $\Gamma_\beta \to \hat{\Gamma}_\beta$. Explicitly,

$$\boxed{\hat{B}_{P_1} = \prod_\pm \Gamma_b(\pm 2iP_1) \Gamma_b(Q \pm 2iP_1)} \quad , \quad \boxed{\hat{C}_{P_1,P_2,P_3} = \prod_{\pm,\pm,\pm} \Gamma_b\left(\frac{Q}{2} \pm iP_1 \pm iP_2 \pm iP_3\right)} ,$$
$$\tag{3.38}$$

where we use the notations

$$b = i\beta \quad , \quad Q = b + b^{-1} . \tag{3.39}$$

By a field renormalization (3.27), the 2-point and 3-point structure constants become

$$\hat{B}_{P_1}^{\text{DOZZ}} = \frac{\Gamma_b(-2iP_1) \Gamma_b(Q + 2iP_1)}{\Gamma_b(2iP_1) \Gamma_b(Q - 2iP_1)} \quad , \quad \hat{C}_{P_1,P_2,P_3}^{\text{DOZZ}} = \frac{\prod_{\pm,\pm,\pm} \Gamma_b\left(\frac{Q}{2} \pm iP_1 \pm iP_2 \pm iP_3\right)}{\prod_{k=1}^3 \Gamma_b(2iP_k) \Gamma_b(Q - 2iP_k)} ,$$
$$\tag{3.40}$$

which are the expressions found by Dorn–Otto and Zamolodchikov–Zamolodchikov when first solving Liouville theory: in particular, their formula for the 3-point structure constant

is called the **DOZZ formula**. In this normalization, the 2-point structure constant coincides with the reflection coefficient, i.e. it obeys the relations

$$\hat{B}_{P_1}^{\text{DOZZ}} \hat{B}_{-P_1}^{\text{DOZZ}} = 1 \quad , \quad \hat{C}_{P_1,P_2,P_3}^{\text{DOZZ}} = \hat{B}_{P_1}^{\text{DOZZ}} \hat{C}_{-P_1,P_2,P_3}^{\text{DOZZ}} \quad , \quad V_{P_1} = \hat{B}_{P_1}^{\text{DOZZ}} V_{-P_1} . \qquad (3.41)$$

A remarkable property of Liouville theory in the DOZZ normalization is that degenerate fields are limits of non-degenerate fields,

$$\lim_{P \to P_{(r,s)}} V_P = V_{\langle r,s \rangle}^d . \qquad (3.42)$$

This relation is natural in the path integral construction of Liouville theory [14], but not necessarily in the bootstrap approach: it is not true in Liouville theory with $c \in (-\infty, 1]$, and for $c \in \mathbb{C} \backslash (-\infty, 1]$ it requires a particular normalization. It is thanks to this relation that generalized minimal models can be obtained as limits of Liouville theory, as announced in Section 2.5.2. This relation follows from the analytic properties of the structure constants, as we will now show in the case $(r, s) = (1, 1)$. (See [7, Section 3.1.5] for the general case $r, s \in \mathbb{N}^*$.) Since $2iP_{(1,1)} = Q$ and $\Gamma_b(x)$ has a pole at $x = 0$, we have $\hat{C}_{P_{(1,1)},P_2,P_3}^{\text{DOZZ}} = 0$ and it may seem that $\lim_{P_1 \to P_{(1,1)}} V_{P_1} = 0$. However, it is not so easy to take the limit of the OPE

$$V_{P_1} V_{P_2} \sim \frac{1}{2} \int_{\mathbb{R}} dP \; \frac{\hat{C}_{P_1,P_2,P}^{\text{DOZZ}}}{\hat{B}_P^{\text{DOZZ}}} V_P , \qquad (3.43)$$

because of the poles of $\hat{C}_{P_1,P_2,P}^{\text{DOZZ}}$. In particular, each factor $\Gamma_b(\frac{Q}{2} - iP_1 \pm (iP_2 - iP))$ has a pole at $P = P_2 \pm (P_{(1,1)} - P_1)$. In the limit $P_1 \to P_{(1,1)}$, these 2 poles coincide at $P = P_2$, with 1 pole coming from above the integration line and 1 from below. As a result, the limit of the integral is not zero:

$$\lim_{P_1 \to P_{(1,1)}} V_{P_1} V_{P_2} \sim \lim_{P_1 \to P_{(1,1)}} \operatorname*{Res}_{P=P_1+P_2-P_{(1,1)}} \frac{\hat{C}_{P_1,P_2,P}^{\text{DOZZ}}}{\hat{B}_P^{\text{DOZZ}}} V_P = V_{P_2} , \qquad (3.44)$$

which implies Eq. (3.42) in the case $(r, s) = (1, 1)$.

### 3.3.2 Non-diagonal CFTs with 2 degenerate fields

Recall that non-diagonal fields $V_{(r,s)}$ have indices $r \in 2\mathbb{Z}$ and $s \in \mathbb{Z} + \frac{1}{2}$. Due to the conservation of diagonality (2.47), we have either 0 or 2 non-diagonal fields in a non-vanishing 3-point function. Let us begin with the latter case. We introduce the ansatz

$$\boxed{C_{P_1,(r_2,s_2),(r_3,s_3)} = (-)^{\frac{r_3}{2}} \prod_{\pm,\pm} \Gamma_\beta^{-1}\left(\frac{\beta+\beta^{-1}}{2} + P_1 \pm P_2 \pm P_3\right) \Gamma_\beta^{-1}\left(\frac{\beta+\beta^{-1}}{2} - P_1 \pm \bar{P}_2 \pm \bar{P}_3\right),}$$

$$(3.45)$$

whose sign prefactor ensures the correct behaviour under permutations (1.90). Moreover, we have the identity

$$C_{P_1,(r_2,s_2),(r_3,s_3)} = C_{-P_1,(r_2,s_2),(r_3,s_3)} , \qquad (3.46)$$

which follows from the cases $(r, s) = (r_2 \pm r_3, s_2 \pm s_3)$ of the identity

$$r, s \in \mathbb{Z} \implies \frac{\prod_\pm S_\beta\left(\frac{\beta+\beta^{-1}}{2} + P_1 \pm P_{(r,s)}\right)}{\prod_\pm S_\beta\left(\frac{\beta+\beta^{-1}}{2} + P_1 \pm P_{(r,-s)}\right)} = (-)^{rs} , \qquad (3.47)$$

where we introduced the **double Sine function**

$$S_\beta(x) = \frac{\Gamma_\beta(x)}{\Gamma_\beta(\beta + \beta^{-1} - x)} \qquad \Longrightarrow \qquad \frac{S_\beta(x + \beta)}{S_\beta(x)} = 2\sin(\pi\beta x) \ . \qquad (3.48)$$

Having worked out the basic properties of our ansatz (3.45), let us check whether it obeys the shift equations. The spectrum (2.48) of D-series minimal models is invariant under $s \to s + 1$, both in the diagonal and non-diagonal sectors. Therefore, when it comes to the shifts from $V^d_{\langle 1,2 \rangle}$, we have to check the 2 shift equations (3.30) and (3.31). This is straightforward, if we remember $r_2, r_3 \in 2\mathbb{Z}$, leading to trivial sign factors $(-)^{r_i} = 1$. We will now deal with the subtler case of the shifts from $V^d_{\langle 2,1 \rangle}$. The spectrum of D-series minimal models is invariant under $r \to r + 2$ but not $r \to r + 1$, so we only have to deal with the shift equation (3.28). If we shift a non-diagonal field, our ansatz (3.45) behaves as

$$\frac{C_{(r_1-1,s_1),(r_2,s_2),P_3}}{C_{(r_1+1,s_1),(r_2,s_2),P_3}} = -\beta^{4\beta^2 r_1} \frac{\prod_{\pm,\pm} \Gamma(\frac{1}{2} - \beta\bar{P}_1 \pm \beta\bar{P}_2 \pm \beta P_3)}{\prod_{\pm,\pm} \Gamma(\frac{1}{2} + \beta P_1 \pm \beta P_2 \pm \beta P_3)} \ , \qquad (3.49)$$

where the overall sign comes from $\prod_\pm \frac{\cos\pi\beta(P_1 \pm P_2 - P_3)}{\cos\pi\beta(\bar{P}_1 \pm P_2 - P_3)} = \prod_\pm (-1)^{s_1 \pm s_2} = -1$. This agrees with Eq. (3.28) with $\bar{P}_3 = P_3$ and $(-)^{2s_2} = -1$. If we now shift the diagonal field, our ansatz leads to

$$\frac{C_{P_1 - \frac{\beta}{2},(r_2,s_2),(r_3,s_3)}}{C_{P_1 + \frac{\beta}{2},(r_2,s_2),(r_3,s_3)}} = \beta^{8\beta P_1} \frac{\prod_{\pm,\pm} \Gamma(\frac{1}{2} - \beta P_1 \pm \beta\bar{P}_2 \pm \beta\bar{P}_3)}{\prod_{\pm,\pm} \Gamma(\frac{1}{2} + \beta P_1 \pm \beta P_2 \pm \beta P_3)} \ . \qquad (3.50)$$

Now this ratio differs from Eq. (3.28) with $\bar{P}_1 = P_1$ by an overall sign. In the case of D-series minimal models, let us cancel this sign . Since diagonal fields are now degenerate, they can be labelled by Kac indices, and subjected to the field renormalization $V^d_{\langle r,s \rangle} \to (-)^{\frac{r+1}{2}} V^d_{\langle r,s \rangle}$. This renormalization flips the sign of Eq. (3.28), while leaving the other shift equations unchanged. As a result, the ansatz (3.45) is now a solution of the shift equations. On the other hand, $C_{P_{(r_1,s_1)},P_{(r_2,s_2)},P_{(r_3,s_3)}}$ is no longer a solution of the shift equations for the diagonal 3-point structure constant: the solution has an extra sign prefactor,

$$\boxed{C^{\mathrm{DMM}}_{\langle r_1,s_1 \rangle \langle r_2,s_2 \rangle \langle r_3,s_3 \rangle} = (-)^{\frac{r_1 + r_2 + r_3 + 1}{2}} C_{P_{(r_1,s_1)},P_{(r_2,s_2)},P_{(r_3,s_3)}}} \ , \qquad (3.51)$$

where $C_{P_1,P_2,P_3}$ is given in Eq. (3.34).

Now that we know the 3-point structure constants for D-series minimal models, let us deduce generalized D-series minimal models. As argued in Section 2.3.2, this amounts to taking the limit $P_{(r_1,s_1)} \to P_1 \in \mathbb{R}$ in the diagonal sector. This is easily done in the case of $C_{P_1,(r_2,s_2),(r_3,s_3)}$, which is a meromorphic function of $P_1$, and is therefore still valid in the limit. When it comes to the diagonal 3-point structure constant (3.51), the sign prefactor does not have a smooth limit, and the structure constant becomes a distribution in the limit [24],

$$\boxed{C^{\mathrm{GDMM}}_{P_1,P_2,P_3} = \beta C_{P_1,P_2,P_3} \sum_{n \in \mathbb{Z}} (-)^n \frac{\prod_{i=1}^3 \cos(2\pi n\beta P_i)}{\cos(\pi n\beta^2)}} \ . \qquad (3.52)$$

The sum over $n$ is divergent, but it converges when inserted in the $s$-channel decomposition of a correlation function with non-diagonal fields, for example $\langle V_{P_1} V_{P_2} V_{(r_3,s_3)} V_{(s_4,s_4)} \rangle$.

Without the field renormalization $V^d_{\langle r,s \rangle} \to (-)^{\frac{r+1}{2}} V^d_{\langle r,s \rangle}$, would we have obtained a smooth diagonal 3-point structure constant $C_{P_1,P_2,P_3}$, and a distribution in the non-diagonal sector? Of course not: renormalizations cannot make such a difference. To see this, we may study the 4-point function $\langle V_{P_1} V_{P_2} V_{(r_3,s_3)} V_{(r_4,s_4)} \rangle$ in the $s$-channel: whatever the normalization, the 4-point structure constant has a factor $C^{\text{GDMM}}_{P_1,P_2,P_s}$ in the limit. Our field renormalization only makes this clearer at the level of 3-point structure constants.

Sign subtleties do not affect 2-point structure constants. For diagonal fields, the same expression $B_{P_1}$ (3.34) as in $c \leq 1$ Liouville theory is still valid. For non-diagonal fields, we apply the relation $B_{(r,s)} \propto C_{P_{(1,1)},(r,s),(r,s)}$ to our ansatz (3.45), and find

$$
B_{(r,s)} = (-)^{\frac{r}{2}} \prod_{\pm} \Gamma_\beta^{-1} \left( \beta \pm 2P \right) \Gamma_\beta^{-1} \left( \beta^{-1} \pm 2\bar{P} \right) , \tag{3.53}
$$

with $\{P, \bar{P}\} = \{P_{(r,s)}, P_{(r,-s)}\}$. (The formula is invariant under $P \leftrightarrow \bar{P}$.)

### 3.3.3 Loop CFTs

From the structure of the loop CFTs' extended spectrum $\widehat{S}^{\text{loop}}$ (2.65), it is clear that shift equations cannot completely determine the structure constants:

- In the case of non-diagonal fields $V_{(r,s)}$, the dependence on $r$ is not constrained, because the spectrum $\widehat{S}^{\text{loop}}$ does not include the degenerate field $V^d_{\langle 2,1 \rangle}$. We only have $V^d_{\langle 1,2 \rangle}$, whose shift equations determine how structure constants behave under $s \to s + 2$, whereas $s$ takes fractional values.

- In the case of diagonal fields $V_P$, the shift equations from $V^d_{\langle 1,2 \rangle}$ determine how structure constants behave under $P \to P + \beta^{-1}$.

Let us nevertheless look for simple solutions of the shift equations. As an ansatz, we introduce the **reference structure constant**

$$
C^{\text{ref}}_{(r_1,s_1)(r_2,s_2)(r_3,s_3)} = \prod_{\epsilon_1,\epsilon_2,\epsilon_3 = \pm} \Gamma_\beta^{-1} \left( \frac{\beta + \beta^{-1}}{2} + \frac{\beta}{2} \left| \sum_i \epsilon_i r_i \right| + \frac{\beta^{-1}}{2} \sum_i \epsilon_i s_i \right) , \tag{3.54}
$$

which is a product of double Gamma functions of combinations of Kac indices, involving the absolute value of $\sum_i \epsilon_i r_i$. This expression is also supposed to be valid if some of the fields are diagonal, via the identification $V_P = V_{(0,2\beta P)}$ (2.66). If all fields are diagonal, it reduces to $C_{P_1,P_2,P_3}$ (3.34). For reference 2-point structure constants of diagonal fields, we simply set $B^{\text{ref}}_P = B_P$. For non-diagonal fields, we define

$$
B^{\text{ref}}_{(r,s)} = \frac{(-)^{rs}}{2 \sin\left( \pi(\{r\} + s) \right) \sin\left( \pi(r + \beta^{-2} s) \right)} \prod_{\pm,\pm} \Gamma_\beta^{-1} \left( \beta \pm \beta r \pm \beta^{-1} s \right) , \tag{3.55}
$$

where $\{r\} \in \{0, \frac{1}{2}\}$ is the fractional part of $r \in \frac{1}{2}\mathbb{N}^*$. This is formally ill-defined if $r \in \mathbb{N}^*$ and $s \in \mathbb{Z}$, but can be regularized by taking a limit from generic values of $s$. From the relation (1.93) between 2-point and 3-point structure constants, we would have expected $B^{\text{ref}}_{(r,s)}$ to coincide with

$$
C^{\text{ref}}_{(0,2\beta P_{(1,1)})(r,s)(r,s)} = \prod_{\pm} \Gamma_\beta^{-2} \left( \beta^{\pm 1} \right) \prod_{\pm,\pm} \Gamma_\beta^{-1} \left( \beta^{\pm 1} + \beta r \pm \beta^{-1} s \right) . \tag{3.56}
$$

This differs from $B_{(r,s)}^{\text{ref}}$ by a factor that is invariant under shifts $s \to s+2$, except the sign factor $(-)^{rs}$:

$$\frac{B_{(r,s)}^{\text{ref}}}{C_{(0,2\beta P_{(1,1)})(r,s)(r,s)}^{\text{ref}}} = \frac{(-)^{rs}4^r}{2\sin\left(\pi(\{r\}+s)\right)} \prod_{\pm} \Gamma_\beta^2(\beta^{\pm 1}) \prod_{j=1-r}^{r-1} \sin\pi(\beta^2 j + s) . \qquad (3.57)$$

In particular, we have $B_{(0,2\beta P)}^{\text{ref}} = \frac{1}{2\sin(2\pi\beta P)^2} B_P$.

The reference 3-point structure constant only obeys the shift equations (3.30), (3.31) and permutation equation (1.90) up to signs. Let us define the normalized 3-point structure constant

$$C^{\text{norm}} = \frac{C}{C^{\text{ref}}} , \qquad (3.58)$$

then the shift equations amount to [33]

$$\frac{C_{(r_1,s_1+1)(r_2,s_2)(r_3,s_3)}^{\text{norm}}}{C_{(r_1,s_1-1)(r_2,s_2)(r_3,s_3)}^{\text{norm}}} = (-)^{2r_3}(-)^{\max(2r_1,2r_2,2r_3,r_1+r_2+r_3)} , \qquad (3.59)$$

$$\frac{C_{(r_1,s_1+1)(r_2,s_2)(r_3,s_3)}^{\text{norm}}}{C_{(r_1,s_1)(r_2,s_2)(r_3,s_3+1)}^{\text{norm}}} \underset{r_i \in \mathbb{N}^*}{=} \begin{cases} (-)^{r_1+r_2} & \text{if } r_2 \geq |r_1 - r_3| , \\ (-)^{r_3} & \text{else} . \end{cases} \qquad (3.60)$$

Let us focus in particular on diagonal fields. If $r_1 = 0$, then $r_2 \equiv r_3 \bmod \mathbb{Z}$, and Eq. (3.59) reduces to the invariance of $C^{\text{norm}}$ under $s_1 \to s_1 + 2$. As a function of the momentum $P$ of a diagonal field, $C^{\text{norm}}$ is therefore periodic with period $\beta^{-1}$, and it is also even by reflection invariance (2.5). So it is a function of the **loop weight**

$$\boxed{w(P) = 2\cos(2\pi\beta P)} . \qquad (3.61)$$

We conjecture that $C^{\text{norm}}$ is in fact a polynom of loop weights, and that its dependence on the central charge is in fact polynomial in the **contractible loop weight**

$$\boxed{n = w\left(P_{(1,1)}\right) = -2\cos(\pi\beta^2)} . \qquad (3.62)$$

(This is the same $n$ as in the $O(n)$ CFT (2.71).) In Section 5.3.3 we will formulate this conjecture in more detail for 4-point structure constants.

Even if the conjecture only applies to loop CFTs, it is interesting to compute the normalized 3-point structure constant associated to $C_{P_1,(r_2,s_2),(r_3,s_3)}$ (3.45) from (generalized) D-series minimal models:

$$C_{P_1,(r_2,s_2),(r_3,s_3)}^{\text{norm}} = (-)^{\frac{r_3}{2}}(-)^{\frac{s_2+s_3}{2}} \prod_{\pm} \prod_{j=-\frac{|r_2\pm r_3|-1}{2}}^{\frac{|r_2\pm r_3|-1}{2}} 2\cos\pi(\beta P_1 + \beta^2 j) , \qquad (3.63)$$

where the product over $j$ runs by steps of 1. Since $r_2, r_3 \in \mathbb{Z}+\frac{1}{2}$, there is one $j = 0$ factor, which is $2\cos(\pi\beta P_1)$. The rest of the product can be written in terms of

$$4\cos\pi(\beta P_1 + \beta^2 j)\cos\pi(\beta P_1 - \beta^2 j) = 2\cos(2\pi\beta P_1) + 2\cos(2\pi\beta^2 j) , \qquad (3.64)$$

which for $j \in \frac{1}{2}\mathbb{Z}^*$ is polynomial in loop weights.

### 3.3.4 Summary table

Let us summarize the solutions of shift equations for 2-point and 3-point structure constants. For each solution we indicate to which models it applies. Lighter green means an alternative field normalization. In the models' names, "(G)MM" means A-series minimal models and generalized minimal models, and "Liou" means Liouville theory.

| Constants | Eqs. | (G)MM | Liou$_{c\leq 1}$ | RWT | Liou | DMM | GDMM | Loop |
|---|---|---|---|---|---|---|---|---|
| $B_{P_1}$ | (3.34) | green | green | green | | green | green | green |
| $C_{P_1,P_2,P_3}$ | (3.34) | green | green | green | | | | green |
| $\widetilde{C}_{\langle r_1,s_1\rangle\langle r_2,s_2\rangle\langle r_3,s_3\rangle}$ | (3.36) | light green | | | | | | |
| $\hat{B}_{P_1}, \hat{C}_{P_1,P_2,P_3}$ | (3.38) | | | | green | | | |
| $\hat{B}_{P_1}^{\mathrm{DOZZ}}, \hat{C}_{P_1,P_2,P_3}^{\mathrm{DOZZ}}$ | (3.40) | | | | light green | | | |
| $B_{(r,s)}$ | (3.53) | | | | | green | green | |
| $C_{P_1,(r_2,s_2),(r_3,s_3)}$ | (3.45) | | | | | green | green | |
| $C^{\mathrm{DMM}}_{\langle r_1,s_1\rangle\langle r_2,s_2\rangle\langle r_3,s_3\rangle}$ | (3.51) | | | | | green | | |
| $C^{\mathrm{GDMM}}_{P_1,P_2,P_3}$ | (3.52) | | | | | | green | |
| $B^{\mathrm{ref}}_{(r,s)}$ | (3.55) | | | | | | | green |
| $C^{\mathrm{ref}}_{(r_1,s_1)(r_2,s_2)(r_3,s_3)}$ | (3.54) | | | | | | | green |

# 4 Numerical bootstrap

In the analytic bootstrap approach of Section 3, we have used the crossing symmetry of 4-point functions that include degenerate fields, and deduced constraints on 2-point and 3-point structure constants. For 4-point functions without degenerate fields, crossing symmetry equations involve infinite sums and are hard to deal with analytically. We will thus need to solve them numerically.

## 4.1 Approaches to crossing symmetry

### 4.1.1 Achievements and limitations of analytic approaches

It is not easy to directly evaluate the crossing symmetry equations (1.105): conformal blocks are complicated, and their sums are even more complicated. Another approach is to construct 4-point functions in channel-independent ways, and then show that they can be decomposed into conformal blocks. This has led to analytic proofs of crossing symmetry in CFTs that can be constructed by perturbing a free bosonic CFT:

- In A-series minimal models, 4-point functions can be constructed as Coulomb gas integrals [34].

- In Liouville theory with $c \geq 25$, 4-point functions can be constructed from chiral vertex operators [35].

- Again in Liouville theory with $c \geq 25$, 4-point functions can be constructed as expectation values in a probabilistic Gaussian free field theory [36].

From these cases, all diagonal CFTs with 2 degenerate fields can be obtained by analytic continuation and taking limits (2.74), although mathematical rigor is lost in the process. In particular, the probabilistic construction only makes sense for $c \geq 25$.

In this text we will focus on a numerical approach to crossing symmetry. This approach is applicable to any CFT whose spectrum is exactly known, and therefore to all the exactly solvable CFTs that we have considered. A similar approach is also applicable to integrable conformal gauge theories such as $\mathcal{N} = 4$ super-Yang–Mills theory in 4 dimensions; these gauge theories however have spectrums that are much denser than in our 2-dimensional CFTs, requiring more sophisticated tools [37].

### 4.1.2 Various flavours of numerics

We consider a 4-point function $\langle V_1 V_2 V_3 V_4 \rangle$, where the 4 primary fields are characterized by their conformal dimensions. We assume that we exactly know the spectrum in each channel. If we do not know the structure constants, we view crossing symmetry as a system of linear equations (1.105) for the 4-point structure constants, normalized by the condition $D_{k_0}^{(s)} = 1$ for some $k_0$.

In the following diagram, we sketch what we can learn from numerically solving crossing symmetry, and give examples from exactly solvable CFTs. Potts connectivities are 4-point functions of the type $\left\langle V_{P_{(0,\frac{1}{2})}}^4 \right\rangle$, which have been an important testing ground for numerical bootstrap techniques.

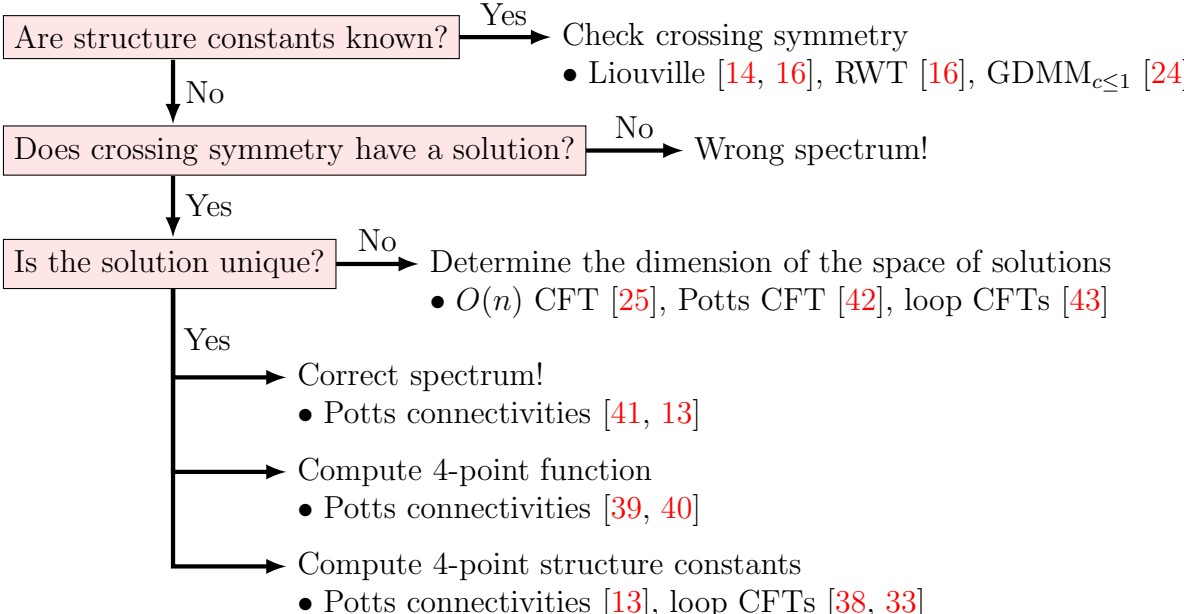

### 4.1.3 Factorization of 4-point structure constants

In order to solve a CFT, we should not only determine 4-point structure constants, but also factorize them into 2-point and 3-point structure constants. Factorization is simple enough if field multiplicities are trivial. However, there are nontrivial field multiplicities in D-series minimal models and in loop CFTs. This complicates factorization, because conformal blocks only depend on conformal dimensions. Solving the crossing symmetry equations (1.105) does not quite give us access to factorized 4-point structure constants $D_k^{(s)} = C_{12}^k C_{k34}$ (1.102), but to coefficients of conformal blocks:

$$D_{\Delta,\bar{\Delta}}^{(s)} = \sum_{k \in \mathcal{S}^{(s)}\big|_{\bar{\Delta}_k = \bar{\Delta}}^{\Delta_k = \Delta}} C_{12}^k C_{k34} \ . \tag{4.1}$$

This is a sum of as many terms as the multiplicity of the primary field $V_{\Delta,\bar{\Delta}}$ in the $s$-channel spectrum $\mathcal{S}^{(s)}$. Therefore, factorization only constrains 4-point structure constants if we know field multiplicities. In principle, knowing the spectrum means knowing field multiplicities. In practice however, in loop CFTs, we know the spectrum of conformal dimensions without knowing the multiplicities.

If the multiplicity $m(V_{\Delta,\bar{\Delta}})$ of $V_{\Delta,\bar{\Delta}}$ in the full spectrum is finite, it can in principle be determined from the values of $D^{(s)}_{\Delta,\bar{\Delta}}$ for all possible 4-point functions $\langle V_{k_1} V_{k_2} V_{k_3} V_{k_4} \rangle$:

$$m\left(V_{\Delta,\bar{\Delta}}\right) = \mathrm{rank}\left( D^{(s)}_{\Delta,\bar{\Delta}} \Big|_{\langle V_{k_1} V_{k_2} V_{k_3} V_{k_4} \rangle} \right)_{(k_1,k_2),(k_3,k_4) \in \mathcal{S}^2 \times \mathcal{S}^2} . \tag{4.2}$$

In principle it is impossible to compute the rank of an infinite matrix, but in practice it may be enough to study a few 4-point functions for guessing a given multiplicity. This has been done in a few examples of diagonal 4-point functions in loop CFTs [38].

## 4.2 Interchiral symmetry

As we have seen in Section 3, degenerate fields play a crucial role by constraining structure constants via shift equations. This suggests that we could extend the conformal algebra into an **interchiral algebra**, which would also include degenerate fields [44]. In the same way as the conformal algebra is generated by the energy-momentum tensors $T, \bar{T}$, leading to conformal Ward identities, an interchiral algebra would be generated by $T, \bar{T}, V^d_{\langle 1,2 \rangle}$ or $T, \bar{T}, V^d_{\langle 2,1 \rangle}, V^d_{\langle 1,2 \rangle}$, and we would consider shift equations as interchiral Ward identities.

Using interchiral symmetry rather than conformal symmetry, the spectrums (2.73) can be rewritten in terms of a smaller number of larger representations. In particular, each spectrum of an A-series or D-series minimal model is a combination of 4 (or fewer) interchiral representations. It would be interesting to reformulate the definition and classification of minimal models in terms of the representation theory of an interchiral algebra.

Here we will focus on the extended spectrum $\widehat{\mathcal{S}}^{\mathrm{loop}}$ (2.65), which involves the following representations of the interchiral algebra generated by $T, \bar{T}, V^d_{\langle 1,2 \rangle}$:

$$\widetilde{\mathcal{R}}^d_{\langle 1,s \rangle} = \bigoplus_{k \in \mathbb{N}} \mathcal{R}^d_{\langle 1,s+2k \rangle} \otimes \overline{\mathcal{R}}^d_{\langle 1,s+2k \rangle} \quad , \ s \in \{1,2\} , \tag{4.3a}$$

$$\widetilde{\mathcal{V}}_{(r,s)} = \bigoplus_{k \in \mathbb{Z}} \mathcal{V}_{P_{(r,s+2k)}} \otimes \overline{\mathcal{V}}_{P_{(-r,s+2k)}} \quad , \ r \in \tfrac{1}{2}\mathbb{N}^*, \ s \in \tfrac{1}{2r}\mathbb{Z} \cap (-1,1], \ (r,s) \notin \mathbb{N}^2 , \tag{4.3b}$$

$$\widetilde{\mathcal{V}}_{(r,s)} = \bigoplus_{k \in \mathbb{N}} \mathcal{W}_{(r,s+k)} \quad\quad\quad , \ r \in \mathbb{N}^*, \ s \in \{0,1\} , \tag{4.3c}$$

$$\widetilde{\mathcal{V}}_P = \bigoplus_{k \in \mathbb{Z}} \mathcal{V}_{P+k\beta^{-1}} \otimes \overline{\mathcal{V}}_{P+k\beta^{-1}} \quad , \ P \in \mathbb{C} , \tag{4.3d}$$

where $\mathcal{R}_{\langle r,s \rangle}$ is a degenerate representation of the Virasoro algebra, $\mathcal{V}_P$ a Verma module, and $\mathcal{W}_{(r,s)}$ a logarithmic representation as described in Section 2.4.1. In fact, we would like to define an interchiral algebra by the conditions that

- the representations (4.3) are irreducible,

- under reasonable assumptions, there are no other irreducible representations.

We will sketch a tentative construction of an interchiral algebra, but we will stop short of developing the technical machinery of interchiral symmetry. The concept can nevertheless be useful: for example, we may ask which boundary conditions break or preserve interchiral symmetry. Most importantly for us, we will compute interchiral blocks, which are what we need for the numerical bootstrap.

### 4.2.1 Towards an interchiral algebra

By definition, the interchiral algebra is supposed to shift momentums of primary fields. We therefore need to replace the Virasoro generator $L_0$, whose eigenvalues are conformal dimensions, with a generator whose eigenvalues are momentums. This can be done by rewriting the Virasoro algebra in terms of an **abelian affine Lie algebra $\hat{\mathfrak{u}}_1$**. The algebra $\hat{\mathfrak{u}}_1$ is the chiral symmetry algebra of free bosonic CFTs, which have a $\mathfrak{u}_1$ symmetry, and where the momentum is conserved. However, this algebra can also be used for studying non-free CFTs, giving rise to Coulomb gas techniques. (See [7, Section 4.1] for a quick review.)

The algebra $\hat{\mathfrak{u}}_1$ has generators $(J_n)_{n \in \mathbb{Z}}$ and relations

$$[J_m, J_n] = -\frac{1}{2} n \delta_{m+n,0} \ . \tag{4.4}$$

For any $c \in \mathbb{C}$, a Virasoro algebra $\mathfrak{V}$ with central charge $c$ can be constructed as a subalgebra of the universal enveloping algebra of $\hat{\mathfrak{u}}_1$:

$$L_n = \sum_{m \in \mathbb{Z}} J_{n-m} J_m + \left(\beta - \beta^{-1}\right)(n+1) J_n \ , \qquad (n \neq 0) \ , \tag{4.5a}$$

$$L_0 = 2 \sum_{m=1}^{\infty} J_{-m} J_m + J_0^2 + \left(\beta - \beta^{-1}\right) J_0 \ , \tag{4.5b}$$

where $\beta$ was defined from $c$ in Eq. (1.15). Let us define an affine primary state $V_P$ of momentum $P$ by

$$J_0 V_P = \left(P - \tfrac{1}{2}\left(\beta - \beta^{-1}\right)\right) V_P \quad , \quad J_{n>0} V_P = 0 \ . \tag{4.6}$$

Then $V_P$ is also a Virasoro primary state, whose conformal dimension is computed from the last 2 terms of Eq. (4.5b), reproducing Eq. (1.17).

Now, let us extend the algebra $\hat{\mathfrak{u}}_1 \times \overline{\hat{\mathfrak{u}}_1}$ by adding generators $D^{\pm}$ such that

$$[J_0, D^{\pm}] = [\bar{J}_0, D^{\pm}] = \pm \beta^{-1} D \quad , \quad [J_{n \neq 0}, D^{\pm}] = [\bar{J}_{n \neq 0}, D^{\pm}] = 0 \ . \tag{4.7}$$

We can deduce the commutation relations of $D^{\pm}$ with Virasoro generators,

$$[L_n, D^+] = 2\beta^{-1} D^+ J_n + \delta_{n,0} D^+ \quad , \quad [L_n, D^-] = 2\beta^{-1} J_n D^- + \delta_{n,0} D^- \ , \tag{4.8}$$

$$[\bar{L}_n, D^+] = 2\beta^{-1} D^+ \bar{J}_n + \delta_{n,0} D^+ \quad , \quad [\bar{L}_n, D^-] = 2\beta^{-1} \bar{J}_n D^- + \delta_{n,0} D^- \ . \tag{4.9}$$

Due to the generators $D^{\pm}$, the algebra is no longer chirally factorized into a left-moving and a right-moving factor: this is why it is called interchiral. We could have preserved chiral factorization by defining independent left-moving and right-moving generators $D^{\pm}, \bar{D}^{\pm}$, but such generators would produce states with spins that are not half-integer.

By construction, the generators $D^{\pm}$ act on affine primary states as $D^{\pm} V_P \propto V_{P \pm \beta^{-1}}$. Therefore, the representation $\widetilde{\mathcal{V}}_P$ (4.3d) is now irreducible as required. However, our algebra is still way too large, and has way too many representations. To correct this, we may impose additional constraints. A natural constraint is

$$[D^+, D^-] = 2\beta^{-1}(J_0 + \bar{J}_0) \ , \tag{4.10}$$

so that $\left(\beta(J_0 + \bar{J}_0), \beta D^+, \beta D^-\right)$ generate the Lie algebra $\mathfrak{sl}_2$. This constraint is not yet enough, because continuous representations of $\mathfrak{sl}_2$ come with 2 parameters, whereas our representations $\widetilde{\mathcal{V}}_P$ depend on only 1 parameter. As an additional constraint, we may set the quadratic Casimir to zero,

$$2(J_0 + \bar{J}_0)^2 + D^+ D^- + D^- D^+ = 0 \ . \tag{4.11}$$

This value of the quadratic Casimir is natural from the point of view of the interchiral identity representation $\widetilde{\mathcal{R}}^d_{\langle 1,1 \rangle}$ (4.3a), whose interchiral primary field $V^d_{\langle 1,1 \rangle}$ is an eigenvector of $J_0 + \bar{J}_0$ with eigenvalue 0. For this value of the Casimir, such a vector is annihilated by $D^-$, and is the lowest-weight state of a discrete representation of $\mathfrak{sl}_2$. As a result, $\widetilde{\mathcal{R}}^d_{\langle 1,1 \rangle}$ is a semi-infinite combination of Virasoro representations, as opposed to the more generic infinite combination $\widetilde{\mathcal{V}}_P$.

Our tentative construction of an interchiral algebra is surely not the final word on the subject. Remaining issues include:

- Explain algebraically what is special with having $D^\pm$ shift momentums by $\beta^{-1}$, as opposed to other numbers.

- Explain why $\widetilde{\mathcal{R}}^d_{\langle 1,2 \rangle}$ is a semi-infinite combination.

- Predict the structure of the logarithmic representations $\mathcal{W}_{(r,s)}$ in Eq. (4.3c), which are supposed to be deduced from the Verma modules $\mathcal{W}_{(r,0)}$ by interchiral symmetry. We will see in Section 5.3.2 how this works at the level of conformal blocks, but an algebraic derivation would be welcome.

- Define the fusion of interchiral representations, from which shift equations for 3-point structure constants should follow.

### 4.2.2 Interchiral blocks

Just like a Virasoro block is the sum of contributions of states in a representation of the Virasoro algebra, an **interchiral block** is associated to a representation of the interchiral algebra. For example, in the case of the representation $\widetilde{\mathcal{V}}_P$ (4.3d), the interchiral block $\widetilde{\mathcal{F}}_P$ is a linear combination of the conformal blocks $|\mathcal{F}_P|^2$, where the coefficients are dictated by shift equations.

In any 4-point function of an A-series or D-series minimal model, all $s$-channel 4-point structure constants are related by shift equations, and the 4-point function involves a single $s$-channel interchiral block. In minimal models, interchiral blocks only differ from correlation functions by overall constants. On the other hand, when solving crossing symmetry equations in loop CFTs, using interchiral blocks reduces the number of unknown structure constants. We will focus on computing the interchiral blocks that are relevant to loop CFTs, i.e. the interchiral blocks that package the shift equations from the degenerate field $V^d_{\langle 1,2 \rangle}$.

Let us compute $s$-channel interchiral blocks for the 4-point function $\left\langle \prod_{i=1}^{4} V_{(r_i,s_i)} \right\rangle$. In terms of Virasoro blocks, the interchiral block for the $s$-channel interchiral representation $\widetilde{\mathcal{V}}_{(r,s)}$ (4.3b) reads

$$\boxed{\widetilde{\mathcal{G}}_{(r,s)} = \sum_{k \in \mathbb{Z}} \frac{D^{(s)}_{(r,s+2k)}}{D^{(s)}_{(r,s)}} \mathcal{F}_{P_{(r,s+2k)}} \bar{\mathcal{F}}_{P_{(-r,s+2k)}}} \ . \tag{4.12}$$

To make this explicit, it remains to compute the ratios of 4-point structure constants. This is in principle a straightforward consequence of the shift equation for the 3-point structure constant (3.30). However, for certain values of the parameters, this shift equation is finite after cancelling poles from the numerator and denominator. Such cancellations can be avoided by writing the shift equations differently. In fact, in shifts of the reference 3-point structure constant $C^{\text{ref}}$ (3.54), no cancelling poles occur. The price to pay is that $C^{\text{ref}}$

only solves shift equations up to signs, see Eq. (3.59). Taking these signs into account, the shift equation (3.30) may be rewritten as

$$\frac{C_{(r_1,s_1-1)(r_2,s_2)(r_3,s_3)}}{C_{(r_1,s_1+1)(r_2,s_2)(r_3,s_3)}} = (-)^{2r_3}(-)^{\max(2r_1,2r_2,2r_3,r_1+r_2+r_3)}\beta^{-4\beta^{-2}s_1}$$

$$\times \prod_{\epsilon_2,\epsilon_3=\pm} \frac{\Gamma\left(\frac{1}{2} + |\epsilon_2 r_2 + \epsilon_3 r_3 - r_1| + \frac{\beta^{-2}}{2}(\epsilon_2 s_2 + \epsilon_3 s_3 - s_1)\right)}{\Gamma\left(\frac{1}{2} + |\epsilon_2 r_2 + \epsilon_3 r_3 + r_1| + \frac{\beta^{-2}}{2}(\epsilon_2 s_2 + \epsilon_3 s_3 + s_1)\right)} , \quad (4.13a)$$

where $2r_i, r_1 + r_2 + r_3 \in \mathbb{Z}$ from Eq. (2.11a). The shift equation for the 2-point structure constant follows from the 3-point case via Eq. (1.93),

$$\frac{B_{(r,s-1)}}{B_{(r,s+1)}} = (-)^{2r}\beta^{-8\beta^{-2}s} \prod_{a\in\{0,1,\beta^{-2},1-\beta^{-2}\}} \frac{\Gamma(a+r-\beta^{-2}s)}{\Gamma(a+r+\beta^{-2}s)} . \quad (4.13b)$$

From these shift equations, we can assemble shift equations for

$$D^{(s)}_{(r,s)} = \frac{C_{(r_1,s_1)(r_2,s_2)(r,s)}C_{(r,s)(r_3,s_3)(r_4,s_4)}}{B_{(r,s)}} , \quad (4.14)$$

and therefore the coefficients of the interchiral block $\widetilde{\mathcal{F}}_{(r,s)}$ (4.12).

The shifts of 2-point and 3-point structure constants are combinations of Gamma functions, and $\Gamma(x)$ has poles for $x \in -\mathbb{N}$. If all involved fields are non-diagonal, i.e. of the type $V_{(r,s)}$ with $r \in \frac{1}{2}\mathbb{N}^*$ and $s \in \frac{1}{2r}\mathbb{Z}$, the arguments of the Gamma functions are never in $-\mathbb{N}$ (assuming $\beta^2 \notin \mathbb{Q}$), and the shifts are finite. If some fields are diagonal, i.e. of the type $V_{(0,2\beta P)}$ with $P \in \mathbb{C}$, then we can hit the poles of Gamma functions. We will consider such cases as accidental and not discuss them further, except when the $s$-channel field is a degenerate field $V^d_{\langle 1,s\rangle} \sim V_{(0,-\beta^2+s)}$ with $s \in \mathbb{N}^*$. In this case, the 3-point ratio (4.13a) for $\left\langle V^d_{\langle 1,s\rangle} V_{(r_1,s_1)} V_{(r_2,s_2)}\right\rangle$ vanishes if $r_1 = r_2$ and $|s_1 - s_2| = s$. This means that the interchiral block does not contain a term for $V^d_{\langle 1,|s_1-s_2|-1\rangle}$, consistently with the fusion rule (2.10). Our 4-point function can have $s$-channel degenerate fields if and only if

$$r_1 = r_2 \quad , \quad r_3 = r_4 \quad , \quad s_1 - s_2, s_3 - s_4 \in \mathbb{Z} \quad , \quad s_3 - s_4 \equiv s_3 - s_4 \bmod 2 , \quad (4.15)$$

in which case the degenerate interchiral block is

$$\boxed{\widetilde{\mathcal{G}}^d = \sum_{k\in\mathbb{N}} \frac{D^{(s)}_{(1,s_0+2k)}}{D^{(s)}_{(1,s_0)}} \left|\mathcal{F}_{P_{(1,s_0+2k)}}\right|^2 , \quad \text{with} \quad s_0 = \max(|s_1-s_2|,|s_3-s_4|) + 1} . \quad (4.16)$$

## 4.3 Computation of Virasoro blocks

In order to solve crossing symmetry equations numerically, we need an efficient way to compute Virasoro blocks. It is possible to compute Virasoro blocks from their definition (1.98), but this is awfully inefficient. We will now review the more efficient recursive representation due to Alexey Zamolodchikov [45], in the case of 4-point $s$-channel blocks on the sphere. The $t$-channel and $u$-channel blocks can then be deduced using Eq. (1.103). For $N$-point blocks on arbitrary Riemann surfaces, see [46].

The idea of Zamolodchikov's recursion is to characterize Virasoro blocks by their analytic properties as functions of the channel dimension, in particular their poles, residues, and behaviour at $\infty$.

### 4.3.1 Poles and residues

As a function of the channel dimension $\Delta$, the $s$-channel Virasoro block $\mathcal{F}_\Delta(z)$ has a simple pole at any $\Delta = \Delta_{(r,s)}$ with $r, s \in \mathbb{N}^*$. The pole is due to the existence of the null vector $L_{\langle r,s\rangle}V_{\Delta_{(r,s)}}$, which is a primary state of dimension $\Delta_{(r,-s)}$. The block's residue at this pole is the sum of the contributions of the null vector and its descendant states, therefore the residue is itself a Virasoro block,

$$\boxed{\operatorname*{Res}_{\Delta=\Delta_{(r,s)}} \mathcal{F}_\Delta(z) = R_{r,s}\mathcal{F}_{\Delta_{(r,-s)}}(z)} . \tag{4.17}$$

The coefficient $R_{r,s}$ is called a **Virasoro block residue**. According to Eq. (1.100), the block's coefficients can be written in terms of the inverse Shapovalov form, which has a pole with residue (1.32), and descendant 3-point functions (1.79), which are regular at $\Delta = \Delta_{(r,s)}$. This leads to the expression

$$\boxed{R_{r,s} = \frac{c^{r,s}(P_1, P_2)c^{r,s}(P_3, P_4)}{b^{r,s}}} , \tag{4.18}$$

where the factors are

$$c^{r,s}(P_1, P_2) = g^{L_{\langle r,s\rangle}}_{\Delta_{(r,s)},\Delta_1,\Delta_2} \quad , \quad b^{r,s} = S'_{L_{(r,s)},L_{(r,s)}}(\Delta_{(r,s)}) . \tag{4.19}$$

Using the expression (1.81) of the Shapovalov form as a 2-point function, we write

$$S_{L_{(r,s)},L_{(r,s)}}(\Delta) = \frac{\langle L_{(r,s)}V_P(0)L_{(r,s)}V_P(\infty)\rangle}{\langle V_P(0)L_{(r,s)}V_P(\infty)\rangle}\Big\langle V_P(0)L_{(r,s)}V_P(\infty)\Big\rangle . \tag{4.20}$$

The first factor has a finite value at $P = P_{(r,s)}$, which coincides with $c^{r,s}(P_{(r,-s)}, P_{(1,1)})$, because $P_{(1,1)}$ is the momentum of the identity field. The second factor has a simple zero, and behaves like $(-)^{rs}c^{r,s}(P, P_{(1,1)})$, where the sign prefactor comes from exchanging the positions of the 2 fields. As a result, we find an expression for $b^{r,s}$ in terms of $c^{r,s}$:

$$b^{r,s} = (-)^{rs}c^{r,s}\left(P_{(r,-s)}, P_{(1,1)}\right) \frac{1}{2P_{(r,s)}} \frac{\partial}{\partial P}c^{r,s}\left(P, P_{(1,1)}\right)\Big|_{P=P_{(r,s)}} . \tag{4.21}$$

To determine $c^{r,s}$, let us deduce how it behaves under $r \to r+1$ from the relation (3.12) between $s$-channel and $t$-channel degenerate 4-point conformal blocks. The idea is to use this relation for 4-point functions of the types $\left\langle V^d_{\langle 2,1\rangle}V_{P_{(r,-s)}}V_{P_1}V_{P_2}\right\rangle$ and $\left\langle V^d_{\langle 2,1\rangle}V_{P_{(r,s)}}V_{P_1}V_{P_2}\right\rangle$. More precisely, let us compare the coefficients $F_{++}$ and $\widetilde{F}_{++}$ that relate the following conformal blocks:

$$\tag{4.22a}$$

$$\tag{4.22b}$$

The relation $V_{P_{(r,-s)}} \propto L_{\langle r,s \rangle} V_{P_{(r,s)}}$ leads to a relation between $F_{++}$ and $\widetilde{F}_{++}$. In fact, it may seem that these 2 coefficients coincide, since the relation (3.12) is the same for a descendant block as for the corresponding primary block. However, the coefficient $F_{++}$ is only valid for blocks that are normalized such that their asymptotics are as in Eq. (1.102). In order to correctly normalize blocks that involve descendant fields $L_{\langle r,s \rangle} V_{P_{(r,s)}}$ and/or $L_{\langle r+1,s \rangle} V_{P_{(r+1,s)}}$, we must multiply them with factors of the type $c^{r,s}$. As a result, we have

$$\frac{c^{r+1,s}(P_1, P_2)}{c^{r,s}(P_1, P_2 + \frac{\beta}{2})} \propto \frac{\widetilde{F}_{++}}{F_{++}} \propto \beta^{-2s} \frac{\prod_{\pm} \Gamma(\frac{1}{2} + \beta P_{(r,-s)} \pm \beta P_1 - \beta P_2)}{\prod_{\pm} \Gamma(\frac{1}{2} + \beta P_{(r,s)} \pm \beta P_1 - \beta P_2)} , \qquad (4.23)$$

where we use the expression (3.13) for $F_{++}$ and $\widetilde{F}_{++}$. By $\propto$ we mean that equalities are modulo $P_1, P_2$-independent factors. Unlike the residue $R_{r,s}$, such factors are not invariant under renormalizations $L_{\langle r,s \rangle} \to \lambda_{r,s}(\beta) L_{\langle r,s \rangle}$. We have used this freedom, and chosen factors that will make $c^{r,s}$ simple.

To solve the shift equation (4.23), and the analogous equation for $\frac{c^{r,s+1}}{c^{r,s}}$, we initialize the recursion by determining $c^{1,1}(P_1, P_2) = \prod_{\pm}(P_1 \pm P_2)$ from Eq. (1.53a). We then find

$$c^{r,s}(P_1, P_2) = \prod_{j=\frac{2}{1}-r}^{r-1} \prod_{k=\frac{2}{1}-s}^{s-1} \prod_{\pm} (P_2 \pm P_1 + P_{(j,k)}) = \frac{\prod_{\pm,\pm} \Gamma_\beta \left( \frac{\beta+\beta^{-1}}{2} + P_2 \pm P_1 \pm P_{(r,s)} \right)}{\prod_{\pm,\pm} \Gamma_\beta \left( \frac{\beta+\beta^{-1}}{2} + P_2 \pm P_1 \pm P_{(r,-s)} \right)} , \qquad (4.24)$$

where the second expression uses the double Gamma function (3.32). As a result of Eq. (3.47), we have $c^{r,s}(P_2, P_1) = (-)^{rs} c^{r,s}(P_1, P_2)$ and therefore $c^{r,s}(P_1, -P_2) = c^{r,s}(P_1, P_2)$. From Eq. (4.21), we then deduce

$$b^{r,s} = \frac{-(-)^{rs}}{2P_{(r,s)} P_{(0,0)}} \prod_{j=1-r}^{r} \prod_{k=1-s}^{s} 2P_{(j,k)} = \frac{-\prod_{\pm} \Gamma_\beta \left( \beta \pm 2P_{(r,s)} \right)}{P_{(r,s)} \Gamma_\beta \left( \beta + 2P_{(r,-s)} \right) \mathrm{Res}_{\beta - 2P_{(r,-s)}} \Gamma_\beta} . \qquad (4.25)$$

While the product formulas are adequate for numerically computing the Virasoro block residues $R_{r,s}$, the expressions in terms of the double Gamma function are convenient for elucidating their relations with structure constants of exactly solvable CFTs, such as Eq. (5.23).

Let us explicitly write these quantities for $rs \leq 4$. Since $c^{r,s}, b^{r,s}$ are invariant under $\left\{ \begin{smallmatrix} r \leftrightarrow s \\ \beta \to \beta^{-1} \end{smallmatrix} \right.$, we may assume $r \geq s$. With the help of the identity

$$\prod_{\pm,\pm} (P_2 \pm P_1 \pm \ell\beta) = (\Delta_1 - \Delta_2)^2 - 2\ell^2 \beta^2 (\Delta_1 + \Delta_2 - 1) + \ell^2(\ell^2 - 1)\beta^4 - \ell^2 , \qquad (4.26)$$

where conformal dimensions and momentums are related by Eq. (1.17), we find

$$c^{1,1}(P_1, P_2) = \Delta_1 - \Delta_2 , \qquad (4.27\mathrm{a})$$

$$c^{2,1}(P_1, P_2) = (\Delta_1 - \Delta_2)^2 - \tfrac{1}{2}\beta^2 (\Delta_1 + \Delta_2 - 1) - \tfrac{3}{16}\beta^4 - \tfrac{1}{4} , \qquad (4.27\mathrm{b})$$

$$c^{3,1}(P_1, P_2) = [\Delta_1 - \Delta_2] \left[ (\Delta_1 - \Delta_2)^2 - 2\beta^2 (\Delta_1 + \Delta_2 - 1) - 1 \right] , \qquad (4.27\mathrm{c})$$

$$c^{4,1}(P_1, P_2) = c^{2,1}(P_1, P_2) \left[ (\Delta_1 - \Delta_2)^2 - \tfrac{9}{2}\beta^2 (\Delta_1 + \Delta_2 - 1) + \tfrac{45}{16}\beta^4 - \tfrac{9}{4} \right] \qquad (4.27\mathrm{d})$$

$$\begin{aligned} c^{2,2}(P_1, P_2) = {}& (\Delta_1 - \Delta_2)^4 + \tfrac{c-13}{6}(\Delta_1 - \Delta_2)^2 (\Delta_1 + \Delta_2 - 1) \\ & + \tfrac{(c-1)(c-25)}{144}(\Delta_1 + \Delta_2 - 1)^2 - \left(1 + \tfrac{(c-1)(c-25)}{96}\right)(\Delta_1 - \Delta_2)^2 \\ & - \tfrac{(c-1)(c-13)(c-25)}{1152}(\Delta_1 + \Delta_2 - 1) + \tfrac{(c-1)(c-9)(c-17)(c-25)}{36864} , \quad (4.27\mathrm{e}) \end{aligned}$$

where the central charge $c$ is given by Eq. (1.15). And we have

$$b^{1,1} = -2 \ , \tag{4.28a}$$

$$b^{2,1} = -4 \left(1 - \beta^4\right) \ , \tag{4.28b}$$

$$b^{3,1} = -24 \left(1 - \beta^4\right) \left(1 - 4\beta^4\right) \ , \tag{4.28c}$$

$$b^{4,1} = -288 \left(1 - \beta^4\right) \left(1 - 4\beta^4\right) \left(1 - 9\beta^4\right) \ , \tag{4.28d}$$

$$b^{2,2} = 8 \left(\beta^2 - \beta^{-2}\right)^2 \left(1 - 4\beta^4\right) \left(1 - 4\beta^{-4}\right) = -\frac{2}{81}(c+2)(c-1)(c-25)(c-28) \ . \tag{4.28e}$$

### 4.3.2 The nome

In order to write Zamolodchikov's recursion for Virasoro blocks, and their large $\Delta$ asymptotics, it is convenient to replace the cross-ratio $z$ with the **nome** $q$, a function of $z$ built from the hypergeometric function:

$$\boxed{q = e^{i\pi\tau}} \quad \text{with} \quad \boxed{\tau = i\frac{{}_2F_1(\frac{1}{2}, \frac{1}{2}, 1, 1-z)}{{}_2F_1(\frac{1}{2}, \frac{1}{2}, 1, z)}} \ . \tag{4.29}$$

The inverse relation is

$$z = \frac{\theta_2^4(q)}{\theta_3^4(q)} \quad , \quad 1 - z = \frac{\theta_4^4(q)}{\theta_3^4(q)} \quad , \quad {}_2F_1\left(\tfrac{1}{2}, \tfrac{1}{2}, 1, z\right) = \theta_3^2(q) \ , \tag{4.30}$$

where $\theta_k(q)$ are **Jacobi theta functions**

$$\theta_2(q) = \sum_{n=-\infty}^{\infty} q^{(n+\frac{1}{2})^2} \quad , \quad \theta_3(q) = \sum_{n=-\infty}^{\infty} q^{n^2} \quad , \quad \theta_4(q) = \sum_{n=-\infty}^{\infty} (-1)^n q^{n^2} \ , \tag{4.31}$$

which obey the identities

$$\sum_{k=2,3,4} (-1)^k \theta_k^4(q) = 0 \quad , \quad \prod_{k=2,3,4} \theta_k(q) = 2q^{\frac{1}{4}} \prod_{m=1}^{\infty} \left(1 - q^{2m}\right)^3 \ . \tag{4.32}$$

This implies in particular the relation

$$q\left(\tfrac{z}{z-1}\right) = -q(z) \ , \tag{4.33}$$

as well as the asymptotic behaviour

$$z(q) \underset{q\to 0}{=} 16q + 128q^2 + O\left(q^3\right) \quad , \quad q(z) \underset{z\to 0}{=} \frac{z}{16} + \frac{z^2}{32} + O\left(z^3\right) \ . \tag{4.34}$$

Let us describe how the map $z \mapsto q$ acts on a few features of the complex $z$-plane. The features in question are related to the $S_3$ subgroup of global conformal transformations that permutes $z = 0, 1, \infty$, called the crossing symmetry group in [47]. The elements of this group are the following 6 global conformal maps:

| Map | $z$ | $\frac{z}{z-1}$ | $\frac{1}{z}$ | $1-z$ | $\frac{1}{1-z}$ | $1-\frac{1}{z}$ |
|---|---|---|---|---|---|---|
| Order | 1 | 2 | 2 | 2 | 3 | 3 |
| Fixed points | $\overline{\mathbb{C}}$ | $0, 2$ | $1, -1$ | $\infty, \frac{1}{2}$ | $e^{\pm i\frac{\pi}{3}}$ | $e^{\pm i\frac{\pi}{3}}$ |

$$\tag{4.35}$$

We will now plot these fixed points in the complex $z$-plane, and plot their images in $q(\mathbb{C})$. We also plot the circles $|z| = 1, |1 - z| = 1$ and the line $\left|\frac{z}{z-1}\right| = 1$, and their images, which split $\mathbb{C}$ and $q(\mathbb{C})$ into 6 fundamental domains of $S_3$:

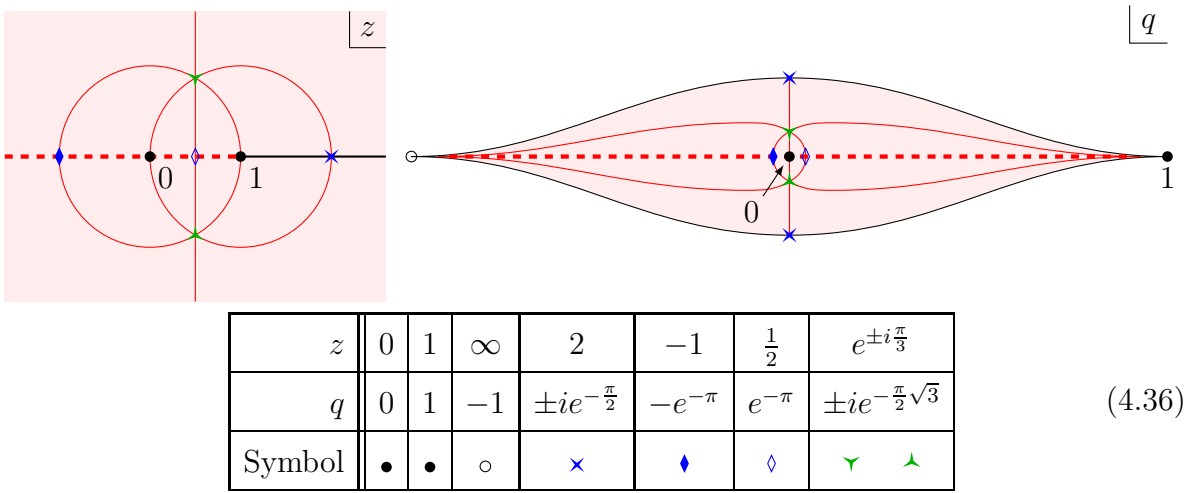

| $z$ | $0$ | $1$ | $\infty$ | $2$ | $-1$ | $\frac{1}{2}$ | $e^{\pm i \frac{\pi}{3}}$ |
|---|---|---|---|---|---|---|---|
| $q$ | $0$ | $1$ | $-1$ | $\pm i e^{-\frac{\pi}{2}}$ | $-e^{-\pi}$ | $e^{-\pi}$ | $\pm i e^{-\frac{\pi}{2}\sqrt{3}}$ |
| Symbol | ● | ● | ○ | ✗ | ◆ | ◇ | ⅄ ⅄ |

(4.36)

In the $z$-plane, the branch cut of the hypergeometric function is taken to be $(1, \infty)$, whose image is the boundary of $q(\mathbb{C})$. The whole complex $z$-plane is mapped into a domain that is much smaller than the unit disc $|q| < 1$: as a result, conformal blocks tend to converge faster as power series in $q$ than in $z$.

### 4.3.3 Zamolodchikov's recursion

As a function of the nome, the Virasoro block may be rewritten as a prefactor, times a power series $H_\Delta(q)$, which has a trivial asymptotic behaviour:

$$\mathcal{F}_\Delta(q) = (16q)^{\Delta - \frac{c-1}{24}} \left[ \prod_{k=2,3,4} \left( \theta_k^{-4}(q) \right)^{\Delta_k - \frac{c-1}{24} + (-1)^k \Delta_1} \right] H_\Delta(q) ,$$

(4.37)

$$\text{where} \quad H_\Delta(q) = 1 + \sum_{N=0}^\infty h_N(\Delta)(16q)^N \quad , \quad \lim_{\Delta \to \infty} H_\Delta(q) = 1 .$$

(4.38)

These formulas mean that $\log \mathcal{F}_\Delta(q) \underset{\Delta \to \infty}{=} O(\Delta)$, and that the $O(\Delta)$ and $O(1)$ terms have simple expressions in terms of the nome. In Section 4.3.4, we will sketch why this is the case. For the moment, let us complete the determination of the Virasoro block. From the poles and residues (4.17) of the Virasoro block $\mathcal{F}_\Delta(z)$, we deduce the poles and residues of the power series $H_\Delta(q)$ (4.37),

$$\underset{\Delta = \Delta_{(r,s)}}{\text{Res}} H_\Delta(q) = (16q)^{rs} R_{r,s} H_{\Delta_{(r,-s)}}(q) .$$

(4.39)

Since we moreover know its large $\Delta$ asymptotic behaviour (4.38), we can rewrite $H_\Delta(q)$ as a sum over its poles,

$$H_\Delta(q) = 1 + \sum_{r,s=1}^\infty \frac{(16q)^{rs} R_{r,s}}{\Delta - \Delta_{(r,s)}} H_{\Delta_{(r,-s)}}(q) .$$

(4.40)

Together with Eq. (4.37), this is called **Zamolodchikov's recursion** for Virasoro blocks. (There is another recursion for $\mathcal{F}_\Delta(z)$, obtained by summing over its poles as function of

$c$ instead of $\Delta$: to distinguish the 2 recursions, we may call them the $\Delta$-recursion and the $c$-recursion.)

The recursive reprentation completely determines the series $H_\Delta(q)$, order by order in powers of $16q$. Let us write the first 4 coefficients, using $\Delta_{(1,1)} = 0$ and Eq. (1.19) for simplifying some expressions:

$$h_1(\Delta) = \frac{R_{1,1}}{\Delta} \ , \tag{4.41a}$$

$$h_2(\Delta) = \frac{R_{1,1}^2}{\Delta} + \frac{R_{2,1}}{\Delta - \Delta_{(2,1)}} + \frac{R_{1,2}}{\Delta - \Delta_{(1,2)}} \ , \tag{4.41b}$$

$$h_3(\Delta) = \frac{R_{1,1}^3}{\Delta} + \frac{\left(3\Delta - \Delta_{(2,1)}\Delta_{(2,-1)}\right) R_{1,1}R_{2,1}}{\Delta(\Delta - \Delta_{(2,1)})(1 - \Delta_{(2,1)})\Delta_{(2,-1)}}$$
$$+ \frac{\left(3\Delta - \Delta_{(1,2)}\Delta_{(1,-2)}\right) R_{1,1}R_{1,2}}{\Delta(\Delta - \Delta_{(1,2)})(1 - \Delta_{(1,2)})\Delta_{(1,-2)}} + \frac{R_{3,1}}{\Delta - \Delta_{(3,1)}} + \frac{R_{1,3}}{\Delta - \Delta_{(1,3)}} \ , \tag{4.41c}$$

$$h_4(\Delta) = \frac{R_{1,1}^4}{\Delta} + \frac{\left(4\Delta - \Delta_{(2,1)}(\Delta_{(2,-1)} + 1)\right) R_{1,1}^2 R_{2,1}}{\Delta(\Delta - \Delta_{(2,1)})(1 - \Delta_{(2,1)})\Delta_{(2,-1)}} + \frac{\left(4\Delta - \Delta_{(1,2)}(\Delta_{(1,-2)} + 1)\right) R_{1,1}^2 R_{1,2}}{\Delta(\Delta - \Delta_{(1,2)})(1 - \Delta_{(1,2)})\Delta_{(1,-2)}}$$
$$+ \frac{R_{2,1}^2}{2(\Delta - \Delta_{(2,1)})} + \frac{\left(4\Delta - (\Delta_{(2,1)} - \Delta_{(1,2)})^2 - 2(\Delta_{(2,1)} + \Delta_{(1,2)})\right) R_{2,1}R_{1,2}}{(\Delta - \Delta_{(2,1)})(\Delta - \Delta_{(1,2)})(\Delta_{(2,-1)} - \Delta_{(1,2)})(\Delta_{(1,-2)} - \Delta_{(2,1)})}$$
$$+ \frac{R_{1,2}^2}{2(\Delta - \Delta_{(1,2)})} + \frac{\left(4\Delta - \Delta_{(3,1)}\Delta_{(3,-1)}\right) R_{1,1}R_{3,1}}{\Delta(\Delta - \Delta_{(3,1)})(1 - \Delta_{(3,1)})\Delta_{(3,-1)}} + \frac{\left(4\Delta - \Delta_{(1,3)}\Delta_{(1,-3)}\right) R_{1,1}R_{1,3}}{\Delta(\Delta - \Delta_{(1,3)})(1 - \Delta_{(1,3)})\Delta_{(1,-3)}}$$
$$+ \frac{R_{4,1}}{\Delta - \Delta_{(4,1)}} + \frac{R_{2,2}}{\Delta - \Delta_{(2,2)}} + \frac{R_{1,4}}{\Delta - \Delta_{(1,4)}} \ . \tag{4.41d}$$

In Zamolodchikov's recursion, some properties of the blocks are not manifest. In particular, $h_N(\Delta)$ is a meromorphic function of the central charge, whose poles depend on $\Delta$. However, the recursion writes $h_N(\Delta)$ as a sum of terms that are not always invariant under $\beta \to \beta^{-1}$, and therefore not meromorphic in $c$. Moreover, due to the denominator of $R_{r,s}$ (4.18), and also due to factors of the type $\frac{1}{\Delta_{(r,-s)} - \Delta_{(r',s')}}$, these terms have $\Delta$-independent poles at rational values of $\beta^2$. As a result, the recursion is only valid if $\beta^2 \notin \mathbb{Q}$. Finding a recursion for $\beta^2 \in \mathbb{Q}$ is an open problem, whose simplest incarnation is in the case of 1-point Virasoro blocks on the torus [48].

### 4.3.4 Behaviour for large channel dimensions

The asymptotic behaviour (4.37), (4.38) of Virasoro blocks in the limit $\Delta \to \infty$ is the most important piece of the derivation of Zamolodchikov's recursion. As far as we know, there is no complete derivation of this behaviour in the literature: in particular, Zamolodchikov's original argument [45] is elliptic. However, there are ideas that could well lead to a derivation: the algebraic idea of **exponentiation**, which states that $\log \mathcal{F}_\Delta$ grows only linearly in $\Delta$, and the conformal map to the pillow geometry, which explains the relevance of the nome.

In the expansion (1.99) of the Virasoro block as powers of $z$, the second-order coefficient (1.101b) behaves as $f_2(\Delta) = O(\Delta^2)$, and higher-order coefficients involve higher powers of $\Delta$. The linear growth of $\log \mathcal{F}_\Delta$ is therefore a non-trivial phenomenon, resulting from cancellations of faster-growing contributions to $\mathcal{F}_\Delta$. Virasoro blocks exponentiate also in limits where $c \to \infty$ while $\Delta, \Delta_i \to \infty$, starting with the **heavy limit** where $\frac{\Delta}{c}, \frac{\Delta_i}{c}$ are kept fixed [49].

Exponentiation is not easy to deduce from the computation of blocks as sums over Virasoro descendants, which leads to expressions for their coefficients in terms of 2 vectors and 1 matrix, see Eq. (1.101b) for the case of the $O(z^2)$ coefficient. It is better to sum over another basis of the Verma module $\mathcal{V}_\Delta$, namely the **oscillator basis** of descendants for the abelian affine Lie algebra $\hat{\mathfrak{u}}_1$, which is related to the Virasoro algebra by Eq. (4.5a). Since $\hat{\mathfrak{u}}_1$ is abelian, the corresponding matrix is diagonal, and easy to compute. This has led to a proof of exponentiation in the heavy limit [50], which can be generalized to other limits, including the limit $\Delta \to \infty$ we are interested in.

Once we know that Virasoro blocks exponentiate, it remains to compute the terms in $O(\Delta)$ and $O(1)$ of $\log \mathcal{F}_\Delta$. According to Eq. (4.37), these terms are (relatively) simple functions of the nome. We will now review a geometrical interpretation of these functions [51], which suggests how they could be derived. The idea is to flatten the 4-punctured sphere into a **pillow**, i.e. a double cover of a parallelogram of size $\pi \times \pi\tau$ or equivalently a $\mathbb{Z}_2$ quotient of a torus of size $2\pi \times 2\pi\tau$, such that the 4 punctures are mapped to the pillow's corners:

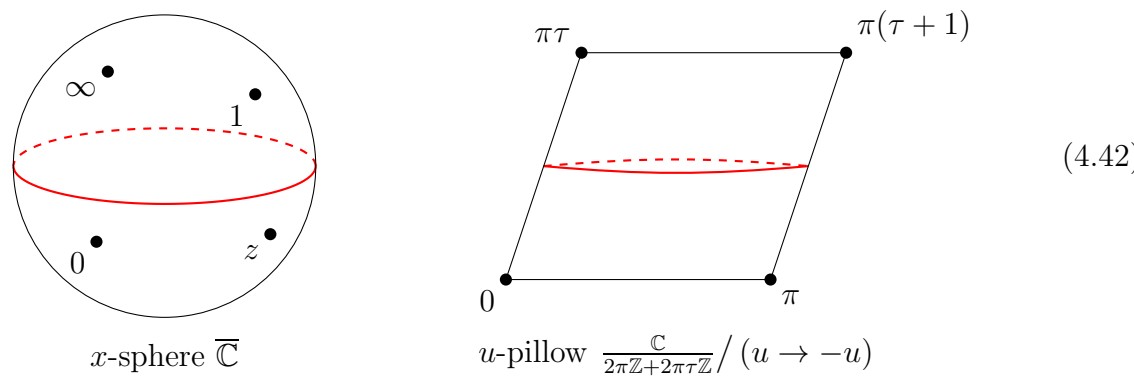

$$(4.42)$$

The relation between the coordinate $x$ on the sphere and the coordinate $u$ on the pillow is

$$u(x) = \frac{1}{{}_2F_1\left(\frac{1}{2}, \frac{1}{2}, 1, z\right)} \int_0^x \frac{dx'}{\sqrt{x'(x'-1)(x'-z)}} , \quad (4.43)$$

where the $z$-dependent prefactor is such that $u(z) = \pi$, and the square root's sign ambiguity leads to the identification $u \sim -u$. To compute $s$-channel blocks on the pillow, we imagine that time runs upwards, so that any constant time slice has length $2\pi$ (as drawn on Figure (4.42)). Such a slice scans the whole pillow if time runs for a duration $\pi\tau$. This leads to the factor $e^{i\pi\tau\Delta} = q^\Delta$ in the conformal block, where the $s$-channel dimension $\Delta$ is an eigenvalue of the $s$-channel Hamiltonian.

The conformal transformation from the sphere to the pillow modifies conformal blocks, which get multiplied by $c, \Delta_i$-dependent factors. These factors must be regularized, because the transformation is singular, with the 4 fields sitting at the singularities [51, Appendix D]. In $\mathcal{F}_\Delta(q)$ (4.37), these factors appear as exponentials of $\Delta_i$ with $i = 1, 2, 3, 4$, coming from the transformation of the 4 fields, and exponentials of $c$, coming from the Weyl anomaly. The regularized pillow Virasoro blocks are therefore

$$\mathcal{F}_\Delta^{\text{pillow}}(q) = (16q)^{\Delta - \frac{c}{24}} \prod_{m=1}^\infty \left(1 - q^{2m}\right)^{-\frac{1}{2}} H_\Delta(q) , \quad (4.44)$$

where we used Eq. (4.32) for simplifying the $c, \Delta_i$-independent factors.

Therefore, the derivation of the $\Delta \to \infty$ asymptotics of Virasoro blocks is surely easier on the pillow than on the sphere. The infinite product in Eq. (4.44) might naturally emerge from a calculation using the oscillator basis. More fundamentally, it would be interesting to understand why blocks simplify in the pillow geometry, and whether this is the unique geometry with this property.

## 4.4 Numerically solving crossing symmetry

Assuming we know the spectrum $\mathcal{S}$, the crossing symmetry equations (1.105) form a system of linear equations for the 4-point structure constants $D_k^{(x)}$. There are infinitely many unknowns if the spectrum is infinite. There are also infinitely many equations: 2 equations ($s = t$ and $t = u$) for each value of the cross-ratio $z \in \overline{\mathbb{C}}\backslash\{0, 1, \infty\}$. The number of solutions of crossing symmetry is

$$\mathcal{N} = \dim \left\{ \left( D_k^{(x)} \right)_{k,x} \bigg| \text{crossing symmetry is obeyed} \right\} \in \mathbb{N} \cup \{\infty\} . \qquad (4.45)$$

For numerical calculations on a computer, we need a system of finitely many equations with finitely many unknowns. Symmetries, including interchiral symmetry, can be used for reducing the number of unknowns, but do not make this number finite. We therefore need to truncate the system, by ignoring fields whose total conformal dimension exceeds a given number $\Lambda$. From the $\Lambda$-dependence of the solutions of the truncated system, we can see whether they converge towards a solution of the original system. And we can also infer the dimension of the space of solutions.

### 4.4.1 Reducing the number of unknowns

Some 4-point functions $\left\langle \prod_{i=1}^4 V_{\Delta_i, \bar{\Delta}_i} \right\rangle$ have discrete $\mathbb{Z}_2$ symmetries that reduce the number of unknowns. We will sketch the ideas here, and refer the reader to [33, Section 2.4] for technical details:

- The **parity transformation** $z \to \bar{z}$ exchanges the left- and right-moving Virasoro algebras, and therefore acts on non-diagonal fields as $V_{\Delta, \bar{\Delta}} \to V_{\bar{\Delta}, \Delta}$. If our 4-point function is parity-invariant i.e. if $\Delta_i = \bar{\Delta}_i$, then the space of solutions of crossing symmetry equations decomposes into a parity-even subspace with $D_{\Delta, \bar{\Delta}}^{(x)} = D_{\bar{\Delta}, \Delta}^{(x)}$, and a parity-odd subspace with $D_{\Delta, \bar{\Delta}}^{(x)} = -D_{\bar{\Delta}, \Delta}^{(x)}$.

- If $(\Delta_1, \bar{\Delta}_1) = (\Delta_2, \bar{\Delta}_2)$ in our 4-point function, then the permutation $z_1 \leftrightarrow z_2$ is a symmetry of the crossing symmetry equations. Again, this allows us to decompose the space of solutions into permutation-even and permutation-odd subspaces. Of course, if the fields $V_{\Delta_1, \bar{\Delta}_1}$ and $V_{\Delta_2, \bar{\Delta}_2}$ are actually identical, then only permutation-even solutions are relevant. However, if we allow non-trivial field multiplicities, then these 2 fields may differ while having the same conformal dimensions, and permutation-odd solutions may also be relevant. Our permutation acts on 4-point structure constants as

$$\left( D_{\Delta, \bar{\Delta}}^{(s)}, D_{\Delta, \bar{\Delta}}^{(t)}, D_{\Delta, \bar{\Delta}}^{(u)} \right) \mapsto \left( (-)^S D_{\Delta, \bar{\Delta}}^{(s)}, (-)^{S+S_1+S_3} D_{\Delta, \bar{\Delta}}^{(u)}, (-)^{S+S_1+S_3} D_{\Delta, \bar{\Delta}}^{(t)} \right) , \quad (4.46)$$

so that permutation-even solutions only involve $s$-channel fields with even conformal spins $S \in 2\mathbb{Z}$.

Let us now assume that the degenerate field $V_{\langle 1,2 \rangle}^d$ exists. As we saw in Section 4.2, this determines how 4-point structure constants $D_{(r,s)}^{(x)}$ behave under $s \to s + 2$, and we can rewrite the crossing symmetry equations (1.105) as

$$\forall z \in \overline{\mathbb{C}}\backslash\{0, 1, \infty\} , \quad \sum_{k \in \widetilde{\mathcal{S}}^{(s)}} D_k^{(s)} \widetilde{\mathcal{G}}_{\Delta_k, \bar{\Delta}_k}^{(s)}(z) = \sum_{k \in \widetilde{\mathcal{S}}^{(t)}} D_k^{(t)} \widetilde{\mathcal{G}}_{\Delta_k, \bar{\Delta}_k}^{(t)}(z) = \sum_{k \in \widetilde{\mathcal{S}}^{(u)}} D_k^{(u)} \widetilde{\mathcal{G}}_{\Delta_k, \bar{\Delta}_k}^{(u)}(z) ,$$

$$(4.47)$$

where $\widetilde{\mathcal{G}}^{(x)}_{\Delta,\bar{\Delta}}$ is now an interchiral block (4.12), and $\widetilde{\mathcal{S}}^{(x)} = \mathcal{S}^{(x)}\big|_{-1<s\leq 1}$ is a reduced spectrum. In loop CFTs, the reduced spectrum is still infinite, but the list (2.67) of non-diagonal fields $V_{(r,s)}$ with integer spins reduces to $2r$ fields for any given value of $r \in \frac{1}{2}\mathbb{N}^*$:

$$V_{(\frac{1}{2},0)}, \tag{4.48a}$$

$$V_{(1,0)}, V_{(1,1)}, \tag{4.48b}$$

$$V_{(\frac{3}{2},0)}, V_{(\frac{3}{2},\pm\frac{2}{3})}, \tag{4.48c}$$

$$V_{(2,0)}, V_{(2,\pm\frac{1}{2})}, V_{(2,1)}. \tag{4.48d}$$

The 4-point function $\left\langle \prod_{i=1}^4 V_{(r_i,s_i)} \right\rangle$ is parity-invariant if $r_i s_i = 0$. It is parity-invariant modulo interchiral symmetry if $s_i \in \mathbb{Z}$, and permutation-invariant modulo interchiral symmetry if $r_1 = r_2$ and $s_1 - s_2 \in 2\mathbb{Z}$. A $\mathbb{Z}_2$ invariance modulo interchiral symmetry still induces a $\mathbb{Z}_2$ action on the space of solutions, and focussing on an eigenspace of that action still reduces the number of unknowns by roughly half. However, eigenvalues no longer have a simple behaviour under $z \to \bar{z}$ or $z_1 \leftrightarrow z_2$.

### 4.4.2 Truncating to a finite system

Given a spectrum $\mathcal{S}$ and a cutoff $\Lambda \in \mathbb{R}_+$, let us define the truncated spectrum

$$\mathcal{S}_\Lambda = \left\{ k \in \mathcal{S} \big| \Re\left(\Delta_k + \bar{\Delta}_k\right) < \Lambda \right\} . \tag{4.49}$$

We assume that truncated channel spectrums $\mathcal{S}^{(x)}_\Lambda$ are finite, i.e. there are finitely many primary states in each channel. Unlike the condition (2.12) that OPE spectrums be bounded from below, this is not required by consistency of the CFT: for example, in Liouville theory, truncated spectrums are continuous for $\Lambda > \Re\left(\frac{c-1}{12}\right)$. Our finiteness condition in only necessary for our numerical method to apply. Moreover, we need to assume that the terms that we truncate out tend to zero as $\Lambda \to +\infty$. In Liouville theory, it can be shown that $D^{(x)}_\Delta \left| \mathcal{F}^{(x)}_\Delta(q) \right|^2 \underset{\Delta\to+\infty}{\propto} |q|^{2\Delta}$ with $|q| < 1$ [16], as a consequence of the behaviour of Virasoro blocks (4.37) and of structure constants. In loop CFTs, a similar behaviour is observed numerically.

The truncation scheme remains the same if we use interchiral blocks rather than conformal blocks: in each interchiral block, we keep only the conformal blocks whose total conformal dimension obeys the cutoff (4.49).

We then have to approximate conformal blocks so that they can be computed in finite time. Virasoro blocks are defined as infinite series, and Zamolodchikov's recursion also leads to an infinite series, see Section 4.3.3. In principle, in order to make errors of the same order of magnitude as when truncating the spectrum, we should use the same cutoff $\Lambda$, now applied to descendant states rather than primary states. This means that in the series $H_\Delta(q)$ (4.38), we should keep the term of order $q^N$ provided

$$\Re\left(\Delta + \bar{\Delta}\right) + N + \bar{N} < \Lambda . \tag{4.50}$$

However, it is impractical to perform correlated truncations of the left-moving and right-moving blocks, with truncation parameters that moreover depend on the channel dimensions $\Delta, \bar{\Delta}$. In practice, the lowest dimension $\min_{k\in\mathcal{S}} \Re\left(\Delta_k + \bar{\Delta}_k\right)$ is typically not far from zero, and we simply truncate all blocks to order $\Lambda$, i.e. we keep terms such that

$$N, \bar{N} < \Lambda . \tag{4.51}$$

Finally, now that we have a finite number of unknowns, we can afford to use only finitely many equations. One commonly-used possibility is to expand the crossing symmetry equations near a point $z_0$, and to keep finitely many terms in the expansion. In order to solve the bootstrap equations that involve only the $s$-channel and $t$-channel decompositions, it is natural to use $z_0 = \frac{1}{2}$, as this is the finite fixed point of $z \mapsto 1 - z$. In order to solve our 3-channel bootstrap equations, $z_0 = e^{i\frac{\pi}{3}}$ would be more natural, see Table (4.35). Another possibility is to refrain from expanding around a point, and to use a number of different values of $z$ instead. These values should be chosen reasonably far from one another, from the singularities $z = 0, 1, \infty$, and also from the real line $z \in \mathbb{R}$, where the conformal blocks $\mathcal{G}^{(x)}_{\Delta,\bar{\Delta}}$ and $\mathcal{G}^{(x)}_{\bar{\Delta},\Delta}$ coincide. Under these conditions, we can draw these values randomly, resulting in a finite set $Z_\Lambda$.

After these truncations and approximations, the crossing symmetry equations (1.105) reduce to a finite system of the type

$$
\forall z \in Z_\Lambda , \quad \sum_{k \in \mathcal{S}^{(s)}_\Lambda} D^{(s)}_k \mathcal{G}^{(s)}_{\Lambda|\Delta_k,\bar{\Delta}_k}(z) = \sum_{k \in \mathcal{S}^{(t)}_\Lambda} D^{(t)}_k \mathcal{G}^{(t)}_{\Lambda|\Delta_k,\bar{\Delta}_k}(z) = \sum_{k \in \mathcal{S}^{(u)}_\Lambda} D^{(u)}_k \mathcal{G}^{(u)}_{\Lambda|\Delta_k,\bar{\Delta}_k}(z) .
$$
(4.52)

After choosing a normalization, for example $D^{(s)}_{k_0} = 1$, we choose $Z_\Lambda$ such that $1 + |Z_\Lambda| = \sum_{x \in \{s,t,u\}} \left| \mathcal{S}^{(x)}_\Lambda \right|$, so that the system has as many equations as unknowns. Then there is a unique solution $\left( D^{(x)|\Lambda,Z_\Lambda}_k \right)_{k,x}$.

### 4.4.3   Studying one solution

In the limit $\Lambda \to \infty$, the solutions $\left( D^{(x)|\Lambda,Z_\Lambda}_k \right)_{k,x}$ of the truncated system converge towards an exact solution $\left( D^{(x)}_k \right)_{k,x}$ of crossing symmetry provided

$$
\forall k, x, Z_\Lambda , \quad \lim_{\Lambda \to \infty} D^{(x)|\Lambda,Z_\Lambda}_k = D^{(x)}_k .
$$
(4.53)

However, it is computationally expensive to use large values of the cutoff $\Lambda$ in order to test convergence. In practice, we instead diagnose convergence by studying how strongly $D^{(x)|\Lambda,Z_\Lambda}_k$ depend on $Z_\Lambda$. To do this quantitatively, we introduce the **relative deviation** of a structure constant for 2 choices $Z^{(1)}_\Lambda, Z^{(2)}_\Lambda$ of sets of points:

$$
\epsilon^{(x)|\Lambda}_k = \left| 1 - \frac{D^{(x)|\Lambda,Z^{(1)}_\Lambda}_k}{D^{(x)|\Lambda,Z^{(2)}_\Lambda}_k} \right| .
$$
(4.54)

While the relative deviation depends on $Z^{(1)}_\Lambda, Z^{(2)}_\Lambda$, we are only interested in its order of magnitude, which typically does not, so we omit this dependence in the notation $\epsilon^{(x)|\Lambda}_k$. When computing relative deviations, we encounter the following situations, depending on the number $\mathcal{N}$ (4.45) of solutions of crossing symmetry:

| Situation | Relative deviation |
|---|---|
| $\mathcal{N} \neq 1$ | $\log \epsilon^{(x)|\Lambda}_k \sim -2$ |
| $\mathcal{N} = 1, D^{(x)}_k \neq 0$ | $\log \epsilon^{(x)|\Lambda}_k \sim \Delta_k + \bar{\Delta}_k - \Lambda$ |
| $\mathcal{N} = 1, D^{(x)}_k = 0$ | $\log \epsilon^{(x)|\Lambda}_k \sim 0$ |

(4.55)

If there is no solution or if there are several solutions, relative deviations remain quite large, irrespective of $\Lambda$. A unique solution leads to small relative deviations, but only for structure constants that are nonzero.

For example, consider the 4-point function $\left\langle V_{(\frac{3}{2},\frac{2}{3})} V_{(\frac{1}{2},0)} V_{(1,0)} V_{P_{(0,\frac{4+i}{10})}} \right\rangle$ at $\beta = \frac{5}{4+i}$. Taking interchiral symmetry into account, we write the spectrums

$$\widetilde{S}^{(s)} = \left\{ V_{P_{(0,\frac{1+i}{10})}} \right\} \cup \left\{ V_{(r,s)} \right\}_{\substack{r \in \mathbb{N}^* \\ s \in \frac{1}{r}\mathbb{Z} \\ -1 < s \leq 1}} \quad , \quad \widetilde{S}^{(t)} = \widetilde{S}^{(u)} = \left\{ V_{(r,s)} \right\}_{\substack{r \in \frac{5}{2}+\mathbb{N} \\ s \in \frac{1}{r}\mathbb{Z} \\ -1 < s \leq 1}} \quad , \tag{4.56}$$

and impose the normalization condition $D^{(s)}_{P_{(0,\frac{1+i}{10})}} = 1$. The resulting crossing symmetry equations have a unique solution: we will justify this in Section 5.2.2, where that solution is characterized by the combinatorial map $\quad$ and the signature $\sigma = (0,\frac{5}{2},\frac{5}{2})$. We will now display numerical data, obtained with the following numerical parameters:

|  | | |
|---:|:---:|:---:|
| $\Lambda$ | 30 | 70 |
| #digits | 24 | 56 |
| running time | $\sim 200s$ | $\sim 4200s$ |

(4.57)

Times are indicated for Python code [52] running on an old desktop computer. For each value of $\Lambda$, we display the real parts of $s$- and $t$-channel structure constants together with their relative deviations, for all $r \leq \frac{7}{2}$ as well as $(r,s) \in \left\{ (4,0),(4,\frac{1}{4}),(\frac{9}{2},0),(\frac{9}{2},\frac{2}{9}) \right\}$. To

save space, we show only about 15 of the 24 or 56 digits:

| $(r,s)$ | $\epsilon^{(s)\|30}_{(r,s)}$ | $\Re D^{(s)\|30}_{(r,s)}$ | $\epsilon^{(s)\|70}_{(r,s)}$ | $\Re D^{(s)\|70}_{(r,s)}$ |
|---|---|---|---|---|
| $(1,0)$ | $10^{-18}$ | $\underline{0.82242208337639668}719$ | $10^{-45}$ | $0.82242208337639669048$ |
| $(1,1)$ | $10^{-17}$ | $-\underline{1.454893553160664480}3$ | $10^{-45}$ | $-1.4548935531606644841$ |
| $(2,0)$ | $10^{-13}$ | $-\underline{0.00058231020815326}98$ | $10^{-40}$ | $-0.0005823102081533416$ |
| $(2,\frac{1}{2})$ | $1.4$ | $-8.9601788211935 \times 10^{-17}$ | $1.8$ | $-8.924630755397 \times 10^{-44}$ |
| $(2,1)$ | $10^{-13}$ | $\underline{0.0005927888539317}2$ | $10^{-40}$ | $0.00059278885393164$ |
| $(2,-\frac{1}{2})$ | $1.3$ | $4.7020237099170 \times 10^{-17}$ | $1.7$ | $-2.961377839207 \times 10^{-44}$ |
| $(3,0)$ | $10^{-6}$ | $-\underline{5.6619}023163079 \times 10^{-11}$ | $10^{-32}$ | $-5.6619788484916 \times 10^{-11}$ |
| $(3,\frac{1}{3})$ | $1.5$ | $6.8733214151865 \times 10^{-15}$ | $2.4$ | $1.3028663659915 \times 10^{-41}$ |
| $(3,\frac{2}{3})$ | $1.2$ | $4.2685780225150 \times 10^{-15}$ | $2.4$ | $-1.45320554160523 \times 10^{-43}$ |
| $(3,1)$ | $10^{-6}$ | $\underline{1.4729916053293} \times 10^{-9}$ | $10^{-33}$ | $1.4730032519748 \times 10^{-9}$ |
| $(3,-\frac{1}{3})$ | $1$ | $-1.9732803811763 \times 10^{-15}$ | $1$ | $-2.1375702240518 \times 10^{-41}$ |
| $(3,-\frac{2}{3})$ | $1$ | $-8.3839892985966 \times 10^{-15}$ | $0.9$ | $-6.7477400595850 \times 10^{-42}$ |
| $(4,0)$ | $0.6$ | $4.9465341395038 \times 10^{-10}$ | $10^{-19}$ | $-2.5225903605711 \times 10^{-17}$ |
| $(4,\frac{1}{4})$ | $0.6$ | $-3.7885783283454 \times 10^{-9}$ | $1.4$ | $-2.95230287596625 \times 10^{-35}$ |

| $(r,s)$ | $\epsilon^{(t)\|30}_{(r,s)}$ | $\Re D^{(t)\|30}_{(r,s)}$ | $\epsilon^{(t)\|70}_{(r,s)}$ | $\Re D^{(t)\|70}_{(r,s)}$ |
|---|---|---|---|---|
| $(\frac{5}{2},0)$ | $10^{-14}$ | $-\underline{0.00287010107811272}060$ | $10^{-43}$ | $-0.00287010107811285690$ |
| $(\frac{5}{2},\frac{2}{5})$ | $10^{-14}$ | $-\underline{0.00403849708907029}441$ | $10^{-43}$ | $-0.00403849708907038200$ |
| $(\frac{5}{2},\frac{4}{5})$ | $10^{-14}$ | $\underline{0.0015317926329114}5239$ | $10^{-42}$ | $0.00153179263291161481$ |
| $(\frac{5}{2},-\frac{2}{5})$ | $10^{-14}$ | $\underline{0.00352412033056577}69$ | $10^{-43}$ | $0.0035241203305659808$ |
| $(\frac{5}{2},-\frac{4}{5})$ | $10^{-14}$ | $\underline{0.00695701472834329}452$ | $10^{-43}$ | $0.0069570147283419432 7917$ |
| $(\frac{7}{2},0)$ | $10^{-8}$ | $\underline{0.0000044445}8314475906$ | $10^{-35}$ | $0.0000044445 8367504768$ |
| $(\frac{7}{2},\frac{2}{7})$ | $10^{-8}$ | $\underline{0.0000041215 2}1383854464$ | $10^{-35}$ | $0.0000041215 20804451911$ |
| $(\frac{7}{2},\frac{4}{7})$ | $10^{-8}$ | $\underline{0.0000035139 10}538266622$ | $10^{-35}$ | $0.0000035139 10885535083$ |
| $(\frac{7}{2},\frac{6}{7})$ | $10^{-8}$ | $\underline{0.0000028787 94}266748157$ | $10^{-35}$ | $0.0000028787 94192189494$ |
| $(\frac{7}{2},-\frac{2}{7})$ | $10^{-8}$ | $\underline{0.0000040290 73}578395895$ | $10^{-35}$ | $0.0000040290 73277638004$ |
| $(\frac{7}{2},-\frac{4}{7})$ | $10^{-8}$ | $\underline{0.0000029308 93}945610566$ | $10^{-35}$ | $0.0000029308 94076496366$ |
| $(\frac{7}{2},-\frac{6}{7})$ | $10^{-8}$ | $\underline{0.0000017184 66}836222695$ | $10^{-35}$ | $0.0000017184 66769985650$ |
| $(\frac{9}{2},0)$ | $0.5$ | $2.2616940515940 \times 10^{-8}$ | $10^{-23}$ | $3.230122853228 \times 10^{-12}$ |
| $(\frac{9}{2},\frac{2}{9})$ | $0.5$ | $-3.5998726869265 \times 10^{-8}$ | $10^{-23}$ | $3.298879973039 \times 10^{-12}$ |

$$(4.58)$$

Let us interpret these results:

- Whenever a relative deviation is small, which in the case $\Lambda = 30$ means $\leq 10^{-6}$, we expect that we obtain an approximation of the corresponding structure constant. For example, $\epsilon^{(t)|30}_{(\frac{7}{2},\frac{2}{7})} \sim 10^{-8}$ suggests that $D^{(t)|30}_{(\frac{7}{2},-\frac{2}{7})}$ gives about 8 correct digits of the exact solution $D^{(t)}_{(\frac{7}{2},-\frac{2}{7})}$. This is confirmed by comparing with $D^{(t)|70}_{(\frac{7}{2},-\frac{2}{7})}$, which has the same 7 leading nonzero digits. In all $\Lambda = 30$ results, we have underlined the digits that agree with $\Lambda = 70$ results: the number of correct digits is always accurately estimated by the relative deviation.

- When a relative deviation is $O(1)$, we have either a vanishing structure constant, or a non-vanishing structure constant that can only be estimated with a larger cutoff. The $\Lambda = 30$ results indicate that $D^{(s)}_{(3,\frac{1}{3})} = 0$, because $V_{(3,0)}$ and $V_{(3,\frac{1}{3})}$ have similar total conformal dimensions, so according to Table (4.55) they should have similar

deviations if $D^{(s)}_{(3,\frac{1}{3})} \neq 0$. This is confirmed by the $\Lambda = 70$ results $\epsilon^{(s)|70}_{(3,\frac{1}{3})} = O(1)$ and $D^{(s)|70}_{(3,\frac{1}{3})} \ll D^{(s)|30}_{(3,\frac{1}{3})}$. Similarly, the $\Lambda = 30$ results are enough for concluding $D^{(s)}_{(2,\pm\frac{1}{2})} = D^{(s)}_{(3,\pm\frac{1}{3})} = D^{(s)}_{(3,\pm\frac{2}{3})} = 0$.

- In the cases $\epsilon^{(s)|30}_{(4,\star)}, \epsilon^{(t)|30}_{(\frac{9}{2},\star)} = O(1)$, the $\Lambda = 30$ results are not enough for knowing whether the corresponding structure constants vanish. The $\Lambda = 70$ results show that $D^{(s)}_{(4,s)} = 0 \iff s \notin \mathbb{Z}$ and $D^{(t)}_{(\frac{9}{2},\star)} \neq 0$. Based on all these results, it is natural to conjecture $\forall r, s, \ D^{(s)}_{(r,s)} = 0 \iff s \notin \mathbb{Z}$.

### 4.4.4 Counting solutions

We would like to determine the dimension $\mathcal{N}$ (4.45) of the space of solutions, and to find a basis of solutions.

If we view the system of crossing symmetry equations as an infinite matrix $\mathcal{C}$, then $\mathcal{N}$ is the number of singular values of $\mathcal{C}$ that vanish. Can we deduce $\mathcal{N}$ from the matrix $\mathcal{C}_\Lambda$ of a truncated system? The truncation has 2 effects on the set of singular values: making the set finite, and making the vanishing elements nonzero but tiny. Therefore, if $\mathcal{N} < \infty$, then $\mathcal{N}$ is the number of *tiny* singular values of $\mathcal{C}_\Lambda$ for $\Lambda$ *large enough*. But large matrices tend to have many small singular values, which tend to zero as $\Lambda \to \infty$. The tiny singular values tend to zero faster than the others, but we do not have a quantitative criterion for characterizing them. (See [43] for examples.)

In practice, instead of studying singular values, we count solutions by a rather pedestrian method, which does not require high values of $\Lambda$. To begin with, let us determine whether $\mathcal{N} > 0$. For a solution $\left( D^{(x)|\Lambda,Z_\Lambda}_k \right)_{k,x}$ of the truncated system (4.52), let $Z^{(x)}_\Lambda(z)$ be the corresponding $x$-channel 4-point function. For $z_0 \notin Z_\Lambda$, we compute the violation of crossing symmetry

$$\delta_\Lambda(z_0) = \max_{x \in \{t,u\}} \left| Z^{(x)}_\Lambda(z_0) - Z^{(s)}_\Lambda(z_0) \right| . \tag{4.59}$$

We expect this violation to be small if $\mathcal{N} > 0$, and larger if $\mathcal{N} = 0$. To state this quantitatively, we introduce the quantity

$$\eta_\Lambda = \max_{k \in \mathcal{S}^{(s)}_\Lambda} \left| \epsilon^{(s)|\Lambda}_k D^{(s)|\Lambda,Z_\Lambda}_k \mathcal{G}^{(s)}_{\Lambda|\Delta_k \bar{\Delta}_k}(z_0) \right| . \tag{4.60}$$

Then we expect

- $\mathcal{N} = 0 \implies \delta_\Lambda(z_0) = O(\eta_\Lambda)$: the large deviation $\epsilon^{(s)|\Lambda}_k$ measures the failure of crossing symmetry.

- $\mathcal{N} > 1 \implies \delta_\Lambda(z_0) \ll \eta_\Lambda$: the large deviation reflects the existence of several solutions, and the 3 channels should agree up to a small discrepancy that is due to our truncation of the system.

Once we know that $\mathcal{N} > 1$, the idea is to add $\mathcal{N}$ equations to the crossing symmetry equations. Then the solution becomes unique, leading to small relative deviations as in Section 4.4.3. Without loss of generality, we may use equations of the type $D^{(x)}_k = 0$, in addition to the normalization condition $D^{(s)}_k = 1$. This is equivalent to removing primary fields from the spectrum, until we have small relative deviations. When this is achieved, we have $\mathcal{N} \leq \mathcal{R} + 1$, where $\mathcal{R}$ is the number of removed fields. We have an inequality

rather than an equality, because it may happen that some of our additional equations are in fact obeyed by all solutions, and are therefore redundant with crossing symmetry equations. To eliminate this possibility, we should test whether removing all $\mathcal{R}$ primary fields is necessary for having small relative deviations. If the set of removed fields is minimal, then $\mathcal{N} = \mathcal{R} + 1$.

The method of removing primary fields is computationally cheap, because it only involves deleting columns of the matrix $\mathcal{C}_\Lambda$, without recomputing $\mathcal{C}_\Lambda$, or changing the cutoff $\Lambda$. This method is however pedestrian, because it involves inspecting relative deviations for various sets of removed fields, and determining a minimal set by trial and error. It would be nice to find a simple way of determining $\mathcal{N}$ from $\mathcal{C}_\Lambda$.

The method of removing primary fields can also work if $\mathcal{N} = \infty$. For example, in the case of 4-point connectivities in the Potts CFT, using only the first equation in (1.105) (i.e. leaving the $u$-channel unconstrained) leads to an infinite-dimensional space of solutions, instead of the finite-dimensional spaces that otherwise occur in loop CFTs. Nevertheless, it is possible to determine a basis of solutions, where each element corresponds to a set of removed fields that is infinite before truncation [13].

# 5 Correlation functions in loop CFTs

We will now take a short break from CFT, and build correlation functions as statistical sums over loop ensembles. This will motivate the introduction of combinatorial maps, which give rise to a basis of solutions of crossing symmetry. We can then forget about the statistical sums, which are not universal objects, while keeping the combinatorial maps, which are intrinsic features of the CFT.

## 5.1 Statistical sums over loop ensembles

### 5.1.1 Sums over configurations of non-intersecting loops

In statistical physics, loop models have correlation functions of the type

$$Z(\mathcal{E}) = \sum_{E \in \mathcal{E}} W(E) \ , \tag{5.1}$$

where $E$ is a set of non-intersecting closed loops on the Riemann sphere $\overline{\mathbb{C}}$, belonging to an ensemble $\mathcal{E}$, and $W$ is a weight function.

If we discretize $\overline{\mathbb{C}}$ and replace it with a finite lattice, the ensemble $\mathcal{E}$ becomes finite, and each configuration $E$ is made of finitely many loops: $|\mathcal{E}|, |E| < \infty$. Then the sum $Z(\mathcal{E})$ is manifestly finite, and can be numerically evaluated on a computer. The price to pay for the lattice discretization is that conformal symmetry is broken, and reemerges only in a large lattice size limit — provided $Z(\mathcal{E})$ does have a limit.

An alternative is the mathematical notion of a conformal loop ensemble, which is an infinite ensemble $\mathcal{E}$ of configurations of infinitely many loops: $|\mathcal{E}| = |E| = \infty$. This ensemble is conformally invariant by construction.

We will not need a precise definition of a loop ensemble: we only assume $|E| < \infty$ for technical simplicity. We assume that the energy $\log W(E)$ is extensive: then the weight of a configuration is a product of weights of individual loops $w(\ell)$. On the sphere, all loops are topologically the same, and must have the same weight by conformal symmetry. Therefore,

$$W(E) = \prod_{\ell \in E} w(\ell) = n^{|E|} \ , \tag{5.2}$$

where $n$ is the **contractible loop weight**.

### 5.1.2 Legless punctures

Our correlation function $Z(\mathcal{E})$ is the analog of a 0-point function in CFT. In order to have analogs of an $N$-point function, we introduce $N$ **legless punctures** on our sphere with positions $z_1, \ldots, z_N$, which will modify the weights of loops. Without violating conformal symmetry, we can let the weight of a loop depend on how the loop splits the set $\{z_1, \ldots, z_N\}$ into 2 subsets, i.e. on the loop's combinatorial properties. It cannot however depend on the loop's topology in $\overline{\mathbb{C}} \backslash \{z_1, \ldots, z_N\}$, because we want $Z(\mathcal{E})$ to be a single-valued function of $z_1, \ldots, z_N$. For example, in the case $N = 4$, the following 2 loops are topologically different but combinatorially equivalent, and must therefore have the same weight:

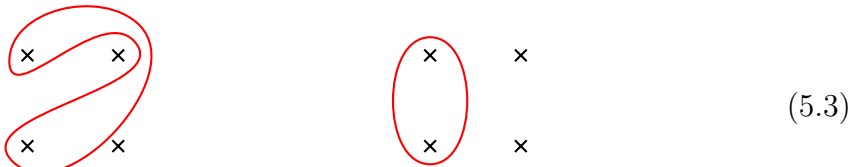

$$(5.3)$$

This leads to 8 combinatorially inequivalent classes of loops, with 8 different weights: contractible loops with weight $n$, $i$-loops around the puncture at $z_i$ with weight $w_i$, and $s$-loops, $t$-loops, $u$-loops that split the 4 punctures in pairs:

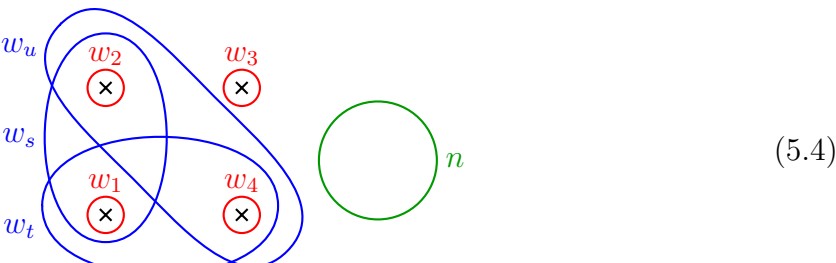

$$(5.4)$$

Since loops do not intersect, a configuration $E$ that contains $s$-loops has no $t$-loops or $u$-loops.

### 5.1.3 Legged punctures and combinatorial maps

We also introduce **legged punctures** (sometimes called watermelon operators), where open loops can end. Such a puncture has a valency $2r \in \mathbb{N}^*$, with $r = 0$ corresponding to the previously introduced legless punctures. In the presence of $N$ punctures, the number of open loops is

$$\sum_{i=1}^{N} r_i \in \mathbb{N} \ .$$

$$(5.5)$$

Conformal symmetry allows us to specify how punctures are connected by open loops: this information is called a **combinatorial map** [43]. A combinatorial map is a graph whose vertices correspond to punctures, and whose edges correspond to open loops. For example, there exist 4 combinatorial maps with vertices of valencies $3, 2, 1, 0$:

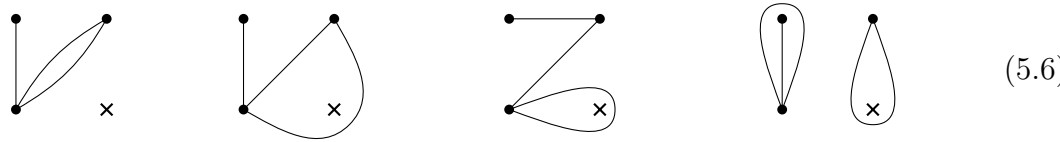

$$(5.6)$$

Notice that we forbid edges that can be pulled inside vertices, as the corresponding open loops are indistinguishable from contractible loops. This is the reason why we do not count the following map:

$$(5.7)$$

To a combinatorial map $M$, we associate the ensemble $\mathcal{E}_M$ of configurations of non-intersecting open and closed loops, such that the open loops connect the punctures according to $M$. The map $M$ dictates which types of closed loops can exist: for example, the last map in Eq. (5.6) allows closed loops with weights $n, w_4, w_s$. Conformal symmetry allows the weight of a loop configuration to depend on the relative angles of open loops at punctures. More precisely, at any puncture we have a weight factor that depends on an angular momentum $s$,

$$\boxed{w_{(r,s)}(E) = \exp\left\{ \frac{i}{2}s\sum_{k=1}^{2r}\theta_k \right\}}\,. \tag{5.8}$$

where the angles are measured with respect to a reference direction, and a reference open loop whose angle is $\theta_1$:

$$\theta_k \in [\theta_1, \theta_1 + 2\pi)$$

$$(5.9)$$

The reference direction would define each angle modulo $2\pi$. Thanks to the reference open loop and the cyclic ordering of open loops around the puncture, our angles are actually defined modulo a collective shift $(\theta_k) \to (\theta_k + 2\pi)$. This shift leaves our weight factor invariant provided

$$rs \in \mathbb{Z}\,. \tag{5.10}$$

Under a change of reference direction or reference open loop, the weight of a configuration $E$ changes by an $E$-independent factor.

We therefore define correlation functions that depend on the punctures $z_1, \ldots, z_N$ with their parameters $(r_1, s_1), \ldots, (r_N, s_N)$, on the map $M$ and on the weights of closed loops $\{w_k\}$:

$$\boxed{Z\left(\{z_i\}, \{(r_i, s_i)\}, \{w_k\}\Big|\mathcal{E}_M\right) = \sum_{E\in\mathcal{E}_M} \prod_{i=1}^{N} w_{(r_i,s_i)}(E) \prod_{\substack{\ell\in E \\ \ell \text{ closed}}} w(\ell)}\,. \tag{5.11}$$

### 5.1.4 Cases of the $O(n)$, $PSU(n)$ and Potts models

In the $O(n)$ model, all possible legged punctures are allowed. But there are no legless punctures: all closed loops have the same weight $n$, whether or not they are contractible. The loop model is obtained by reformulating a model of spins on a lattice, where each spin is an $n$-dimensional vector that transforms in the fundamental representation of the group $O(n)$.

In the $PSU(n)$ model, we add the restriction that all loops (closed and open) can be oriented such that any 2 neighbouring loops have opposite orientations. Equivalently, the Riemann sphere should be bicolorable, with each loop separating 2 regions of different colors. This allows only punctures with even valencies $2r \in 2\mathbb{N}^*$:

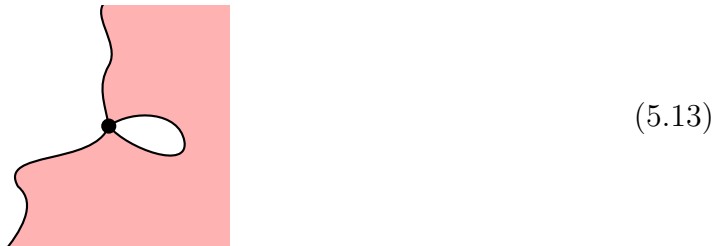

$$(5.12)$$

Moreover, we introduce legless punctures that detect the color of the region they belong to. Since that color does not change if we add 2 closed loops around the puncture, the puncture's weight must obey $w^2 = n^2$, therefore $w = -n$.

The Potts model is the same as the $PSU(n)$ model, except that the fundamental geometrical objects are clusters instead of loops, with loops forming the boundaries of clusters. In general, loops and clusters are equivalent: in the above example, a puncture of valency 4 is a point where 2 large clusters meet, where a cluster is large if it cannot be pulled inside the puncture. In the following example, an open loop can be pulled inside the puncture, and we have a squeezed cluster rather than the meeting of 2 different clusters:

$$(5.13)$$

However, there is a subtle difference between loops and clusters in the case of a puncture of valency 2. In the following 2 examples, which are equivalent in the loop model, the puncture belongs to a large cluster in the first case but not in the second one:

$$(5.14)$$

The correct condition for a puncture to belong to 1 large cluster is in fact that no closed loop goes around that puncture. Therefore, the Potts model has punctures with valencies $2r \in 2(\mathbb{N}+2)$, and legless punctures of weight $w = 0$.

## 5.2 Relation with loop CFTs

The claim that loop CFTs describe the critical limit of loop models is supported by a large body of work, including direct numerical checks. However, it is not easy to derive it. In fact, it is already tricky to derive the relation between the central charge $c$ and the contractible loop weight $n$. See for example [53, Appendix A] for a discussion of Kondev's original argument, and for an attempt to patch its perceived flaws with the help of the torus partition function.

We will now propose an argument that starts with relating non-diagonal primary fields $V_{(r,s)}$ with legged punctures. This is relatively straightforward, because both objects have

discrete parameters. We will then deduce the relation between diagonal fields $V_P$ and legless punctures, and finally the relation between the central charge and the contractible loop weight.

### 5.2.1 Relations between punctures and primary fields

Our starting assumption is that in the critical limit, a legged puncture tends to a primary field of a loop CFT. We have used the same notation $r, s$ for these punctures' parameters as for the non-diagonal fields $V_{(r,s)}$. Let us justify this notation by showing that $r, s$ take the same values in the loop model as in the CFT:

- In the CFT, as a result of the existence of the degenerate field $V^d_{\langle 1,2 \rangle}$, the first Kac index $r$ is half-integer, and conserved modulo $\mathbb{Z}$, see Eq. (2.11a). For a legged puncture, $2r$ is the valency, and it obeys the same constraints, see Eq. (5.5).

- In the CFT, $rs$ is the conformal spin, and it is integer for bosonic fields. Under a rotation around the puncture, which we interpret as $\theta_k \mapsto \theta_k + \theta$, the weight factor (5.8) picks up a factor $e^{irs\theta}$: according to Eq. (1.85), this means that our legged puncture has the conformal spin $rs$.

Therefore, a legged puncture can be identified with a non-diagonal primary field $V_{(r,s)}$ in a CFT where $V^d_{\langle 1,2 \rangle}$ exists.

Next, let us consider a legless puncture of weight $w$. We again assume that it tends to a primary field. Since $w$ is a continuous parameter, this must be a diagonal field. To determine its momentum, let us use the idea (2.66) that a diagonal field is a non-diagonal field with $r = 0$. The analogous idea for punctures is that the weight $w$ of a closed loop is related to the weight factor (5.8) associated to a rotation $\theta = 2\pi$. In fact, since our closed loop is not oriented, we have to add the rotations by $2\pi$ and $-2\pi$, and we find $w = 2\cos(\pi s)$, which is equivalent to Eq. (3.61).

Finally, the identity field $V^d_{\langle 1,1 \rangle}$ should not change the weight of loops. Assuming that the corresponding loop weight is still given by Eq. (3.61), we obtain $n = w(P_{(1,1)})$, which boils down to the expression (3.62) for the contractible loop weight $n$ in terms of the central charge. We do not discuss the loop model interpretation of the non-trivial degenerate fields $V^d_{\langle 1,s \geq 2 \rangle}$, because we will not study their correlation functions. These fields cannot simply correspond to legless punctures, if only because $w(P_{(1,3)}) = w(P_{(1,1)}) = n$.

### 5.2.2 Correlation functions

According to our identification of punctures with primary fields, the loop model $N$-point function (5.11) corresponds to a loop CFT $N$-point function of the type $\left\langle \prod_{i=1}^N V_{(r_i,s_i)} \right\rangle$. In the bootstrap approach, $N$-point functions are defined as single-valued solutions of crossing symmetry equations. To write these equations, we need to know the spectrum in any channel. We introduce the following notations for spectrums:

$$\mathcal{S}_\sigma \underset{\sigma \in \frac{1}{2}\mathbb{N}^*}{=} \left\{ V_{(r,s)} \right\}_{\substack{r \in \sigma + \mathbb{N} \\ s \in \frac{1}{r}\mathbb{Z}}} \quad , \quad \mathcal{S}_0(P) = \left\{ V_{P+k\beta^{-1}} \right\}_{k \in \mathbb{Z}} \bigcup \left\{ V_{(r,s)} \right\}_{\substack{r \in \mathbb{N}^* \\ s \in \frac{1}{r}\mathbb{Z}}} . \tag{5.15}$$

Let $\mathcal{Z}$ be the space of solutions of crossing symmetry for $\left\langle \prod_{i=1}^N V_{(r_i,s_i)} \right\rangle$, where in each channel the spectrum is either $\mathcal{S}_0(P)$ or $\mathcal{S}_{\frac{1}{2}}$, as determined by the conservation of $r$ mod 1. This spectrum includes all non-diagonal fields that are allowed by the conservation of $r$ mod 1, plus a family of diagonal fields whenever $r$ is integer.

Given a 4-point combinatorial map $M$, let us define the $x$-channel **signature** $\sigma^{(x)} \in \frac{1}{2}\mathbb{N}$ as half the minimum number of lines in $M$ that are crossed by an $x$-loop. For example,

the map  has signature $(\sigma^{(s)}, \sigma^{(t)}, \sigma^{(u)}) = (0, 2, 2)$. Then let us define the spectrum $\mathcal{S}_M = (\mathcal{S}_M^{(s)}, \mathcal{S}_M^{(t)}, \mathcal{S}_M^{(u)}) = (\mathcal{S}_{\sigma^{(s)}}, \mathcal{S}_{\sigma^{(t)}}, \mathcal{S}_{\sigma^{(u)}})$. In the case $\sigma^{(x)} = 0$, there can exist $x$-loops with weight $w_x = 2\cos(2\pi\beta P_x)$, and we take the $x$-channel spectrum to be $\mathcal{S}_0(P_x)$.

We conjecture that each combinatorial map $M$ with vertices of valencies $(2r_1, \dots, 2r_N)$ corresponds to a solution $Z_M = \lim_{\text{critical}} Z(\mathcal{E}_M)$ of the crossing symmetry equations with spectrums $\mathcal{S}_M$, and that these solutions form a basis of the space of solutions [43]:

$$\mathcal{Z} = \text{Span}\left\{Z_M\right\}_M \ . \tag{5.16}$$

For example, the existence of the 4 combinatorial maps (5.6) implies that for any allowed values of the parameters $s_1, s_2, s_3, P_4$, the space of 4-point functions of the type $\left\langle V_{(\frac{3}{2}, s_1)} V_{(1, s_2)} V_{(\frac{1}{2}, s_3)} V_{P_4} \right\rangle$ has dimension 4.

A subtelty is that crossing symmetry equations for $Z_M$ may in fact have several linearly independent solutions. To see this, notice that signatures lead to a natural partial ordering on combinatorial maps, with $M \geq M' \iff \forall x, \sigma^{(x)}(M) \geq \sigma^{(x)}(M')$. If $M \geq M'$, then $\forall x, \ \mathcal{S}_M^{(x)} \subset \mathcal{S}_{M'}^{(x)}$, and both $Z_M, Z_{M'}$ solve the crossing symmetry equations with spectrum $\mathcal{S}_{M'}$. This happens in particular if 2 maps have the same signature, for example  and . On the other hand, if $M$ is maximal for our ordering, we expect that the corresponding crossing symmetry equations have a 1d space of solutions, proportional to $Z_M$.

Can we interpret the elements of our space $\mathcal{Z}$ as 4-point functions $\left\langle \prod_{i=1}^{4} V_{(r_i, s_i)} \right\rangle$? A potential puzzle is that the 4-point functions should only depend on the 4 fields $V_{(r_i, s_i)}$, whereas our solutions can depend on channel weights, for example on $w_s$ if $r_1 + r_2 \equiv 0 \mod 1$. The proposed explanation is that our 4-point functions involve not only 4 primary fields, but also a combinatorial defect with parameter $w_s$ [38]. In the presence of this defect, $s$-channel diagonal fields $V_{P_s}$ such that $w(P_s) = w_s$ appear in the $s$-channel spectrum, and the structure constants of non-diagonal fields may depend on $w_s$. By combinatorial defect we mean a defect that only depends on which punctures are inside or outside the defect. This implies that the defect is topological, but not all topological defects are combinatorial, see Figure (5.3). In the $O(n)$ CFT on the torus, adding combinatorial defect would not spoil modular invariance, but the partition function would become a function of the defect's weight.

In particular CFTs, there may exist more or fewer solutions than combinatorial maps. For example, the map  corresponds to a solution with 1 $s$-channel diagonal field. In the $O(n)$ model, all diagonal fields are degenerate, and their fusion rules would forbid this solution, since $V_{(1,s)}^d \notin V_{(\frac{3}{2}, s_1)} V_{(\frac{1}{2}, s_2)}$ [25]. In the Potts model on the other hand, the spectrum includes degenerate fields of weights $n, -n$ and diagonal fields of weight 0. In some 4-point functions, diagonal fields with different weights can propagate in the same channel, leading to more solutions than maps [42].

### 5.2.3 Critical limits

Starting from a statistical model on a finite lattice, the large lattice limit may give rise to a conformal field theory, provided the lattice couplings are set to certain critical values. The same loop CFT can arise as the critical limit of statistical models with various lattice geometries, various rules for drawing loops on the lattice, and various parameters. These non-universal features are forgotten in the critical limit, and all these models belong to

the same universality class. (See the book [54] for an excellent introduction to these concepts.)

The existence of relevant fields in the CFT influences the critical limit. A primary field is called **relevant** if $\Re(\Delta + \bar{\Delta}) < 2$, **marginal** if $\Re(\Delta + \bar{\Delta}) = 2$, or **irrelevant** if $\Re(\Delta + \bar{\Delta}) > 2$. Any given lattice model has a finite set of couplings, which correspond to relevant fields of the CFT, and have to be tuned to their critical values for the critical limit to exist. If there exist relevant fields that do not correspond to any coupling, the critical limit ceases to exist, unless it is protected by some symmetry. Since the dimensions of primary fields depend on the parameter $n$ or $Q$, there exist **critical regions** in the complex $n$-plane or $Q$-plane where a given model has a critical limit, and the boundary of these regions are **marginality lines**, where a field becomes marginal.

Let us plot the marginality lines of a few primary fields, which will give us an idea on how critical regions can look like. For simplicity, we focus on degenerate diagonal fields $V^d_{\langle 1,s \rangle}$. In the $O(n)$, $PSU(n)$ and Potts models, these fields are invariant under the global symmetry, which therefore does not protect the critical limit from the corresponding couplings. The marginality line of $V^d_{\langle r,s \rangle}$ is

$$M_{(r,s)} = \left\{ \Re\Delta_{(r,s)} = 1 \right\} = \left\{ \Re\left(P_{(r+1,s-1)}P_{(r-1,s+1)}\right) = 0 \right\} . \tag{5.17}$$

In the complex $\beta^2$-half-plane, the line $M_{(1,s)} = \left\{ \Re\beta^{-2} = \frac{2}{s-1} \right\}$ is a circle with a center at $\beta^2 = \frac{s-1}{4}$, which goes through the origin. The field $V^d_{(1,s)}$ is irrelevant inside the circle, and relevant outside. Here we plot the circles for $s = 2, 3, \ldots, 9$ in the right half-plane (2.14), with dashed lines for the even values, which do not appear in the $O(n)$ CFT:

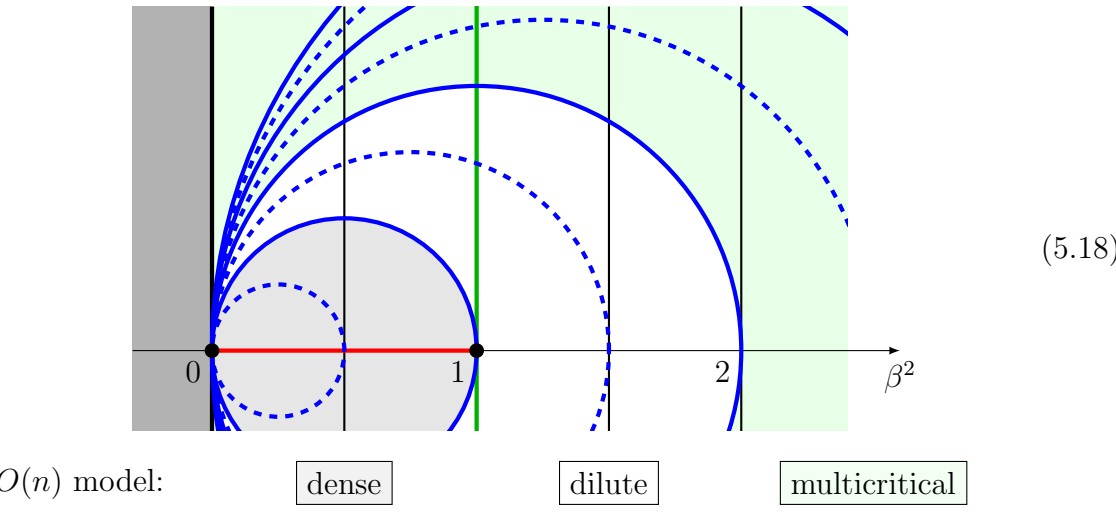

$$\tag{5.18}$$

$O(n)$ model: | dense | | dilute | | multicritical |

When it comes to the complex $n$-plane, we first notice that the relation (3.62) maps any strip $k < \Re\beta^2 \le k + 1$ to the full complex $n$-plane. This is one reason why the same CFT can describe different phases of a statistical model. For example, the $O(n)$ model's dense phase is usually defined for $\beta^2 \in (0,1)$, and the dilute phase for $\beta^2 \in (1,2)$, with $\beta^2 > 2$ corresponding to multicritical phases. The $O(n)$ model has 1 lattice coupling $K$, contributing a factor $K^{\#\{\text{occupied edges}\}}$ to the weight function (5.2), and corresponding to the field $V^d_{\langle 1,3 \rangle}$. In the dense phase, that field is irrelevant, and the critical limit is obtained for a range of values of the coupling. In the dilute phase it is relevant, and the critical limit is obtained for the critical value of the coupling, which is $K_c = \frac{1}{\sqrt{2+\sqrt{2-n}}}$ for the honeycomb lattice. In multicritical phases, criticality can only be achieved by adding extra couplings to the model, and fine-tuning their values.

In the diagram (5.18) we have extended these phases to the complex $\beta^2$-plane, by taking marginality lines as phase boundaries. Let us now plot the marginality lines in the

$n$-plane that corresponds to $0 < \Re\beta^2 < 1$:

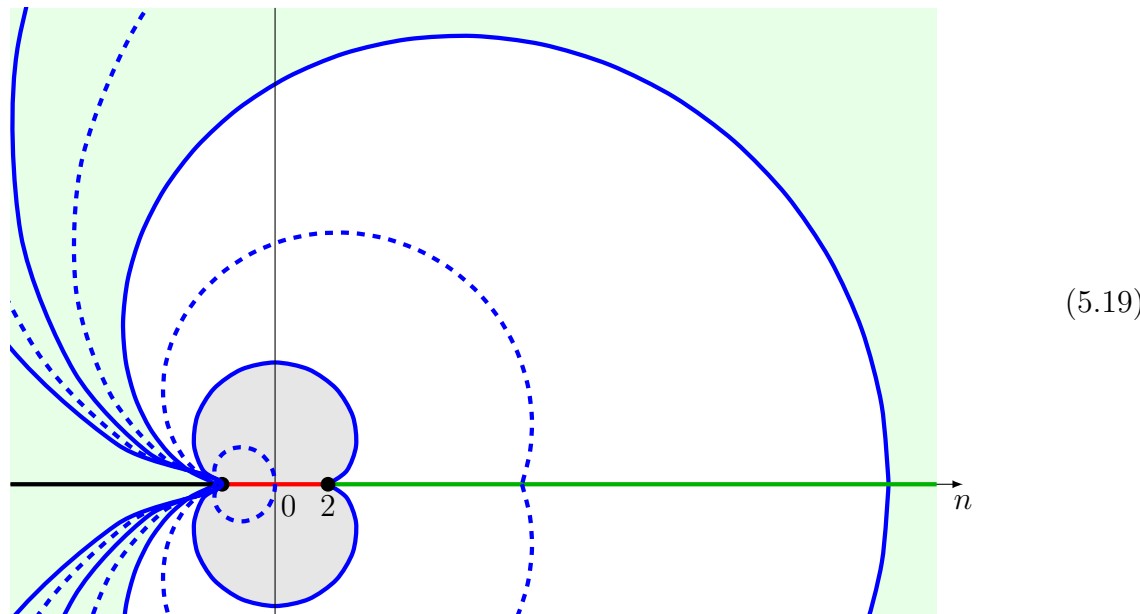

$$(5.19)$$

Numerical investigations of the zeros of the partition function suggest that the gray region within the criticality line $M_{(1,3)}$ is indeed the critical region of certain lattice loop models [55]. In fact, in the critical limit, zeros of the partition function condense not only on criticality lines, but also inside critical regions, on lines where the ground state changes. The ground state, i.e. the state with the lowest dimension (in real part), can be $V_{\langle 1,1 \rangle}^d, V_{(1,0)}, V_{(2,0)}$ or $V_{P_{(0,\frac{1}{2})}}$, depending on $\beta^2$ and on the model. On a line where the 2 lowest states have the same dimension, the free energy is singular.

A puzzle has arisen in the $O(n)$ CFT, when it was found that there is 1 field $V_{(2,0)}$ that is invariant under $O(n)$ [56]. From the point of view of the renormalization group flow triggered by the field $V_{\langle 1,3 \rangle}^d$, this field is *dangerously irrelevant*, i.e. it becomes relevant along the flow, and could prevent the flow from reaching the dense phase from the dilute phase. But numerical studies show that the flow does reach the dense phase. The puzzle is resolved by noticing that $V_{\langle 1,3 \rangle}^d$ is degenerate, and cannot produce the non-diagonal field $V_{(2,0)}$ by repeated fusion, even if $O(n)$ symmetry would allow it [57]. Therefore, it is global symmetry itself that is *dangerously irrelevant* to the understanding of this flow.

If we now consider correlation functions, the existence of a critical limit might depend on the case, and in particular on the weights $w$ of legless punctures. And which diagonal fields $V_P$ do we obtain in the limit? Since the relation (3.61) between $w$ and $P$ is not bijective, the answer is not obvious. (A simple possibility would be $0 \leq \Re(\beta P) \leq \frac{1}{2}$.)

## 5.3  Analytic properties of structure constants

Whenever crossing symmetry equations have a unique solution, numerical bootstrap calculations can give us numerical data for the corresponding 4-point structure constants. From these data, we would like to infer exact expressions. To do this, we will constrain structure constants by making assumptions on their analytic properties. These assumptions will determine each structure constant up to a polynomial in loop weights.

### 5.3.1 Analyticity in the channel momentum

We consider the 4-point function $Z(P) = \left\langle \prod_{i=1}^{4} V_{(r_i,s_i)} \right\rangle$, with an $s$-channel spectrum $\mathcal{S}_0(P)$ (5.15). The decomposition of $Z(P)$ into $s$-channel conformal blocks is

$$Z(P) = \sum_{k \in \mathbb{Z}} D_{P+k\beta^{-1}} \left| \mathcal{F}_{P+k\beta^{-1}} \right|^2 + \sum_{r=1}^{\infty} \sum_{s \in \frac{1}{r}\mathbb{Z}} D_{(r,s)}(P) \mathcal{G}_{(r,s)} . \tag{5.20}$$

Besides the Virasoro blocks $\mathcal{F}_P$, this involves conformal blocks $\mathcal{G}_{(r,s)}$ that gather the contributions of non-diagonal primary fields $V_{(r,s)}$ and their descendants, with $s \notin \mathbb{Z} \implies \mathcal{G}_{(r,s)} = \mathcal{F}_{P_{(r,s)}} \bar{\mathcal{F}}_{P_{(r,-s)}}$.

Analytic bootstrap equations from the existence of $V_{\langle 1,2 \rangle}^d$ imply $Z(P) = Z(P+\beta^{-1})$, so $Z(P)$ is in fact a function of the channel weight $w(P)$ (3.61), but we find it more convenient to write it as a function of $P$. In particular, we have $D_{(r,s)}(P) = D_{(r,s)}(P+\beta^{-1})$. Moreover, the analytic bootstrap equations determine how $D_P$ behaves under $P \to P + \beta^{-1}$. Let us assume that diagonal structure constants are reference structure constants,

$$D_P = \frac{C_{P(r_1,s_1)(r_2,s_2)}^{\mathrm{ref}} C_{P(r_3,s_3)(r_4,s_4)}^{\mathrm{ref}}}{B_P} , \tag{5.21}$$

where $B_P$ is given in Eq. (3.34) and $C^{\mathrm{ref}}$ in Eq. (3.54). For any given 4-point function $Z(P)$, this assumption only amounts to an overall choice of normalization.

Then we further assume that $Z(P)$ is a holomorphic function of $P \in \mathbb{C}$. From the point of view of loop models, this is a natural assumption, because $Z(P)$ is the critical limit of polynomials in loop weights. From the point of view of the CFT, this is a non-trivial assumption, because the diagonal terms have poles, which must therefore cancel. To begin with, according to Eq. (4.17), the Virasoro block $\mathcal{F}_P$ has a simple pole at $P = P_{(r,s)}$ with $r, s \in \mathbb{N}^*$, so that

$$\mathcal{F}_P = \frac{R_{r,s}^{\#}}{P - P_{(r,s)}} \mathcal{F}_{P_{(r,-s)}} + \mathcal{F}_{P_{(r,s)}}^{\mathrm{reg}} + O(P - P_{(r,s)}) \qquad \text{with} \qquad R_{r,s}^{\#} = 2P_{(r,s)} R_{r,s} . \tag{5.22}$$

Therefore, $|\mathcal{F}_P|^2$ has a double pole, with a residue proportional to $\left| \mathcal{F}_{P_{(r,-s)}} \right|^2$. Now, $\lim_{P \to P_{(r,s)}} Z(P)$ involves another contribution proportional to $\left| \mathcal{F}_{P_{(r,-s)}} \right|^2$, from the term $k = s$. And the structure constant $D_{P+s\beta^{-1}}$ does have a double pole at $P = P_{(r,s)}$, because $B_P$ has a double zero at $P = P_{(r,-s)}$. The cancellation of the double poles of $Z(P)$ requires

$$R_{r,s}^{\#} \bar{R}_{r,s}^{\#} D_{P_{(r,s)}} + \lim_{P \to P_{(r,-s)}} \left( P - P_{(r,-s)} \right)^2 D_P = 0 . \tag{5.23}$$

It can be checked that this equation is satisfied by the Virasoro block residues $R_{r,s}$ (4.18). This equation is relevant not only to loop CFTs, but also to (generalized) minimal models [18].

Simple poles of $Z(P)$ must cancel too, i.e. $\mathrm{Res}_{P=P_{(r,s)}} Z(P) = 0$ for any $(r, s) \in \mathbb{N}^* \times \mathbb{Z}$. At $P = P_{(r,0)}$, $B_P$ has a simple zero, and $D_P |\mathcal{F}_P|^2$ has a simple pole with a residue proportional to $\left| \mathcal{F}_{P_{(r,0)}} \right|^2$. This is cancelled by the term $D_{(r,0)}(P) \mathcal{G}_{(r,0)}$, provided $D_{(r,0)}(P)$ has a simple pole, and

$$\underset{P=P_{(r,0)}}{\mathrm{Res}} D_{(r,0)}(P) + \underset{P=P_{(r,0)}}{\mathrm{Res}} D_P = 0 . \tag{5.24}$$

Finally, let us consider the simple pole of $|\mathcal{F}_P|^2$ at $P = P_{(r,s)}$ with $r, s \in \mathbb{N}^*$. The residue at this pole includes the term $\mathcal{F}^{\mathrm{reg}}_{P_{(r,s)}} \bar{\mathcal{F}}_{P_{(r,-s)}}$, which is also present in $\mathcal{G}_{(r,s)}$, so that we must have

$$\mathop{\mathrm{Res}}_{P=P_{(r,s)}} D_{(r,s)}(P) + \bar{R}^{\#}_{r,s} D_{P_{(r,s)}} = 0 \ . \tag{5.25}$$

In fact, the equation $\mathrm{Res}_{P=P_{(r,s)}} Z(P) = 0$ with $r, s \in \mathbb{N}^*$ leads to the determination of the conformal block $\mathcal{G}_{(r,s)}$, which turns out to be logarithmic.

### 5.3.2 Logarithmic conformal blocks

For $r, s \in \mathbb{N}^*$, the residue $\mathrm{Res}_{P=P_{(r,s)}} D_P |\mathcal{F}_P|^2$ involves a number of terms, which cannot be canonically separated into the blocks $\mathcal{G}_{(r,s)}$ and $\mathcal{G}_{(r,-s)}$. Rather, this residue is a logarithmic conformal block, which sums up the contributions of all the states in 1 indecomposable representation that includes both primary fields $V_{(r,s)}$ and $V_{(r,-s)}$, see [13] for more detail on that representation. By convention we call $\mathcal{G}_{(r,s)}$ the logarithmic conformal block, and set $\mathcal{G}_{(r,-s)} = 0$, so that

$$\mathcal{G}_{(r,s)} = \frac{1}{\bar{R}^{\#}_{r,s} D_{P_{(r,s)}}} \left( \mathrm{Res}_{P=P_{(r,s)}} + \mathrm{Res}_{P=P_{(r,-s)}} \right) D_P |\mathcal{F}_P|^2 \ . \tag{5.26}$$

More explicitly, using Eq. (5.23), this can be expressed as

$$\mathcal{G}_{(r,s)} = \left( \mathcal{F}^{\mathrm{reg}}_{P_{(r,s)}} - R^{\#}_{r,s} \mathcal{F}'_{P_{(r,-s)}} \right) \bar{\mathcal{F}}_{P_{(r,-s)}} + \frac{R^{\#}_{r,s}}{\bar{R}^{\#}_{r,s}} \mathcal{F}_{P_{(r,-s)}} \left( \bar{\mathcal{F}}^{\mathrm{reg}}_{P_{(r,s)}} - \bar{R}^{\#}_{r,s} \bar{\mathcal{F}}'_{P_{(r,-s)}} \right)$$
$$+ R^{\#}_{r,s} \left( \frac{D'_{P_{(r,s)}}}{D_{P_{(r,s)}}} - \lim_{P \to P_{(r,-s)}} \left[ \frac{2}{P - P_{(r,-s)}} + \frac{D'_P}{D_P} \right] \right) \left| \mathcal{F}_{P_{(r,-s)}} \right|^2 \ , \tag{5.27}$$

where the derivatives are with respect to $P$, and $\mathcal{F}^{\mathrm{reg}}_{P_{(r,s)}}$ was defined in Eq. (5.22). The logarithmic nature of the block $\mathcal{G}_{(r,s)}$ comes from the asymptotic behaviour $\mathcal{F}_P(z) \underset{z \to 0}{\propto} z^{P^2}$ (1.102) of Virasoro blocks. When taking derivatives in $P$, or regularizing near a pole, this power-like prefactor gives rise to terms in $\log z$. Logarithmic terms are however absent if $R^{\#}_{r,s} = \bar{R}^{\#}_{r,s} = 0$, which occurs if fusion rules allow the degenerate field $V^d_{\langle r,s \rangle}$ to propagate in the $s$-channel of our 4-point function. In this case, the logarithmic block reduces to

$$\mathcal{G}_{(r,s)} \underset{R^{\#}_{r,s} = \bar{R}^{\#}_{r,s} = 0}{=} \mathcal{F}_{P_{(r,s)}} \bar{\mathcal{F}}_{P_{(r,-s)}} + \frac{R^{\#}_{r,s}}{\bar{R}^{\#}_{r,s}} \mathcal{F}_{P_{(r,-s)}} \bar{\mathcal{F}}_{P_{(r,s)}} \ . \tag{5.28}$$

This is a combination of the blocks for the Verma modules of $V_{(r,s)}$ and $V_{(r,-s)}$. Their relative coefficient $\frac{R^{\#}_{r,s}}{\bar{R}^{\#}_{r,s}}$ can be determined either by taking a limit in $s_i$ with $r_i$ fixed [25], or by using the shift equations (4.13), applied to $\frac{R^{\#}_{r,s}}{\bar{R}^{\#}_{r,s}} = \frac{D_{(r,-s)}}{D_{(r,s)}}$. Just like Virasoro blocks, logarithmic blocks can be combined into interchiral blocks of the type

$$\widetilde{\mathcal{G}}_{(r,s_0)} = \sum_{k \in 2\mathbb{N}} \frac{D_{(r,s_0+2k)}}{D_{(r,s_0)}} \mathcal{G}_{(r,s_0+2k)} \ , \tag{5.29}$$

where $s_0 \in \{0, 1\}$, and the ratios of 4-point structure constants are again determined by the shift equations (4.13).

### 5.3.3 Polynomials in loop weights

For a combinatorial map $M$, let us write the $s$-channel 4-point structure constants $D_{(r,s)}$ after factoring out the reference 4-point structure constant

$$D^{\text{ref}}_{(r,s)} = \frac{C^{\text{ref}}_{(r,s)(r_1,s_1)(r_2,s_2)}C^{\text{ref}}_{(r,s)(r_3,s_3)(r_4,s_4)}}{B^{\text{ref}}_{(r,s)}} \ , \tag{5.30}$$

which is a combination of the reference 2-point (3.55) and 3-point (3.54) structure constants. If $M$ has a vanishing $s$-channel signature $\sigma = 0$, then $D_{(r,s)}$ depends on the weight $w$ of $s$-channel loops. Furthermore, if $s \in \mathbb{Z}$, then $D_{(r,s)}$ has a pole at $w = w(P_{(r,s)})$, whose residue is determined by our assumption that the 4-point function is holomorphic as a function of $w$. Computing the residue using Eqs. (5.25) and (5.21), we find

$$D_{(r,s)} = D^{\text{ref}}_{(r,s)} \left( d_{(r,s)} - \delta_{\sigma,0}\delta_{s\in\mathbb{Z}} \frac{(-)^{(r+1)s}\rho^{r,s}_{(r_1,s_1)(r_2,s_2)}\rho^{r,s}_{(r_4,s_4)(r_3,s_3)}}{w - w(P_{(r,s)})} \right) \ , \tag{5.31}$$

where $d_{(r,s)}$ is a holomorphic function of $w$ if $\sigma = 0$, and we define

$$\rho^{r,s}_{(r_1,s_1)(r_2,s_2)} = (-)^{s\min(r,|r_1-r_2|)\delta_{r_1<r_2}} \prod_{\pm} \prod_{j=-\frac{r-1-|r_1\pm r_2|}{2}}^{\frac{r-1-|r_1\pm r_2|}{2}} 2\cos\pi\left(j\beta^2 + \frac{s-s_1\mp s_2}{2}\right) \ . \tag{5.32}$$

The crucial observation is that $\rho^{r,s}$ is a polynomial in $n$. Moreover, if $r_1 = 0$ or $r_2 = 0$, it is also a polynomial in the corresponding loop weight $w_1$ or $w_2$. This leads us to conjecture that $d_{(r,s)}$ is also a polynomial in $n$, in the weights $w_i$, and in the $s$-channel weight $w$: not only in the case $\sigma = 0, s \in \mathbb{Z}$ where $D_{(r,s)}$ depends on a channel weight and has a pole, but in all cases. Thanks to this conjecture, we can extract analytic expressions for $d_{(r,s)}$ from numerical results for $D_{(r,s)}$.

## 5.4 Examples of 4-point structure constants

We will now write the first few 4-point structure constants for 4-point functions that correspond to the combinatorial maps $\begin{smallmatrix}\times & \times \\ \times & \times\end{smallmatrix}$ , $\Big|\ \Big|$ and $\oslash$ . More examples can be found in [33].

### 5.4.1 Case of $\left\langle \prod_{i=1}^4 V_{P_i} \right\rangle$ $\begin{smallmatrix}\times & \times \\ \times & \times\end{smallmatrix}$

We have a unique solution of crossing symmetry for any 4 fields $V_{P_i}$ and channel fields $V_{P_s}, V_{P_t}, V_{P_u}$. Due to their invariance under shifts $d^{(x)}_{(r,s)} = d^{(x)}_{(r,s+2)}$ and parity $d^{(x)}_{(r,s)} = d^{(x)}_{(r,-s)}$, we need only write structure constants for $0 \leq s \leq 1$:

$$d^{(s,t,u)}_{\text{diag}} = 1 \ , \tag{5.33a}$$

$$d^{(s)}_{(1,0)} = w_t + w_u \quad , \quad d^{(t)}_{(1,0)} = w_s + w_u \quad , \quad d^{(u)}_{(1,0)} = w_s + w_t \ , \tag{5.33b}$$

$$d^{(s)}_{(1,1)} = w_u - w_t \quad , \quad d^{(t)}_{(1,1)} = w_u - w_s \quad , \quad d^{(u)}_{(1,1)} = w_s - w_t \ . \tag{5.33c}$$

$$2d^{(s)}_{(2,0)} = (n^2 - 4)\left[w_t^2 + w_u^2 + 2w_s - 4\right] - (n-2)(w_t + w_u)(w_1 + w_2)(w_4 + w_3)$$
$$- (n+2)(w_t - w_u)(w_1 - w_2)(w_4 - w_3) \ , \tag{5.33d}$$

$$2d^{(s)}_{(2,\frac{1}{2})} = n^2(w_u^2 - w_t^2) + n(w_t - w_u)(w_1 + w_2)(w_4 + w_3)$$
$$+ n(w_t + w_u)(w_1 - w_2)(w_4 - w_3) \ , \quad (5.33\text{e})$$

$$2d^{(s)}_{(2,1)} = (n^2 - 4)\left[w_t^2 + w_u^2 - 2w_s - 4\right] - (n + 2)(w_t + w_u)(w_1 + w_2)(w_4 + w_3)$$
$$- (n - 2)(w_t - w_u)(w_1 - w_2)(w_4 - w_3) \ , \quad (5.33\text{f})$$

Formulas for $d^{(s,t,u)}_{(3,*)}$ are known but rather complicated.

In the diagonal 4-point function $\left\langle \prod_{i=1}^4 V_{P_i} \right\rangle$, it is particularly easy to study the factorization (4.1) of 4-point structure constants into 3-point structure constants. In particular, we see that $d^{(s)}_{(2,*)}$ is a sum of 3 factorized terms. Due to pole terms, $D^{(s)}_{(2,0)}$ and $D^{(s)}_{(2,1)}$ are sums of 4 terms, while $D^{(s)}_{(2,\frac{1}{2})}$ is still a sum of 3 terms. And the presence of several terms is not due to combinatorial defects: in the case $w_s = w_t = w_u = n$ where the defects are absent, $D^{(s)}_{(2,*)}$ loses only 1 term.

The trivial-looking result $d^{(s,t,u)}_{\text{diag}} = 1$ (5.33a) implies that the diagonal 3-point structure constant $C_{P_1,P_2,P_3}$ reduces to the reference 3-point structure constant, which for $c \leq 1$ coindices with Liouville theory's 3-point structure constant. This coincidence of structure constants between loop models and Liouville theory, in spite of their very different properties, has been a source of puzzlement [58, 38]. Ultimately, the coincidence must be due to $C_{P_1,P_2,P_3}$ being the unique solution of the shift equations from $V^d_{\langle 2,1\rangle}$ and $V^d_{\langle 1,2\rangle}$. But in loop models, why would $C_{P_1,P_2,P_3}$ obey the shift equation from $V^d_{\langle 2,1\rangle}$? This degenerate field does not belong to the spectrum, and we cannot add it because its OPE with non-diagonal fields $V_{(r,s)}$ would generate fields with non-integer spins. In fact, this raises the more general issue of explaining why structure constants are built from the double Gamma function, which is characterized as the solution of 2 shift equations.

### 5.4.2  Case of $\left\langle V^4_{(\frac{1}{2},0)} \right\rangle$

We now have a diagonal field $V_{P_s}$ in the $s$-channel, but no diagonal fields in the other channels. Structure constants depend on $n$ and on $w_s$. They are known up to $r = 4$ in the $s$-channel and $r = 3$ in the $t, u$-channels. By invariance under the permutation $V_1 \leftrightarrow V_2$, we have $rs \equiv 1 \bmod 2 \implies D^{(s)}_{(r,s)} = 0$:

$$d^{(s)}_{\text{diag}} = 1 \ , \quad (5.34\text{a})$$
$$d^{(s)}_{(1,0)} = 0 \ , \quad (5.34\text{b})$$
$$d^{(s)}_{(2,0)} = n^2 - 4 \ , \quad (5.34\text{c})$$
$$d^{(s)}_{(2,1)} = -(n^2 - 4) \ , \quad (5.34\text{d})$$
$$3d^{(s)}_{(3,0)} = -8n^2(n - 2)^2(n + 2) \ , \quad (5.34\text{e})$$
$$3d^{(s)}_{(3,\frac{2}{3})} = 4(n^2 - 1)(n^2 - 3)(n - 2) \ , \quad (5.34\text{f})$$
$$2d^{(s)}_{(4,0)} = n^2(n - 2)^3(n + 1)^2(n + 2)$$
$$\times \left[w_s(n + 2)^2(n - 1)^2 + 2n^4 - 6n^2 - 8n + 16\right] \ , \quad (5.34\text{g})$$
$$2d^{(s)}_{(4,\frac{1}{2})} = -n^3(n^2 - 2)(n^2 - 3)$$
$$\times \left[w_s n(n^2 - 2)(n^2 - 3) - 4n^4 + 4n^3 + 8n^2 + 4n - 16\right] \ , \quad (5.34\text{h})$$
$$2d^{(s)}_{(4,1)} = n^2(n^2 - 4)^3(n - 1)^2 \left[w_s(n + 1)^2 - 2n^2 - 8n - 10\right] \ . \quad (5.34\text{i})$$

$$d^{(t)}_{(1,0)} = 1 \; , \tag{5.34j}$$

$$d^{(t)}_{(1,1)} = -1 \; , \tag{5.34k}$$

$$2d^{(t)}_{(2,0)} = (n-2)\left[w_s(n+2) - 8\right] \; , \tag{5.34l}$$

$$2d^{(t)}_{(2,\frac{1}{2})} = -n(w_s n - 4) \; , \tag{5.34m}$$

$$2d^{(t)}_{(2,1)} = w_s(n^2 - 4) \; , \tag{5.34n}$$

$$3d^{(t)}_{(3,0)} = n^2(n-2)^2\left[w_s^2(n+2)^2 - 4w_s(n+2) + n^2 + 8\right] \; , \tag{5.34o}$$

$$3d^{(t)}_{(3,\frac{1}{3})} = -(n-1)^2(n+1)$$
$$\times \left[(n^2-3)(n+1)w_s^2 - 2(n+1)(2n-3)w_s - 2(n-2)(n^2+4n+1)\right] \; , \tag{5.34p}$$

$$3d^{(t)}_{(3,\frac{2}{3})} = (n^2-3)(n^2-1)\left[w_s^2(n^2-1) - 2w_s(2n-1) - 2(n+1)(n-2)\right] \; , \tag{5.34q}$$

$$3d^{(t)}_{(3,1)} = -(n^2-4)^2\left[w_s^2 n^2 - 4w_s n + (n+2)^2\right] \; . \tag{5.34r}$$

### 5.4.3  Case of $\left\langle V_{(\frac{3}{2},0)} V_{(1,1)} V_{(1,0)} V_{(\frac{1}{2},0)} \right\rangle$ 

In this case there are no channel diagonal fields, and the structure constants only depend on $n$. To write the structure constants with $r \leq 3$, we use the golden ratio $\varphi = \frac{1+\sqrt{5}}{2} = 2\cos\left(\frac{\pi}{5}\right)$:

$$d^{(s)}_{(\frac{1}{2},0)} = 0 \; , \tag{5.35a}$$

$$3d^{(s)}_{(\frac{3}{2},0)} = n + 2 \; , \tag{5.35b}$$

$$3d^{(s)}_{(\frac{3}{2},\frac{2}{3})} = -2(n-1) \; , \tag{5.35c}$$

$$5d^{(s)}_{(\frac{5}{2},\frac{2}{5})} = 4\cos\left(\tfrac{\pi}{10}\right)\left[-n^4 + 4n^2 - 2 + \varphi(n+1)(n^2-3)\right] \; , \tag{5.35d}$$

$$5d^{(s)}_{(\frac{5}{2},\frac{4}{5})} = 4\cos\left(\tfrac{3\pi}{10}\right)\left[n^4 - 4n^2 + 2 + \varphi^{-1}(n+1)(n^2-3)\right] \; . \tag{5.35e}$$

$$d^{(t)}_{(1,0)} = d^{(t)}_{(1,1)} = d^{(t)}_{(2,0)} = d^{(t)}_{(3,0)} = 0 \; , \tag{5.35f}$$

$$2d^{(t)}_{(2,\frac{1}{2})} = \sqrt{2}n^2 \; , \tag{5.35g}$$

$$d^{(t)}_{(2,1)} = n^2 - 4 \; , \tag{5.35h}$$

$$3d^{(t)}_{(3,\frac{1}{3})} = (n-4)(n-1)(n+1)^2(n^2-3) \; , \tag{5.35i}$$

$$3d^{(t)}_{(3,\frac{2}{3})} = \sqrt{3}(n^2-1)^2(n+2)(n-3) \; , \tag{5.35j}$$

$$3d^{(t)}_{(3,1)} = 2n^2(n^2-4)^2 \; . \tag{5.35k}$$

$$d^{(u)}_{(\frac{1}{2},0)} = 1 \ , \tag{5.35l}$$

$$d^{(u)}_{(\frac{3}{2},0)} = d^{(u)}_{(\frac{5}{2},0)} = 0 \ , \tag{5.35m}$$

$$3d^{(u)}_{(\frac{3}{2},\frac{2}{3})} = -2\sqrt{3} \ , \tag{5.35n}$$

$$5d^{(u)}_{(\frac{5}{2},\frac{2}{5})} = 4\cos\left(\tfrac{\pi}{10}\right)\left[n^2 - 2 + \varphi\right] \ , \tag{5.35o}$$

$$5d^{(s)}_{(\frac{5}{2},\frac{4}{5})} = 4\cos\left(\tfrac{3\pi}{10}\right)\left[-n^2 + 2 + \varphi^{-1}\right] \ . \tag{5.35p}$$

The coefficients of these polynomials are trigonometric numbers, related to the values of the second Kac index.

## 5.5 Outlook

### 5.5.1 Towards solving loop CFTs

In the conformal bootstrap approach, solving loop CFTs on the sphere means being able to compute $N$-point functions for any $N \in \mathbb{N}^*$. If OPEs exist, this problem reduces to knowing the 3-point structure constants. (See Section 1.3.3.) However, most known results are for 4-point structure constants, as we saw in Section 5.4. The factorization of 4-point into 3-point structure constants can in principle be explored numerically, as we discussed in Section 4.1.3. But a case-by-case analysis of factorization would be tedious and not necessarily very illuminating, so long we do not understand the underlying algebraic structure.

Since correlation functions correspond to combinatorial maps, factorization should be formulated in terms of combinatorial maps. However, 3-point combinatorial maps on the sphere correspond to 3-point functions, not to 3-point structure constants. A 3-point function $\left\langle \prod_{i=1}^3 V_{(r_i,s_i)} \right\rangle$ only depends on the 3 fields' conformal dimensions, and for any $r_1, r_2, r_3 \in \frac{1}{2}\mathbb{N}$ such that $\sum r_i \in \mathbb{N}$ there exists exactly one 3-point combinatorial map on the sphere with vertices of valencies $2r_1, 2r_2, 2r_3$. On the other hand, 3-point structure constants may also depend on field multiplicities and fusion multiplicities. Since 4-point structure constants do not always factorize into products of 3-point structure constants, we know that some of these multiplicities are nontrivial. Therefore, 3-point structure constants should be described by richer combinatorial objects.

In fact, it is not even clear that OPEs exist in loop CFTs. The fact that 4-point functions are combinations of conformal blocks is a necessary condition for the existence of OPEs, but it is not sufficient. This condition may seem empty at first sight, because conformal blocks form a basis of functions of the cross-ratio. However, only blocks that correspond to channel fields in the loop CFTs' spectrum appear, and this makes the condition nontrivial. More fundamentally, we may try to deduce the existence of OPEs from the definition of statistical loop models. Technically, this might be done by constructing lattice loop models, and studying the algebraic properties of the resulting diagram algebras, see [59] for work in that direction.

If and when loop models can be solved at generic central charge, there will remain the challenge of taking rational limits. Such limits are relevant to applications such as percolation ($\beta^2 = \frac{2}{3}$) or polymers ($\beta^2 = \frac{1}{2}$). These special cases are more difficult than the general case, because representations of the Virasoro algebra have more complicated structures, see for instance [60]. Hints of these algebraic complications are visible in the fusion rules of degenerate representations, which are simpler for generic $\beta^2$ (1.64) than for rational $\beta^2$ (2.29). In the case of a 4-point function, we know that conformal blocks

diverge in the limit $\beta^2 \to \frac{q}{p} \in \mathbb{Q}$, but we expect that the divergences cancel so that the 4-point function converges. When a linear combination of divergent blocks $\mathcal{G} = \sum_k D_k \mathcal{G}_k$ converges, the interpretation is that the corresponding modules of the conformal algebra combine into a bigger indecomposable module in the limit. Such phenomenons also occur in CFTs with 2 degenerate fields, in particular in the limit GMM$\to$AMM of Section 2.5.2.

### 5.5.2 Global symmetries in $d = 2$ and $d \geq 3$

As we stated in Section 2.4.3, the $O(n)$, $PSU(n)$ and Potts CFTs have global symmetries, described by the groups $O(n)$, $PSU(n)$ and $S_Q$ respectively. Usually, in physics, we would expect such unbroken symmetries to have a strong influence. Models with different symmetry groups would behave differently, and we would formulate our analysis in group-theoretic language: spectrums as representations, correlation functions as invariant tensors.

However, the $O(n)$, $PSU(n)$ and Potts CFTs share many conformal dimensions and even some correlation functions. This is because they are all formulated in terms of similar loop ensembles, see Section 5.1.4. Moreover, if we decompose the spectrum of the $O(n)$ CFT into $O(n)$ representations, as we sketched in Eq. (2.72), we find that a primary field $V_{(r,s)}$ with a given conformal dimension generally does not transform in an irreducible representation, but in a larger representation. This shows that the model's global symmetry is larger than $O(n)$.

The origin of this larger symmetry is the requirement that loops do not intersect, as can be seen in the context of lattice models [27]. The lattice $O(n)$ model describes the dynamics of a periodic spin chain of length $L \in \mathbb{N}$, whose space of states is $\mathcal{H}_L = (\mathbb{R}^n)^{\otimes L}$. In the linear group $GL(\mathcal{H}_L)$, the commutant of $O(n)$ is the Brauer algebra $\mathcal{B}_L$, a diagram algebra where loops are allowed to intersect. Forbidding intersections, we would obtain a subalgebra of $\mathcal{B}_L$. The commutant of that subalgebra is larger than $O(n)$, and may be thought of as the global symmetry algebra of the lattice $O(n)$ model.

Invariant tensors of $O(n)$ can be represented in terms of loops that encode contractions of indices: for example, given 4 vectors $v_1, v_2, v_3, v_4 \in \mathbb{R}^n$, we can build the 3 invariants

$$(v_1 \cdot v_2)(v_3 \cdot v_4) \qquad\qquad (v_1 \cdot v_4)(v_2 \cdot v_3) \qquad\qquad (v_1 \cdot v_3)(v_2 \cdot v_4) \tag{5.36}$$

where $(v_i \cdot v_j)$ is the scalar product of 2 elements of the vector representation $\mathbb{R}^n$ of $O(n)$, in other words the image of $v_i \otimes v_j$ under the equivariant map from $\mathbb{R}^n \otimes \mathbb{R}^n$ to the identity representation. In order to represent all invariant tensors, we must allow loops to intersect: the geometry of the plane where we draw the loops is not relevant to representation theory. Forbidding intersections amounts to eliminating the invariant tensors that violate the larger symmetry.

When it comes to a 4-point function $\left\langle \prod_{i=1}^4 V_{(r_i, s_i)} \right\rangle$, the number of 4-point invariant tensors is larger than the number of solutions of crossing symmetry (= the number of combinatorial maps). If $\sum r_i = 2, 3$ these numbers actually coincide, while if $\sum r_i \geq 4$ the inequality becomes strict in most cases (for example, $6 > 5$ for $\left\langle V_{(\frac{3}{2},0)}^2 V_{(\frac{1}{2},0)}^2 \right\rangle$), and the number of solutions grows much slower with $r_i$ [43]. This slower growth is a manifestation of the larger symmetry at the level of 4-point functions.

In $d \geq 3$ dimensions, the $O(n)$ and $PSU(n)$ models can still be formulated in terms of loops, but from a distance there is no way to tell whether 2 loops intersect or not. The case

$d \geq 3$ is therefore similar to the case $d = 2$ with intersecting loops, and we expect no larger symmetry. In the critical $O(n)$ model for $d \geq 3$ or $d = 2$ with intersecting loops, we expect that a puncture with a given valency $2r$ gives rise to the same fields as in the $O(n)$ CFT, transforming in the same irreducible $O(n)$ representations. However, all their conformal dimensions are modified, due to the breaking of the larger global symmetry, the breaking of interchiral symmetry, and if $d \geq 3$ the breaking of local conformal symmetry down to global conformal symmetry. As a result, for any conformal dimension that appears in the spectrum, we expect 1 primary field that transforms in an irreducible representation of $O(n)$. If we write a 4-point function as a linear combination of 4-point invariant tensors, the coefficients are all solutions of crossing symmetry, and have no reason to be linearly dependent. Therefore, we expect as many solutions of crossing symmetry as 4-point invariant tensors.

In the $d \geq 3$ Potts model, the boundaries of clusters are not loops, but objects of dimension $d - 1 \geq 2$. Therefore, we do not expect the critical Potts model to share non-trivial conformal dimensions with the critical $O(n)$ and $PSU(n)$ models.

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

*A note on the identity module in* $c = 0$ *CFTs* (85)

# Index