# Peer review of "Exactly solvable conformal field theories"

_SciPost Physics Lecture Notes_

## Round 2 · Referee Report · Anonymous (Referee 1) · 2025-8-11

Strengths

1- This is an extensive and useful set of lectures covering basics and also current topics, written in the author's idiosyncratic style. The content is certainly worthy of publication as a set of lecture notes.

2- I found the sections on newer ideas such as interchiral blocks and combinatorial maps especially interesting. These are topics of current research and these lecture notes provide a very good introduction to the subject that goes beyond the original papers.

3- The discussion of numerical analysis was clear and helpful

4- To pick out just one other point, I liked the discussion of the Zamolodchikov recursion which I found far more informative than the original paper.

Weaknesses

1 - I think there are places where a literal reading of the text will lead to confusion and misunderstanding in the uninitiated. These notes are of course fine as the basis for a set of lectures where students can query the lecturer during the lectures, but as a stand-alone document I think they need to avoid any such places where a lone reader, on their own, will not be able to work out what is actually meant (when what is in the text is, as written, literally wrong).

2 - It is not clear which results on the existence of particular conformal field theories are rigorous and which are "probably" true. In the later sections on current research I think this is perfectly fine, but on the earlier sections on, say, minimal models, I found it unhelpful not to know whether the methods here are a genuinely rigorous alternative to other approaches.

Report

I think this meets the general criteria for SciPost Physics Lecture Notes and should be published in due course. There are places (see my comments) where there are possible mistakes, or where it is not as clear as it could be.

Requested changes

Warnings issued while processing user-supplied markup:

  • Inconsistency: plain/Markdown and reStructuredText syntaxes are mixed. Markdown will be used.
    Add "#coerce:reST" or "#coerce:plain" as the first line of your text to force reStructuredText or no markup.
    You may also contact the helpdesk if the formatting is incorrect and you are unable to edit your text.

1- Page 4, para 1. I think the book "Conformal Field Theory" was actually completed in 1996 and published in 1997, to go by the authors' comments at the start. I think that Philippe Di Francesco also consistently spells his name with a capital "D" to "Di", rather than "di" as here. Just a suggestion to check.

2- Page 5, para 4. I find the use of $\Re z$ for the real part of $z$ very strange, as does most of the internet; I would regard $\text{Re}\ z$ and ${\text Im}\ z$ as the actual de facto usage for the real and imaginary parts of a complex number $z$ and suggest using them here.

3- Page 6, para 3. Let me say just once that I find the use of "spectrums" very annoying. I think it detracts from the readability of the text, although not, of course, from its comprehensibillity.

4- Page 1, equation (1.5). I think there is a serious confusion here between an operator and the value it takes in a representation. If $\bf 1$ is an operator, and in particular a central operator, then it will take a single value in any irreducible representation, and that cannot be determined ahead of time. If it takes the value $1/2$ in a representation, then the central charge of that representation is (from (1.5)) $c/2$. In order for $c$ in (1.5) to be the actual central charge of a representation, we need $\bf 1$ to act as the identity in that representation and that is not said here. I can see that the author is trying to get away from having $\bf c$ as an operator and $c$ as its value in a representation, but he has done this at the expense of being confusing and incorrect, this should be fixed one way or another.

5- Section 1.1.2, page 8. I do not think that the author has made any actual mistakes in his logic, but I think he really should make the point at the start of section 1.1.2 that $L_0$ need not be diagonalisable in an indecomposable representation. He talks about "the eigenvalues" without saying that $L_0$ need not be diagonalisable, which could confuse the newcomer to the subject (and surely these lectures should not confuse the newcomer).

6- Page 8, Equation (1.1). The operator 1 is not in the set $\cal L$, as defined As defined, $\cal L$ consists of monomials of at least one generator $L_{n}$, as indicated by the lower limit $i=1$.

7- Page 10, table (1.16). Not an error, but.. - while the entries are correct, they are more complicated than need be and actually obscure the fact that the operators $L_{(r,s)}$ and $L_{(s,r)}$ take the same form when expressed in terms of $h$ and $c$, for example, both $L_{< 12> }$ and $L_{<21>}$ take the form

$$ L_{-1}^2 - \frac{4h+2}3 L_{-2} \;, $$
Both $L_{<13>}$ and $L_{<31>}$ take the form
$$ L_{-1}^3 - (2+2h)L_{-2}L_{-2} + h(1+h)L_{-3}
$$
I do not suggest necessarily including these expressions, but I do I think a comment to the effect that $L_{<rs>}$ and $L_{<sr>}$ are interchanged by $\beta \to 1/\beta$ and invariant under $\beta \to - \beta$ would be helpful.

8- Page 11, analysis after (1.21). This whole analysis relies on the possibility of an equality

$$ L_{<{r_3}{s_3}>}L_{<{r_2}{s_2}>} L_{<{r_1}{s_1}>} V_{\Delta_{(r_1,s_1)}} = L_{<{r_4}{s_4}>} V_{\Delta_{(r_4,s_4)}} $$
I would guess, from table (1.16), that the author has taken the conventional normalisation $L_{< rs >} = (L_{-1})^{rs} + \ldots$, but as far as I could tell, this is not stated. Sorry if I missed it, but the normalisation should be stated.

9- Page 11, analysis after (1.21). The relations between $r_1$ and $s_i$ look wrong to me. I just considered this one case, and it appeared to me there must be a mistake somewhere as it did not work for me:

$$ \beta=1\;,\;\; c=1\;,\;\; P_{(r,s)} = \frac 12 (r-s) \;. $$
and the equality, true for $(r_1,s_1)=(1,1)$, $ (r_2,s_2)=(1,3)$, $ (r_3,s_3)=(1,5), \Delta_{(1,1)}=\Delta_{(r_4,s_4)}=0$, and $(r_4,s_4)$ equal to either $(3,3)$ or $(-3,-3)$,
$$ L_{<15>} L_{< 13>} L_{< 11>} V_0 = L_{<{r_4}{s_4}>} V_0 $$
Then, \begin{align} &P_{(r_1,s_1)}=P_{(1,1)} = 0 \;,\;\; P_{(-r_1,s_1)}=P_{(-1,1)} = -1 \;,\;\; \ &P_{(r_2,s_2}=P_{(1,3)} = -1 \;,\;\; P_{(r_2,-s_2)}=P_{(1,-3)} = 2 \;,\;\; \ &P_{(r_3,s_3)} = P_{(1,5)} = -2 \;, \end{align} so
$$ \epsilon = \frac{P_{(-r_1,s_1)}}{P_{(r_2,s_2}} = 1 \;,\;\; \eta = \frac{P_{(r_2,-s_2)}}{P_{(r_3,s_3}} = -1 \;.\;\; $$
The possible correct choices for $(r_4,s_4)$ are $(3,3)$ and $(-3,-3)$, but if we follow the text we get $ r_4 = | \epsilon r_1 - r_2 + \eta r_3| = | 1\cdot1 - 1 + (-1)\cdot 1| =1 \;,\;\;$$ s_4 = | \epsilon s_1 + s_2 + \eta s_3|= |1\cdot 1 + 3 + (-1)\cdot 5| = 1 $ and $ r_4 s_4 = 1 \neq \sum_{i=1}^3 r_i s_i = 9 $.

10- Page 12, after (1.24) I do not know a reason why a level $N$ null vector has to have a non-vanishing component $L_{-N}V$, and without this, I do not see why a basis of $\cal R$ is given by all states which do not include $L_{-N}$. Is there a reference the author can give to justify this statement? I am worried it is not true

Just to show this is not a hypothetical possibility, in the case $c=1$, the singular vector $L_{< 22>} |0\rangle$ in the vacuum representation at level 4 is

$$ \left( (L_{-1})^4 +2 L_{-3}L_{-1} -4 L_{-2}(L_{-1})^2 \right) |0\rangle\;, $$
which does not include a term $L_{-4}|0\rangle$ and which I do not see lets one eliminate the mode $L_{-4}$.

11- Page 13, second paragraph. The map (1.28) is not a Lie algebra automorphism, because a Lie algebra automorphism (following the author's suggestion of just looking at Wikipedia),

   " an automorphism of a Lie algebra ${\displaystyle {\mathfrak {g}}}$ is an isomorphism from ${\displaystyle {\mathfrak {g}}}$ to itself, that is, a bijective linear map preserving the Lie bracket."

and clear it does not preserve the bracket because in the next line it can be seen that it introduces a minus sign.

I think the author might mean an anti-automorphism, since this changes the order of the bracket, equivalent to an overall minus sign.

12- Page 14, 1st para I understand what the author means by equation (1.33), but I think this will be very confusing to anybody who reads essentially any other text in which they will instead find

$$ \frac{\partial}{\partial z} V(z) = [L_{-1},V(z)] \;. $$
The author must surely have some notation for the action of the modes at zero, which is normally simply an unadorned $L_n$.

Using the ${}^{(z)}$ notation to indicated the action of a mode on the fields at $z$ is fine, but to avoid confusion it should be used consistently, so that (1.33) should be $\partial V(z)/\partial z = L^{(z)}_{-1}V(z)$ - otherwise I see nothing but confusion between (using the bracket notation of "Conformal Field Theory") $(L_{-n} V)(z) = L^{(z)}_{-n}V(z)$, (action at $z$), and $L_{-n}V(z) = L^{(0)}_{-n} V(z)$, (composition of operators).

13- Page 14. eqn (1.40) I could not find a definition of ${\cal S}$.

14- Page 15, eqn (1,43) I see that the operator $\bf 1$ from (1.5) has been dropped, I think at some stage it should be said, if (1.5) is kept, that $\bf 1$ takes the value 1 in any representation, either before this point, or (at the very latest) here.

15- Page 17, section 1.2.4 I am confused whether the author wants to make a difference between the degenerate state $V^d_{<11>}$ and the fully degenerate state $V^f_{< 11>}$. I would have thought, in a minimal model (p,q), that $V^d_{<11>}$, in which the null vector at level $(p-1)(q-1)$ is not removed, is not the identity field, and it is always the full-degenerate field $V^f_{<11>}$ that is the identity field. At least there should be some comment here why it is not $V^f_{<11>}$ that is the identity field.

16- Page 18, 1st para. I think the definition of fusion mutliplicity is either insufficient, or contradicts standard definitions. I would say that the fusion multiplicity of $R_3$ in $R_1 \times R_2$ is the number of allowed independent couplings in a consistent fusion. For irreducible representations, this is indeed either $0$ or $1$ in the Virasoro case, but I think the author has already shown that it can be higher then 1 when Verma modules are allowed.

In section 1.2.4 he has shown that in the fusion $R_1 \times R_1 \to {\cal V}_0$, the coefficient $C^{{L_{-1}}}$ is not determined, and hence the number of independent coefficients in this OPE is at least two, as $C^1$ and $C^{L_{-1}}$ are independent, and so $m_{R_1,R_1}^{{\cal V}_0} \geq 2$.

I think that it is irreducibility of representations that enforces $m_{i,j}^k \in {0,1}$.

This has implications too for the statement on page 19 that the fusion product of degenerate representations is a finite sum of degenerate representations. For example, in the case $c=1$, we have $h_{(1,3)}=1$, and (as vector spaces)

$$ {\cal R}^d_{<22>} \simeq_{V.S.} {\cal R}^d_{< 11>} + {\cal R}^d_{< 13>} \;,\;\; {\cal V}0 \simeq $$} \sum_{r=1} {\cal R}^d_{< 1d>
so the fusion
$$ {\cal R}^d_{< 21>}\,\times\,{\cal R}^d_{< 21>} \to {\cal R}^d_{<22>} $$
seems perfectly allowed with multiplicity two, as well as, possibly, as
$$ {\cal R}^d_{< 21>}\,\times\,{\cal R}^d_{<21>} \to {\cal V}_0 $$
again with multiplicity 2, at least from the discussion here.

I think the only way to resolve the actual structure of ${\cal R}^d_{<12>}\times{\cal R}^d_{< 12>}$ is the method of Gaberdiel in hep-th/9307183. Gaberdiel does discuss precisely this fusion, but only in the case that representations are irreducible and there is a non-zero scalar product, and so would exclude quite a few of the cases considered here (eg minimal models when ${\cal R}^d_{<12>}$ is degenerate but not fully degenerate.)

Just to summarise, I am not saying that I know that the results in the lectures are wrong, but rather that the arguments as presented do not seem to me to lead to the answers as presented, and I think this should be clarified, and if there is a published proof of the answers as presented here, that should be referred to.

18- Page 35, eqn (2.42) I think it should be made clear here that $q$ is given in terms of $p$ by (2.30).

19- Page 35, eqn (2.43) Is "$p,q\to\infty$" in the right place? Shouldn't it be under "lim"?

There must be a further constraint on $r$ than just (2.43) for $x \in (0,1)$. We could easily take $k=0$, or even $k=[\sqrt p]$, which are both technically $O(p)$, then $\lim r/p=0$. I would guess that the author has in mind $k= p k_0 + o(p)$, rather than just $k=O(p)$.

20- Page 39, table (2.64) Does the generalised model include the representation $V_0$, which has a non-zero null vector of weight 1? I am not sure whether the exclusion of finite-level null-states from representations is intended or not - if $(r,s)$ tend to infinity, then indeed I would think the Verma module is correct, are there not sum finite values of $(r,s)$ that survive this limit, for which the representation should be $V^d_{<rs>}$, not a Verma module? Maybe any confusion would be reduced (on a first reading) by a pointer to equation (3.42) to explain a resolution for Liouville theory (and other models too?)

21- Page 41, last para I am really unsure that the groups $O(n)$ for non-integer $n$ are defined. Having read the author's previous works, I believe that certain aspects of their structure and representations can be continued to complex $n$, such as representation labels, dimensions of representations, etc, but to say that each of these CFTs actually has a group of global symmetries seems to me to go too far.

22- Page 55, after eqn (3.61) Is "polynom" a typo for "polynomial?

23- Page 59, eqn (4.7) Is the $\pm$ missing on the $D$ in the first commutators, ie should this be

$$ [J_0,D^\pm]= \pm \beta^{-1} D^\pm $$

24- Page 86, 2nd para "So long we do not" should be "as long as we do not" or "so long as we do not?

25- Page 90 I find it very odd not to give the journal data for published articles, or publisher/doi for books. Presumably the author is not against journals per se, or why would he be submitting this article to a journal, then why not do the usual thing and give the data? I would hope that SciPost has an opinion on this.

Recommendation

Ask for minor revision

---

## Round 2 · Referee Report · Anonymous (Referee 2) · 2025-10-9

Report

Review on the Lecture Notes « Exactly solved conformal field theories » by Sylvain Ribault October 2025

Summary

These lecture notes deal with the computation of correlation functions in various two-dimensional CFTs, especially the non-rational ones, namely those with an infinite primary operator content.
  • Section 1 gives a basic introduction to CFT on the sphere, including the Virasoro algebra, the null vectors, the OPE and the fusion rules.

  • Section 2 begins with a review of rational and non-rational CFTs known in the literature : A- and D-series of minimal models and generalised minimal models, Runkel-Watts CFT, Liouville CFT. Particular emphasis is made on non-scalar fields in the OPE algebra. The discussion is restricted to CFTs with mutually local fields, which means that their correlation functions on the sphere are single-valued (unlike, for instance, the parafermionic CFTs). Then the author introduces a series of unsolved CFTs labeled by a continuous parameter, which he calls “loop CFTs”, because of their possible relation to the scaling limit of lattice loop models.

  • Section 3 describes the analytic derivation à la Teschner of the conformal bootstrap of four-point functions, in the presence of level 2 degenerate field(s). When inserting such a field in a four-point function, and imposing crossing symmetry, one obtains a shift equation on the OPE coefficients of other fields in the CFT. Although the shift equations only depend on the central charge and the conformal dimensions, the solution of the bootstrap problem depend strongly on the fusion rules. The analytical solution for some of the OPE coefficients in the CFTs of Section 2 is discussed.

  • Section 4 is devoted to the numerical approach of the 2d conformal bootstrap. The basic building blocks are the generic four-point Virasoro conformal blocks, whose power series are obtained numerically using Zamolodchikov’s recursion. Then, the crossing symmetry for a correlation function with a given set of intermediate states yields a numerical linear problem, whose unknowns are of the form C C’, namely the product of two OPE coefficients. Under some particular assumptions on the OPE algebra, the numerical difficulty can be reduced, by forming linear combinations of infinite series of conformal blocks, called “interchiral blocks”. This method provides numerical clues on fusion rules, and numerical estimates for some OPE coefficients, in non-rational and/or non-diagonal CFTs.

  • Section 5 gives arguments supporting some relations between loop CFTs and the actual scaling limit of lattice loop models. In particular, the operator content of the loop CFTs has been designed to include the scaling dimensions of connectivity operators of the lattice loop models. Then some recent progress on the OPE coefficients and the four-point functions of loop CFTs is exposed.

Comments

  1. If these lecture notes aim at teaching CFT to students, then Section 1 should include more physical intuition about the stress-energy tensor (for instance its interpretation as a response to a change of metrics), basic examples of calculations, a more explanatory description of Virasoro minimal models, and at least a basic introduction to CFT on the torus and modular invariance.

  2. Since a large part of the notes deal with “loop CFTs”, they should give some introduction to the physics of lattice loop and cluster models (phase diagram, connectivity operators, etc), and their relation to a compact scalar field with screening operators (“Coulomb gas”). Although it is largely heuristical, this approach yields the critical exponents, and it can be made quite rigorous in some cases.

  3. In Section 1 the usual terminology of representation theory is not always respected. For example, on p.8 the term “highest-weight representation” is used instead of “quotient module”. On p.9 the term “null vector” is given as a synonym of “singular vector”, but they have different meanings : a null vector is a vector which is orthogonal to the module, whereas a singular vector is a primary vector which generates a non-trivial submodule.

  4. In Section 2, some more introduction to Liouville CFT should be given, relating to the standard viewpoint on this theory. For instance, the Liouville Lagrangian is not even mentioned.

  5. In Section 3, the seminar work of Teschner should be cited.

  6. A discussion of indecomposable Virasoro representations is missing, especially in relation with “non-vanishing null vectors” of section 1.1.4. Also, some general statements about the structure of correlation functions and OPEs seem to consider only the case of irreducible representations. It is not clear if indecomposable representations are allowed or not in the author’s definition of “loop CFTs”, but section 5.3.2 suggests that they cannot be avoided. This would be consistent with [38], where it was argued that for some specific values of the loop weight, the lattice loop model scales to a logarithmic CFT. In any case, the case of indecomposable Virasoro representations should be treated seriously.

Conclusion

These notes provide a fair exposition of 2d non-rational CFTs and some modern techniques to solve them. The introduction to 2d CFT is too basic to be useful to a reader who is not familiar to CFT in the beginning, and contains inaccuracies concerning the Virasoro representation theory, especially for the very relevant case of indecomposable representations. There is no introduction to the physics of lattice loop and cluster models, which supposedly scale to the “loop CFT” which is studied in detail here.
These notes give an overview of the author’s recent progress on the subject, and can be helpful to researchers who want to apply the numerical conformal bootstrap in 2d, and/or are curious to understand the author’s viewpoint on non-rational CFTs.

Recommendation

Ask for minor revision

  • validity: -
  • significance: -
  • originality: -
  • clarity: -
  • formatting: -
  • grammar: -

Author:  Sylvain Ribault  on 2025-11-04  [id 5993]

(in reply to Report 2 on 2025-10-09)
Category:
question

Thank you for the suggested changes. Could you state more precisely what you mean by "basic examples of calculations" and "a more explanatory description of Virasoro minimal models"?

Anonymous on 2025-11-05  [id 5997]

(in reply to Sylvain Ribault on 2025-11-04 [id 5993])
Category:
answer to question

Indeed these are simply some suggestions. Some of the most basic examples of calculations in CFT are : - the partition functions and correlation functions of one, two or three primary operators on various domains - the equivalence between Virasoro commutation rules and the OPE T.T

For the minimal models, there is a description of A-series and D-series in Section 2, with a fair amount of examples. Since the focus is on classify the consistent CFTs with a given central charge, what ultimately determines the possible CFTs with non-diagonal spectra is the requirement of modular invariance, which leads to the ADE classification. I think it is disturbing to use the ADE terminology without explaining minimally what it refers to.

---

## Round 3 · Referee Report · Anonymous (Referee 2) · 2025-11-27

Report

The author has addressed my comments.

Recommendation

Publish (meets expectations and criteria for this Journal)

---

## Round 3 · Author Response

Warnings issued while processing user-supplied markup:

  • Inconsistency: plain/Markdown and reStructuredText syntaxes are mixed. Markdown will be used.
    Add "#coerce:reST" or "#coerce:plain" as the first line of your text to force reStructuredText or no markup.
    You may also contact the helpdesk if the formatting is incorrect and you are unable to edit your text.

Reply to Referee 1.

Weakness #1: Hopefully this is addressed to some extent by performing the requested changes, plus some more.

Weakness #2: I gave a generic answer to this concern by adding a paragraph in the introduction, starting from "While a CFT can be defined as a solution". A specific answer for each CFT under consideration is outside the scope of this text. The status of minimal models is not clear to me, it probably depends on the series.

1- Corrected as suggested, see C01.

2- No change: My notation is more compact and it is standard although less popular than the suggested notation.

3- No change: For spectrum, momentum, etc, regular plurals are correct and more logical than Latin nominative plurals (why nominative?), although less popular. I am sorry that they make the text less readable to some readers, but I believe they make things easier for an international audience.

4- Removed the confusion, see C03.

5- Clarified this point, see C04.

6- Clarified this point, see C05.

7- Added suggested comment, see C08.

8- Added a clause about normalization before what is now Eq. (1.23).

9- There was indeed a mistake in the formulas, which is now corrected by flipping the sign of $\epsilon$ in its definition.

10- Removed the offending statement, see C09. While probably true for generic central charge, it is not an essential statement.

11- Removed the offending statement, see C11.

12- No change. Fields are not defined as operators on a space, they are defined as vectors in representations of the Virasoro algebra. So there is no ambiguity, except for readers who have other formalisms in mind. Referring to these other formalisms would risk confusing less knowledgeable readers.

13- Removed $\mathcal{S}$.

14- Yes indeed, see C03.

15- Only added a statement after (2.33) about the identity field in minimal models. Not more, for fear of confusing readers with this subtle point. In a minimal model, there is no field $V^d_{\langle 1,1\rangle}$ distinct from $V^f_{\langle 1,1\rangle}$. In other words, the second singular vector of $V^d_{\langle 1,1\rangle}$ vanishes in all of the model's correlation functions. This would not be true if there existed fields outside the Kac table.

16- Clarified the definition of fusion multiplicity, see C12. I think the good definition is not the number of independent intertwiners, rather it should refer to a decomposition into indecomposables.

17- No remark with that number.

18- and 19- See C13 and C14.

20- When integrating over a continuous spectrum such as $P\in\mathbb{R}$, it does not matter what happens on a set of measure zero such as the set of degenerate momentums $P_{(r,s)}$. The question would matter if we were specifically interested in taking the limit $P\to P_{(1,1)}$ as in (3.42) in a correlation function, but in GDMMs it is not known what happens in this limit.

21- Right, see C18.

22- Fixed.

23- Fixed.

24- Fixed.

25- Mentioned the SciPost reviewers in the acknowledgements. These days, in this field, journal data are not needed for finding papers, and do not give much useful information in terms of quality or else. A journal can be a venue for improving a paper as we are doing right now, but this does not imply that journal data should be displayed to readers in the bibliography. This said, I can add the data if the editors so wish.

=============================

Reply to Referee 2.

1- This text is about the bootstrap approach. More intuition about the stress-energy tensor is not needed, and would require introducing the Lagrangian approach, so it is outside the scope. --- Basic calculations can be found in exercises. Where to find exercises is indicated in the introduction. --- I have added the origin of the A-D-E terminology for minimal models, see C19. --- Torus and modular invariance are outside the scope of this text, as announced in the introduction. The study of modular invariant partition functions was historically relevant, but I am rederiving the same spectrums in a simpler way.

2- For lattice loop models, I have added reference [36]. As suggested in the introduction, the Coulomb gas approach is only of historical interest, and irrelevant to current research. Results from that approach are rederived in a simpler way in this text.

3- Fixed terminology for representation theory, see C06. Fixed terminology on null and singular vectors: the correct term singular vector is now used throughout the text, while null vectors are only mentioned in the context of the Shapovalov form.

4- Other approaches to Liouville theory are outside the scope. See the Wikipedia article on Liouville theory (most of which I wrote) and references therein.

5- Cited in the introduction.

6- For indecomposable representations, see C10. --- It was already said explicitly in Section 2.4.2 that logarithmic representations appear in loop CFTs. I welcome the suggestion to strengthen this aspect. See changes C07 and C17.

---

## Round 3 · List of Changes

C01. "di Francesco, Mathieu and Sénéchal wrote a book" -> "Di Francesco, Mathieu and Sénéchal published a book"

C02. In the introduction, mentioned Teschner's seminal work on the analytic bootstrap in Liouville theory.

C03. In the Virasoro algebra's commutation relations (1.5), the central generator is made implicit, and the surrounding explanations modified accordingly.

C04. Added "(whether or not $L_0$ is diagonalizable in that representation)" when discussing eigenvalues of $L_0$.

C05. After (1.8), added the precision "where the unit operator $1\in \mathcal{L}$ is obtained in the case $k=0$".

C06. In Section 1.1.2, clarified the definition of a highest-weight representation.

C07. At the end of Section 1.1.2, defined logarithmic representations.

C08. Added Eqs. (1.17) and (1.18) on the behaviour of singular vectors under $\beta\to -\beta$ and $\beta\to \beta^{-1}$.

C09. After (1.26), removed the statement about the basis of the quotient representation.

C10. Before (1.27), added statement on reducibility and non-vanishing singular vectors.

C11. At the beginning of Section 1.1.5, the involution is no longer called an automorphism.

C12. Rewritten the second half of Section 1.2.4, in order to clarify the properties of the fusion product, and better define fusion multiplicities.

C13. Added precision on limit of fields in (2.37).

C14. Rewritten (2.42) more correctly.

C15. After (2.60), added comment on singular vectors.

C16. At the beginning of Section 2.4, made the distinction between critical loop models and loop CFTs.

C17. At the end of Section 2.4.2, added a reference on logarithmic representations, and a reference to the section on logarithmic blocks.

C18. Around Eq. (2.72), introduced the distinction between the group and the category of representations.

C19. In Section 2.5.3, mentioned the A-D-E classification of minimal models.

C20. Rewritten Section 3.3.3, removing the concept of reference structure constants.

C21. At the beginning of Section 4.2, added a second paragraph on interchiral symmetry, with a comparison to supersymmetry.

C22. Rewritten the introduction of Section 5.

C23. Revised Section 5.2.1, which now include the formulas for $n$ and $w(P)$.

C24. Rewritten Section 5.3.3.

C25. Simplified Section 5.5.1 in light of more recent developments.

C26. Terminology: replaced 'relative deviation' with 'deviation'.

C27. Around Eq. (1.23), added assumption on singular vector normalization, and flipped the sign of $\epsilon$ in its definition.

---

## Editorial Decision

in_refereeing